# A systematic assessment of uncertainties in large scale soil loss estimation from different representations of USLE input factors - A case study for Kenya and Uganda

Christoph Schürz[1], Bano Mehdi[1,2], Jens Kiesel[3,4], Karsten Schulz[1], and Mathew Herrnegger[1]

[1]Institute for Hydrology and Water Management, University of Natural Resources and Life Sciences, Vienna (BOKU), Vienna, Austria
[2]Institute of Agronomy, University of Natural Resources and Life Sciences, Vienna (BOKU), Tulln, Austria
[3]Leibniz Institute of Freshwater Ecology and Inland Fisheries (IGB), Berlin, Germany
[4]Institute of Natural Resource Conservation, Department of Hydrology and Water Resources Management, Christian Albrechts University of Kiel, Kiel, Germany

**Correspondence:** Christoph Schürz (christoph.schuerz@boku.ac.at); Mathew Herrnegger (mathew.herrnegger@boku.ac.at)

**Abstract.** The Universal Soil Loss Equation (USLE) is the most commonly used model to assess soil erosion by water. The model equation quantifies long-term average annual soil loss as a product of the rainfall erosivity $R$, soil erodibility $K$, slope length and steepness $LS$, soil cover $C$ and support measures $P$. A large variety of methods exist to derive these model inputs from readily available data. However, the estimated values of a respective model input can strongly differ when employing different methods and can eventually introduce large uncertainties in the estimated soil loss. The potential to evaluate soil loss estimates at a large scale are very limited, due to scarce in-field observations and their comparability to long-term soil estimates. In this work we addressed (i) the uncertainties in the soil loss estimates that can potentially be introduced by different representations of the USLE input factors and (ii) challenges that can arise in the evaluation of uncertain soil loss estimates with observed data.

In a systematic analysis we developed different representations of USLE inputs for the study domain of Kenya and Uganda. All combinations of the generated USLE inputs resulted in 972 USLE model setups. We assessed the resulting distributions in soil loss, both spatially distributed and on the administrative level for Kenya and Uganda. In a sensitivity analysis we analyzed the contributions of the USLE model inputs to the ranges in soil loss and analyzed their spatial patterns. We compared the calculated USLE ensemble soil estimates to available in-field data and other study results and addressed possibilities and limitations of the USLE model evaluation.

The USLE model ensemble resulted in wide ranges of estimated soil loss, exceeding the mean soil loss by over an order of magnitude particularly in hilly topographies. The study implies that a soil loss assessment with the USLE is highly uncertain and strongly depends on the realizations of the model input factors. The employed sensitivity analysis enabled us to identify spatial patterns in the importance of the USLE input factors. The $C$ and $K$ factors showed large scale patterns of importance in the densely vegetated part of Uganda and the dry north of Kenya, respectively, while $LS$ was relevant in small scale heterogeneous patterns. Major challenges for the evaluation of the estimated soil losses with in-field data were due to spatial and

temporal limitations of the observation data, but also due to measured soil losses describing processes that are different to the ones that are represented by the USLE.

## 1 Introduction

The Universal Soil Loss Equation (USLE, Wischmeier and Smith, 1965, 1987) formulates the most commonly applied concept to assess soil loss by water erosion (Alewell et al., 2019; Borrelli et al., 2017; Panagos et al., 2015e; Kinnell, 2010). The USLE is an empirical relationship that computes long-term average annual soil loss as a product of six input factors that characterize the erosive forces of the rainfall ($R$), the soil erodibility ($K$), topography ($L$ and $S$), plant cover ($C$), and support practices to mitigate erosion ($P$). Historically, the USLE succeeded earlier attempts to quantify soil erosion by water developed for the Corn Belt region of the United States of America (USA) in the 1940s. First relationships between soil loss on cropland and topography (Zingg, 1940), factors for crops and conservation practices (Smith, 1941), soil erodibility (Browning et al., 1947), and rainfall (Musgrave, 1947) were developed and reported by Wischmeier and Smith (1965). Over several decades extensive soil erosion data were collected in many locations on field plot scale in the USA. Eventually more than 10000 plot-years of field data were analyzed with reference to a "unit plot" to formulate a generally applicable approach for soil loss estimation in the USA (Wischmeier and Smith, 1965; Kinnell, 2010; Renard et al., 2011). The new approach overcame restrictions of previous methods for soil loss estimation to specific regions in the USA and thus was termed "universal" in the literature (Wischmeier and Smith, 1965). Further data were collected over the following decades and the methods to calculate the USLE input factors were substantially revised (Renard et al., 1991, 1997; Govers, 2011). This resulted in an update of the iso-erodent maps, the consideration of seasonality and rock fragments in the K factor, or a consideration of additional sub factors, such as prior land use, for the computation of the C factor (Renard et al., 1997). The revised model was termed as the Revised USLE (RUSLE, Renard et al., 1991). Yet, the general structure of the equation remained unchanged.

In the following we refer to USLE or RUSLE type models as USLE for simplicity. The different revisions of the USLE were summarized in Agriculture Handbooks (Wischmeier and Smith, 1965, 1987; Renard et al., 1997) that proved to be pragmatic and effective tools for soil conservation planning in the USA (Renard et al., 1991, 2011). Not without causing controversies, applications of the USLE model were extended to other land uses than cropland (Renard et al., 1991; Alewell et al., 2019), such as rangeland (Spaeth et al., 2003; Weltz et al., 1998), or woodland (Dissmeyer and Foster, 1980). Due to the principally simple implementation of the USLE model it found a wide application outside of the USA in more than 100 countries (Alewell et al., 2019) at various spatial scales and in various geoclimatic regions (Benavidez et al., 2018). Several studies adopted the methods to calculate the USLE input factors to meet local or regional conditions (e.g., Roose, 1975; Moore, 1979; Bollinne, 1985; Favis-Mortlock, 1998; Angima et al., 2003). Yet, to coin this empirical relationship as being "universal" is misleading for applications outside the USA and to non cropland (Jetten and Favis-Mortlock, 2006). The application of the USLE to conditions different from the plot experiments must be treated as a model extrapolation that is not supported by field data (Bosco et al., 2015; Favis-Mortlock, 1998).

It is well accepted that the USLE does not at all attempt to represent the physical processes to erode and transport soil particles, but rather empirically relates field properties to long term soil loss (Beven and Brazier, 2011; Kinnell, 2010). The USLEs' wide application does not distinguish it to be the best, or only option for soil loss estimation (Evans and Boardman, 2016a). Limitations of the USLE (but also other soil erosion models) have been well documented in the literature (see e.g. Boardman, 1996, 2006). Jetten and Favis-Mortlock (2006) summarize applications of the USLE in Europe, where the validation of calculated soil losses with observed data showed poor results (e.g., Favis-Mortlock, 1998; Bollinne, 1985). Nearing (1998) found that in general soil erosion models tend to over-predict small soil losses and under-predict large soil losses. Kinnell (2010) reports a good performance of a locally adapted variant of the USLE in New South Wales, Australia, but documents the over-prediction of small soil losses and under-prediction of large soil losses when applied to larger domains with a higher variability in agricultural systems (Tiwari et al., 2000; Risse et al., 1993). A recent pan-European soil loss assessment started a broad discussion of the validity of the estimates when compared to in-field soil loss assessments in Great Britain (see the discussion in Panagos et al., 2015e; Evans and Boardman, 2016a; Panagos et al., 2016; Evans and Boardman, 2016b). Several authors question the applicability of the plot scale based USLE to the landscape scale (e.g., Boardman, 2006; Evans, 1995; Govers, 2011), particularly as in large domains other processes such as gully erosion, bank erosion, or sediment deposition can dominate the erosion response (Govers, 2011). Multiple approaches are available from the literature that account, for instance, for the deposition of eroded material by employing concepts such as the sediment delivery ratio (e.g. Rajbanshi and Bhattacharya, 2020; Ferro and Minacapilli, 1995; Graham, 1975). While the USLE principally only accounts for the soil removal and does not consider soil deposition, Evans (2013) concludes that the USLE can be helpful to identify the erosion potential or erosion hot spots, but fails to predict the exact magnitude of soil that is eroded.

The above criticism does not impede the wide application of the USLE. For large scale erosion assessments, the availability of large scale spatial data and methods to infer the USLE inputs facilitate its implementation in GIS (Govers, 2011) and therefore is an attractive option to assess soil erosion. The implementation of remote sensing (satellite) products advances large scale soil loss assessments, particularly in data scarce regions where observations are limited as well as in large domains where in-field data acquisition is infeasible (Alewell et al., 2019; Bosco et al., 2015). This procedure yielded several continental and global estimates of USLE input factors (e.g., Panagos et al., 2017, 2015a, b, c; Vrieling et al., 2010) and soil loss assessments (e.g., Borrelli et al., 2017; Panagos et al., 2015e; Naipal et al., 2015; Yang et al., 2003; Van der Knijff et al., 2000) that were primarily derived from large scale (remote sensing) data products. The implemented remote sensing data products describe (or are a proxy for) features in the landscape (e.g. a DEM represent the topography and the NDVI is often employed to describe vegetation density). In large scale assessments, methods are implemented that employ these large scale data products to infer spatially distributed estimates for the USLE inputs. For each USLE input, various methods exist to generate the spatially distributed estimates for the USLE inputs that use different data sources (see e.g. the review of Benavidez et al., 2018). Thus, differing results in the realizations of a USLE input factor can result from the different computational approaches. However, a typical setup of the USLE combines only one representation of each USLE input in a single model setup and therefore does not depict the variations in the soil loss calculations that may arise from different representations of the USLE input factors.

Because of the multiplicative structure of the USLE, uncertainties in the input factors are decisive for the computation of the soil loss as they are propagated by multiplication (Sonneveld and Nearing, 2003).

Few studies have been conducted to analyze the uncertainties of the calculated soil loss and the sensitivities of soil loss estimates to the USLE input factors. Based on the original USLE data set Risse et al. (1993) performed a comprehensive study to assess the errors in the USLE estimates, evaluated the models' performance to calculate soil loss, and analyzed the influence of the USLE inputs on the model efficiency. Risse et al. (1993) identified the LS factor and the C factor as the most influential inputs. In a meta model study Keyzer and Sonneveld (1997) found that large errors in the soil loss estimates can be expected for high R and LS values, as well as for high and low values for the K factor due to low observation densities in these ranges for these input factors in the original USLE data. Continuing the work of Keyzer and Sonneveld (1997), Sonneveld and Nearing (2003) analyzed the robustness of the USLE model based on different subsets of the original USLE data set and found that the USLE model is not very robust. Falk et al. (2010) employed Bayesian melding to quantify the uncertainties in the soil loss estimates and to identify the USLE inputs that contribute the most to the uncertainties for a catchment in Eastern Australia. In their case study Falk et al. (2010) identified the LS factor to be the most influential USLE input. Based on nine nation wide soil loss data sets, including soil loss estimates for Europe (Panagos et al., 2015e), and the original USLE data set for the USA Estrada-Carmona et al. (2017) performed global sensitivity analysis to identify the dominant USLE input factors. For all nine countries Estrada-Carmona et al. (2017) found that the C factor and the LS factor were the most influential USLE inputs. Bosco et al. (2015) proposed a multi RUSLE model approach to account for the uncertainties in their soil loss estimates and therefore involve the impact of the different representations of the USLE inputs on soil loss estimation.

A widely applied procedure in environmental modelling to gain confidence in a model setup is model validation, which is the evaluation of calculated model outputs against observed data (Beven and Young, 2013; Young, 2001). Beven and Young (2013) further stress the importance of model falsification when a model fails to reproduce observations. For large scale soil loss assessments the possibilities to evaluate calculated soil losses, or spatially distributed estimates of the USLE inputs are very limited (Bosco et al., 2015; Van der Knijff et al., 2000). Typically, studies that monitored soil loss within the study domain rarely exist. Existing in-field data, however, entail issues of their spatial and temporal representativeness (Evans, 2013; Govers, 2011). Large scale meta-analysis studies of soil erosion plot data and sediment yield records exist, such as García-Ruiz et al. (2015), Vanmaercke et al. (2014) for Africa, or Maetens et al. (2012) for Europe. Yet, Boardman (2006) questions the comparability of erosion plot data or in-stream sediment yields with soil losses at the catchment scale. Govers (2011) highlights that USLE estimates reflect long time periods (Wischmeier and Smith (1965) e.g. recommended 20 years). Such time periods are usually not covered by a soil loss monitoring campaign. Eventually, USLE input factor estimates and large scale soil loss assessments are compared to very limited observation data (e.g., Borrelli et al., 2017; Vrieling et al., 2010; Moore, 1979) and in many cases no validation was carried out at all (e.g., Karamage et al., 2017; Van der Knijff et al., 2000).

Acknowledging that soil loss assessments using the USLE is uncertain and that the evaluation of soil loss estimates in large scale assessments has limitations, we formulate and systematically address the following objectives:

i. What are the uncertainties in soil loss estimates that we can expect from the implementation of different model input realizations in the USLE model? How can we interpret uncertain soil loss estimates?

ii. Which USLE model inputs contribute the most to the uncertainties of the soil loss estimates?

iii. How do the USLE ensemble model results compare to other single model studies?

iv. Can we compare the calculated soil loss estimates to in-field soil loss data? Does the evaluation enable us to reduce the uncertainties in the estimated soil losses?

We addressed these questions in a large scale soil loss assessment for Kenya and Uganda and structured our work in the following way: We reviewed methods to calculate USLE inputs that were widely used in previous large scale soil loss assessments and employed selected methods to generate spatially distributed estimates for the study domain (see section 3.2). All combinations of the input factor realizations delineate a USLE model ensemble. The analysis of the USLE ensemble results is outlined in the sections 3.4, and 4.1. We analyzed the impact of the USLE input factors $R$, $LS$, $K$, and $C$ on the calculated ranges of the soil loss estimates in a spatial analysis (see sections 3.5, 4.2, and 5.3). On the national level and for selected erosion prone counties of Kenya and districts of Uganda, we analyze the spatially aggregated mean soil loss estimates and compare them to the results of Fenta et al. (2020) on a national level and to the results of Karamage et al. (2017) on the administrative level for Uganda (sections 4.3 and 5.1). In a final step we compare the ensemble soil loss estimates derived with the USLE model ensemble to selected in-field erosion studies that were conducted in Kenya and Uganda (sections 4.4 and 5.4)

## 2  Study Area

The study area covers the countries of Kenya and Uganda, located in East Africa (Fig. 1). Overall the Sub-Saharan countries experienced drastic land degradation and a decrease in net-primary productivity of the land over the last decades (Bai et al., 2008). The dominant driver for land degradation in the horn of Africa is soil erosion by water (Jones et al., 2013). Large parts of Kenya and Uganda are generally prone to soil loss by water induced erosion.

In total, the study region covers an area of 821405 $km^2$, of which 729622 $km^2$ or 89 % of the surface are analyzed, since lakes and other water bodies are excluded from the analysis. Additionally, 27 administrative units in both countries (Fig. 1a), Table 1) are analysed in detail. The selection of the erosion prone administrative units is based on a visual analysis of Fig. 1a) and on local knowledge and on-site experience.

The study region covers a wide range of factors influencing soil erosion. Fig. 1a) shows the potential erosion risk solely stemming from topography, based on thresholds suggested by Ebisemiju (1988). Large areas with moderate to steep slopes ("moderate risk") are evident in the South-West of Uganda and in a north-to-south band in Kenya, where the Western or Gregory Rift as part of the Great Rift Valley transects the country. The South-West of Uganda is characterized by a hilly topography with low elevation differences. In contrast, the erosion prone regions in Kenya are mostly characterized by larger elevation differences, e.g. escarpments. Very steep slopes that exhibit a high risk of erosion from topography are evident around mountain massifs, e.g. Ruwenzori (5109 m a.s.l., Uganda), Mt. Elgon (4321 m a.s.l., Uganda and Kenya) or Mt. Kenya (5199 m a.s.l., Kenya). Additionally, high erosion risk prone areas are evident in the south-western corner of Uganda and along the Rift Valley in the northern part of Kenya. Fig. 1b) shows the mean annual MODIS NDVI (Didan, 2015) for the period 2001

**Table 1.** Administrative units analysed in more detail. The locations are shown in Fig. 1a). The slope and elevation statistics are based on SRTM v4.1 90m DEM (Jarvis et al., 2008).

| Nr. | Country | Greater Region | Administrative unit | Area (km$^2$) | Mean slope (deg) | Max. slope (deg) | Mean elev. (m) | Max. elev. (m) | Min. elev. (m) |
|---|---|---|---|---|---|---|---|---|---|
| 1 | Uganda | - | Kiruhura | 4636 | 4.39 | 28.96 | 1310 | 1670 | 1178 |
| 2 | Uganda | Lake Bunyoni | Ntungamo | 2062 | 7.57 | 43.61 | 1497 | 2224 | 1279 |
| 3 | Uganda | Lake Bunyoni | Kabale | 1740 | 14.79 | 46.15 | 1990 | 2601 | 1355 |
| 4 | Uganda | Lake Bunyoni | Kisoro | 733 | 11.95 | 49.44 | 1983 | 3861 | 1338 |
| 5 | Uganda | Lake Bunyoni | Kanungu | 1335 | 8.61 | 46.52 | 1388 | 2499 | 912 |
| 6 | Uganda | Ruwenzori | Kasese | 3402 | 8.81 | 60.54 | 1493 | 5034 | 878 |
| 7 | Uganda | Ruwenzori | Kabarole | 1825 | 8.01 | 48.94 | 1515 | 3996 | 626 |
| 8 | Uganda | Ruwenzori | Bundibugyo | 2265 | 5.65 | 52.24 | 1002 | 4659 | 612 |
| 9 | Uganda | - | Nebbi | 2922 | 3.71 | 34.70 | 1039 | 1873 | 612 |
| 10 | Uganda | - | Kaabong | 7301 | 5.87 | 61.41 | 1416 | 2720 | 834 |
| 11 | Uganda | Mt. Elgon | Bukwo | 529 | 12.28 | 53.35 | 2420 | 4204 | 1253 |
| 12 | Uganda | Mt. Elgon | Kapchorwa | 1215 | 8.00 | 53.39 | 1823 | 4265 | 1062 |
| 13 | Uganda | Mt. Elgon | Sironko | 1106 | 7.15 | 60.43 | 1619 | 4280 | 1045 |
| 14 | Uganda | Mt. Elgon | Bududa | 253 | 16.99 | 61.70 | 2103 | 4314 | 1216 |
| 15 | Uganda | Mt. Elgon | Mbale | 522 | 5.50 | 71.23 | 1288 | 2351 | 1083 |
| 16 | Uganda | Mt. Elgon | Manafwa | 606 | 8.34 | 57.77 | 1608 | 3319 | 1139 |
| 17 | Kenya | Mt. Elgon | Bungoma | 3036 | 5.15 | 45.12 | 1859 | 4304 | 1213 |
| 18 | Kenya | S-W Kenya | Kisii | 1353 | 6.24 | 32.83 | 1750 | 2190 | 1394 |
| 19 | Kenya | S-W Kenya | Nyamira | 897 | 6.70 | 31.99 | 1888 | 2214 | 1509 |
| 20 | Kenya | S-W Kenya | Bomet | 2384 | 5.14 | 30.29 | 1997 | 2465 | 1693 |
| 21 | Kenya | Cherangani Hills | Elgeyo-Marakwet | 3058 | 9.97 | 60.70 | 2122 | 3517 | 920 |
| 22 | Kenya | Cherangani Hills | West Pokot | 9328 | 8.70 | 67.15 | 1443 | 3524 | 691 |
| 23 | Kenya | - | Samburu | 21250 | 6.81 | 66.83 | 1185 | 2834 | 296 |
| 24 | Kenya | Mt. Kenya | Nyeri | 3380 | 7.39 | 54.88 | 2284 | 5035 | 1210 |
| 25 | Kenya | Mt. Kenya | Kirinyaga | 1491 | 4.41 | 45.27 | 1619 | 4747 | 1057 |
| 26 | Kenya | Mt. Kenya | Embu | 2780 | 4.89 | 38.56 | 1191 | 4760 | 520 |
| 27 | Kenya | - | Makueni | 8297 | 3.84 | 58.42 | 1065 | 2120 | 404 |

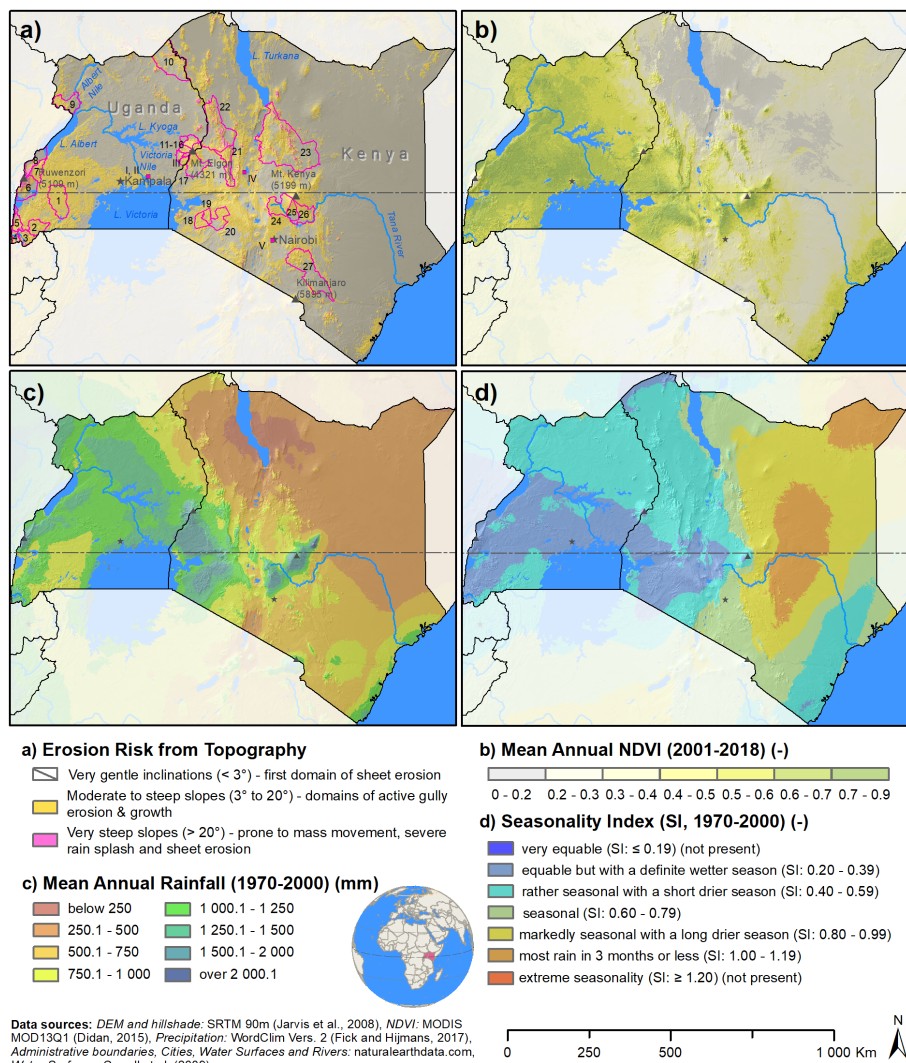

**a) Erosion Risk from Topography**

▱ Very gentle inclinations (< 3°) - first domain of sheet erosion

▢ Moderate to steep slopes (3° to 20°) - domains of active gully erosion & growth

▢ Very steep slopes (> 20°) - prone to mass movement, severe rain splash and sheet erosion

**c) Mean Annual Rainfall (1970-2000) (mm)**

- below 250
- 250.1 - 500
- 500.1 - 750
- 750.1 - 1 000
- 1 000.1 - 1 250
- 1 250.1 - 1 500
- 1 500.1 - 2 000
- over 2 000.1

**b) Mean Annual NDVI (2001-2018) (-)**

0 - 0.2  0.2 - 0.3  0.3 - 0.4  0.4 - 0.5  0.5 - 0.6  0.6 - 0.7  0.7 - 0.9

**d) Seasonality Index (SI, 1970-2000) (-)**

- very equable (SI: ≤ 0.19) (not present)
- equable but with a definite wetter season (SI: 0.20 - 0.39)
- rather seasonal with a short drier season (SI: 0.40 - 0.59)
- seasonal (SI: 0.60 - 0.79)
- markedly seasonal with a long drier season (SI: 0.80 - 0.99)
- most rain in 3 months or less (SI: 1.00 - 1.19)
- extreme seasonality (SI: ≥ 1.20) (not present)

**Data sources:** *DEM and hillshade:* SRTM 90m (Jarvis et al., 2008), *NDVI:* MODIS MOD13Q1 (Didan, 2015), *Precipitation:* WordClim Vers. 2 (Fick and Hijmans, 2017), *Administrative boundaries, Cities, Water Surfaces and Rivers:* naturalearthdata.com, *Water Surfaces:* Carroll et al. (2009)

250    500    1 000 Km

**Figure 1.** Study area of Kenya and Uganda. A classification of the soil erosion risk after Ebisemiju (1988) (a), the mean annual MODIS NDVI as a proxy for vegetation cover (b), mean annual rainfall (c), and the rainfall seasonality index (SI, Walsh and Lawler, 1981) (d) are plotted to characterize spatial properties of the study region. The boundaries for administrative units where the mean soil loss was assessed are shown with pink outlines in panel a). Locations of soil loss assessments from previous studies that were used for comparison are shown as pink squares. The hillshade is plotted in grey in the background to characterize the terrain topography.

- 2018 as a proxy for the vegetation cover. Higher values in NDVI show pixels with high vegetation cover, where a lower risk of water erosion due to ground cover can be assumed, and vice-versa. Kenya exhibits a large variability in NDVI with low values in the arid to semi-arid northern and south-eastern parts. Higher vegetation cover is present at the coast towards

the Indian Ocean, around Mt. Kenya, but also around Lake Victoria in the western part of the country. Uganda shows a rather homogeneous vegetation distribution, with some semi-arid areas in the north-east showing a lower vegetation cover.

Fig. 1c) shows the long-term mean annual rainfall (based on WorldClim Version2 for the period 1970 – 2000, Fick and Hijmans, 2017) as a proxy for the erosivity by rainfall. This assumes that larger annual rainfall values lead to higher erosion rates. Rainfall and vegetation cover are clearly connected. Hence, a more homogeneous rainfall pattern is visible for Uganda. Dryer areas in the south-west and north-east receive around $750 - 1000 \, \mathrm{mm \, yr^{-1}}$ of precipitation. The center of the country is wetter with around $1000 - 1500 \, \mathrm{mm \, yr^{-1}}$. In Kenya, wetter areas are evident around Lake Victoria and Mt. Kenya, receiving $1500 - 2000 \, \mathrm{mm \, yr^{-1}}$ or even higher. The northern part of the country only receives $250 - 500 \, \mathrm{mm \, yr^{-1}}$. Here, areas around Lake Turkana are very dry, with an annual precipitation of less than $250 \, \mathrm{mm \, yr^{-1}}$. In accordance with vegetation cover, the coast is wetter ($1000 - 1250 \, \mathrm{mm \, yr^{-1}}$). Between the coast and the central highlands, a dry belt is visible ($500 - 750 \, \mathrm{mm \, yr^{-1}}$). To accompany the distribution of the mean annual rainfall, the seasonality of the rainfall (SI, Walsh and Lawler, 1981) is illustrated in Fig. 1d). The rainfall around the Lake Victoria is classified as as equable with a definite wetter season. The rainfall in the remaining parts of Uganda and along the coast of Kenya is rather seasonal with a short drier season. North and central Kenya are markedly seasonal with long dry seasons and only short wet periods.

## 3 Methods and Data Basis

### 3.1 The Universal Soil Loss Equation (USLE)

The general form of USLE-type equation is as follows:

$$A = R \times K \times LS \times C \times P \tag{1}$$

where $A$ is the long-term average annual soil loss in $\mathrm{tons \, ha^{-1} \, yr^{-1}}$, $R$ is the rainfall erosivity in $\mathrm{MJ \, mm \, ha^{-1} \, h^{-1} \, yr^{-1}}$, $K$ is the soil erodibility factor in $\mathrm{tons \, h \, MJ^{-1} \, mm^{-1}}$, $L$ and $S$ are the unitless slope length factor and the slope steepness factor (that are usually evaluated together as the topographic factor $LS$ (Renard et al., 1997)), $C$ is the unitless cover management factor, and $P$ is the unitless support practice factor.

### 3.2 Estimation of USLE model inputs

To address the impact of different USLE input factor realizations on the simulation of the soil loss $A$, we generated a set of realizations for each of the four USLE input factors $R$, $K$, $LS$, and $C$. Methods to calculate the inputs were considered that were either used in previous large scale applications or that were specifically developed for Eastern Africa (or regions with similar climatic, topographic, and vegetation conditions). The implemented methods are described below. Further details to the input factor generation is provided in the supplementary materials section S.1. The support practice factor $P$ was excluded from the analysis, as large scale data to derive estimates for $P$ are very limited. Previous large scale studies, for example, inferred the $P$ factor from relationships with the land use (e.g., Yang et al., 2003), the land cover and slope (Fenta et al., 2020),

implemented a global estimate of $P$ for the entire study region (e.g., Karamage et al., 2017), or did not consider the $P$ factor (e.g., Borrelli et al., 2017).

The rainfall erosivity factor $R$ relates the intensity of rainfall events to the kinetic energy that is available to erode soil particles (Wischmeier and Smith, 1987; Panagos et al., 2015a). Rainfall intensity records are hardly available for large domains.
Thus, large scale erosion studies often employ long-term monthly average or long-term annual average precipitation sums to infer $R$. We implemented long-term monthly precipitation provided by WorldClim Version2 (Fick and Hijmans, 2017) with a spatial resolution of 30 seconds. The monthly precipitation sums were aggregated to a long-term annual precipitation. To account for the seasonality of the rainfall the monthly precipitation sums were employed to calculate the Modified Fournier Index (MFI, Arnoldus, 1980). In total, we considered six methods that relate long-term mean annual precipitation ($P_{annual}$)
to $R$ and one method that relates the $MFI$ to $R$ (Fig. 2a)).

Roose (1975) and Moore (1979) developed relationships between mean annual rainfall sums and $R$ based on station data in Western and Eastern Africa, respectively. Karamage et al. (2017) used the method developed by Lo et al. (1985) to calculate $R$ for Uganda. The method of Renard and Freimund (1994) was developed for USA precipitation station data and has been employed in global applications (e.g., Naipal et al., 2015; Yang et al., 2003). Nakil (2014) developed a relationship between
precipitation and $R$ for the highly variable rainfall patterns of the west coast of India. To assess and analyze the rainfall erosivity in Eastern Africa, Fenta et al. (2017) used two methods to infer $R$ from long-term annual precipitation and from the $MFI$, respectively. Additionally, we considered recent products by Panagos et al. (2017) and Vrieling et al. (2014) that inferred $R$ estimates from high temporal precipitation data. While Panagos et al. (2017) derived global estimates for $R$ on a 1km grid based on a large global rainfall intensity data set to assemble the GloREDa data base, Vrieling et al. (2014) used the 3 hourly
TRMM Multi-satellite Precipitation Analysis (TMPA) product (Huffman et al., 2007) to infer $R$ estimates for the African continent in a 0.25° spatial resolution. In total we included seven realizations for $R$ in this study (Fig. 2 a)).

The soil erodibility factor $K$ describes the tendency of a soil to erode due to the erosive force of precipitation or surface runoff and can be related to soil physical and chemical properties (Panagos et al., 2014). Direct assessments of the soil erodibility are only available at a plot scale. Large scale erosion studies employ transfer functions that infer the soil erodibility from soil
properties that are easier to acquire. Several global soil data products are available that provide physical and chemical soil properties with different spatial resolution. We implemented soil information from SoilGrids250m (Hengl et al., 2017) and the Global Soil Dataset for use in Earth System Models (GSDE, Shangguan et al., 2014). Layers of mass fractions of sand (Sa), silt (Si), and clay (Cl), the soil organic carbon content (orgC) and the fraction of coarse fragments (CRF) were acquired for the available soil depths and weighted average values for 0-10 cm were calculated. The aggregated soil layers were used in three
transfer functions that were employed in previous large scale studies to compute $K$. We applied the method of Wischmeier and Smith (1987) and followed the procedure suggested by Panagos et al. (2014) and Borrelli et al. (2017) to compute $K$ from the SoilGrids250m layers. The method of Wischmeier and Smith (1987) requires Sa, Si, Cl and organic matter content (OM) as inputs. Additionally, information on soil structure (s) and soil permeability (p) is relevant. Borrelli et al. (2017) derived these properties from soil classes according to the World Reference Base (WRB) and the USDA soil texture classification
systems that are available for SoilGrids250m. GSDE does not provide soil class layers. Thus, the parameters s and p were

kept constant when using the GSDE as input, following a procedure by Tamene and Le (2015). We further implemented the methods of Williams (1995) and Torri et al. (1997). Both methods require values of Sa, Si, Cl and OM as inputs. The soil products SoilGrids250m and GSDE in combination with three transfer functions resulted in six realizations of the $K$ factor (Fig. 2b)).

The slope length and steepness factor $LS$ represents the influence of the terrain topography on soil erosion (Panagos et al., 2015b). A digital elevation model (DEM) is the basis to derive the $LS$ factor. In this study we implemented the SRTM v4.1 90m DEM (Jarvis et al., 2008) and the ASTER GDEM V2 (NASA/METI/AIST/Japan Spacesystems, and U.S./Japan ASTER Science Team, 2009) with a 30m resolution. ASTER GDEM V2 data was aggregated and projected to the 90m grid of SRTM v4.1 for comparability, but also because our computation capacities were insufficient to calculate soil erosion rates on a 30m
grid for the study extent. Three methods were applied from Moore et al. (1991), Desmet and Govers (1996), and Böhner and Selige (2006) that are available from the System for Automated Geoscientific Analyses (SAGA) v. 2.1.4 (Conrad et al., 2015). Together with the two DEM products six realizations of the $LS$ factor (Fig. 2c)) were computed. Intermediate steps such as the reprojection of the ASTER GDEM V2, DEM fill, the calculation of flow direction or flow accumulation were processed in ArcMap 10.6 (ESRI, 2012). In the calculation of $LS$ using the method of Desmet and Govers (1996) we followed the steps
described in Panagos et al. (2015b). The use of ASTER GDEM v2 introduced strong noise in the computed $LS$ layers that results from artifacts in the remote sensing data. Particularly, the computed soil erosion in flat areas was strongly affected by the noise signal, rendering the results unusable (see section S.3 and Fig. S.1 in the supplementary document). Thus, we excluded the $LS$ realizations using ASTER GDEM v2 in the analysis and only considered three out of the six generated realizations for the $LS$ factor (Fig. 2 c)).

The cover management factor $C$ subsumes the impacts of vegetation cover and land management on soil erosion (Wischmeier and Smith, 1987; Panagos et al., 2015c). For large scale studies we identified two main approaches to compute $C$ (Fig. 2d)); i) to infer $C$ from vegetation indices from satellite based remote sensing products (e.g., Karamage et al., 2017; Naipal et al., 2015; Tamene and Le, 2015; Van der Knijff et al., 2000) and ii) to join land cover classification products with agricultural statistics and $C$ factor literature values to compile a continuous $C$ factor layer (e.g., Borrelli et al., 2017; Panagos et al., 2015c;
Bosco et al., 2015; Yang et al., 2003).

For the computation of $C$ from NDVI vegetation indices we implemented the method of Van der Knijff et al. (2000), who proposed a non linear relationship between NDVI and $C$. We acquired 16 day MODIS NDVI averages (Didan, 2015) from 2000 to 2012 and aggregated them to a mean NDVI layer. We calculated the annual mean NDVI (see e.g., Van der Knijff et al., 2000; Tamene and Le, 2015) and the mean NDVI averages over the two rainy seasons March to May and October to November
as proposed by Karamage et al. (2017). Both long-term mean NDVI layers were used to compute $C$ factor realizations with the equation of Van der Knijff et al. (2000).

Two land cover products, the MODIS Collection 5 LC with a spatial resolution of 250m (Channan et al., 2014; Friedl et al., 2010) and the ESA CCI LC Map v2.0.7 with a spatial resolution of 300m (ESA, 2017) served as base land cover layers. The agricultural, forest, and naturally vegetated land cover in these maps were superimposed with C factor literature values. The C
factor values for agricultural land uses were calculated based on agricultural statistics. Two agricultural statistics were used that

provide information on crop areas at different spatial scales. i) National agricultural surveys for Kenya on ward level (KNBS, 2015) and for Uganda on county level (UBOS, 2010) were harmonized. ii) Monfreda et al. (2008) provides global gridded crop shares of 175 crops with a spatial resolution of 5 minutes. We assigned $C$ factor literature values from Panagos et al. (2015c) and Angima et al. (2003) to all crops found in the national agricultural surveys and the grid layers from Monfreda et al. (2008).

Based on the crop shares in the administrative units of Kenya and Uganda and for the crop shares in each grid cell of Monfreda et al. (2008) we calculated weighted average $C$ values as proposed in Panagos et al. (2015c). $C$ values for non agricultural land uses of the MODIS LC were estimated according to Panagos et al. (2015c) varying the $C$ values for forest between boundaries based on the MODIS vegetation continuous fields (VCF) tree cover product. ESA CCI LC classifies the land cover as shares between different land uses (e.g. Mosaic cropland (>50%) / natural vegetation (tree, shrub, herbaceous cover) (<50%)). In this

case, $C$ values were estimated by calculating weighted averages between the calculated average $C$ values for agricultural areas and literature values (Panagos et al., 2015c) for non agricultural land uses according to the given fractions of the land cover classes. The combination of the two land cover products and the two agricultural statistic products resulted in four realizations for the $C$ factor.

### 3.3   Estimation of soil loss

In total 9, 6, 6 (3), and 6 realizations were generated for the USLE input factors $R$, $K$, $LS$, and $C$, respectively. The combination of all input factors to assemble USLE model setups would have resulted in 1944 realizations of the USLE model. The $LS$ factor realizations that were generated with the ASTER GDEM V2 were however excluded from the model ensemble, as they showed large noise ratios. The number of analyzed USLE model setups was therefore halved to 972. For the overlay of the generated USLE input layers, all layers were reprojected to the grid of the SRTM v4.1 90m DEM and the long-term mean annual soil

loss $A$ was calculated for all model combinations in the study region of Kenya and Uganda using Eq. 1.

### 3.4   Analysis of spatially distributed soil loss estimates

The ensemble of 972 spatially distributed soil loss estimates with spatial resolution of 90 m were summarized in each grid cell employing descriptive statistical measures. In each grid cell we calculated mean and median values to estimate an average soil loss from the USLE model ensemble. The range of the minimum and maximum soil loss $A$ in a grid cell indicates the variation of the ensemble simulations in a grid cell (i.e. the disagreement between the model setups).

A common concept in the erosion literature is to relate soil loss to soil formation rates and therefore classify the soil loss as sustainable (tolerable) or non-sustainable (e.g., Blanco-Canqui and Lal, 2008; Montgomery, 2007; Van-Camp et al., 2004), or to group soil loss based on the severity of soil removal (e.g., Zachar, 1982; FAO-PNUMA-UNESCO, 1980). Literature values for tolerable levels of soil loss vary between 5 and 12 tons ha$^{-1}$ yr$^{-1}$ on a global scale (Montgomery, 2007; Blanco-Canqui and Lal, 2008; Zachar, 1982). Karamage et al. (2017), Bamutaze (2015), Morgan (2009), or Lufafa et al. (2003) used 10 tons

ha$^{-1}$ yr$^{-1}$ as a threshold value to classify tolerable soil loss for studies conducted in Eastern Africa. In this study low soil losses were classified by employing the same threshold. Yet, no information on soil formation was included and thus the term

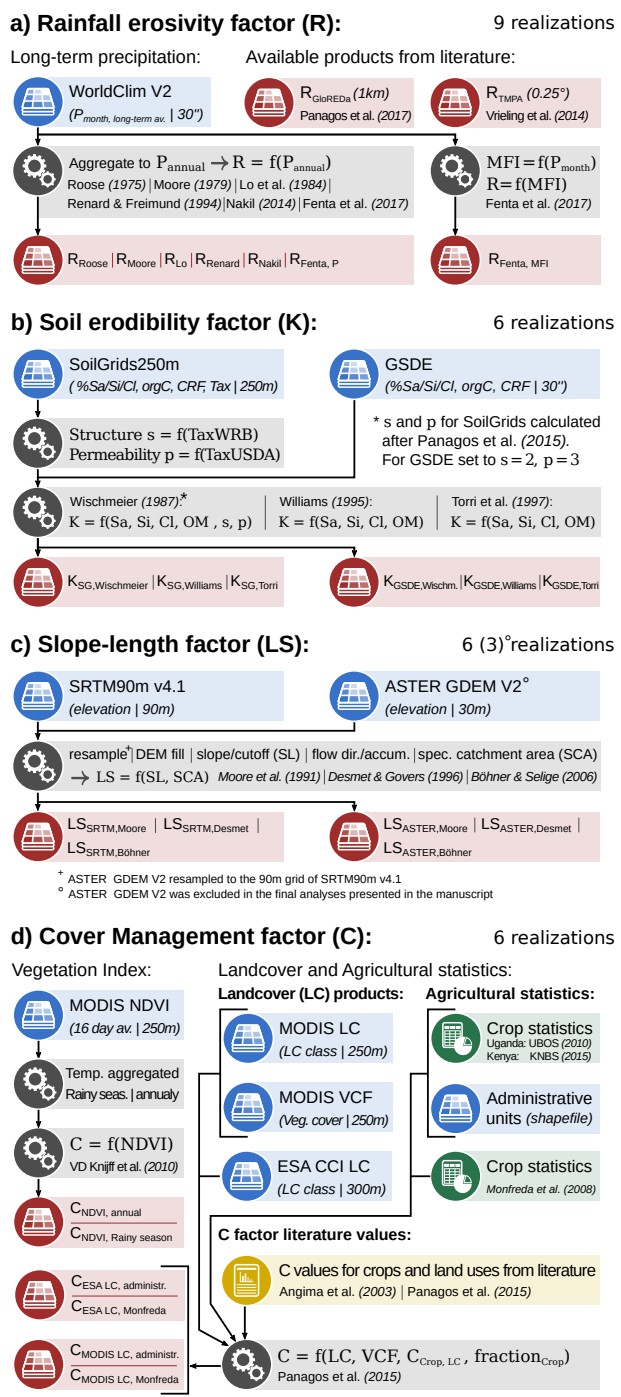

**Figure 2.** Methodological framework to generate the realizations of the USLE model input factors $R$, $K$, $LS$, and $C$.

*tolerable* is misleading. Consequently a soil loss between 0 and 10 tons ha$^{-1}$ yr$^{-1}$ is defined as *slight* soil loss, as suggested by Fenta et al. (2020).

For soil loss levels larger than 10 tons ha$^{-1}$ yr$^{-1}$ we implemented the soil removal classification after FAO-PNUMA-UNESCO (1980, implemented e.g. in Hernando and Romana (2015) or Olivares et al. (2016)) where a soil loss between 10 and 50 tons ha$^{-1}$ yr$^{-1}$ is considered to be moderate, a soil loss between 50 and 200 tons ha$^{-1}$ yr$^{-1}$ to be high, and a soil loss larger than 200 tons ha$^{-1}$ yr$^{-1}$ to be severe. In each grid cell we classified the simulated soil losses from the 972 USLE model setups into the four defined soil loss classes and calculated the frequencies for each soil loss class as follows:

$$f_{i,m,n} = \begin{cases} 0 & \text{if } A_{i,m,n} \notin [A_{class,lower}; A_{class,upper}) \\ 1 & \text{if } A_{i,m,n} \in [A_{class,lower}; A_{class,upper}) \end{cases} \tag{2}$$

$$f_{m,n} = \frac{\sum_{i=1}^{N} f_{i,m,n}}{N} \tag{3}$$

where $f_{m,n}$ is the frequency of models that calculated a soil loss between the defined boundaries $A_{class,lower}$ and $A_{class,upper}$ of the respective class in the grid cell $(m,n)$ and based on the $N = 972$ USLE model setups. A step function assigns the probabilities $p_{i,m,n} = 1$ or $p_{i,m,n} = 0$ to a model $i$ if the soil loss $A_{i,m,n}$ that was calculated with the model $i$ for the grid cell $(m,n)$ is included or excluded from a class interval.

### 3.5 Analysis of the USLE input factors

In the case of a simple model, such as the USLE, uncertainties in the inputs can be analytically propagated through the model to infer the uncertainties in the outputs (Beven and Brazier, 2011). Thus, the sensitivity of the calculated soil loss for the ranges of the input factors can be analyzed analytically. We assessed the importance of the USLE input factors on the simulation of the soil loss in each grid cell by calculating the fraction between the range in soil loss that is caused by an input factor $I_j$ and the total range of $A$ that results from the entire model ensemble in that grid cell:

$$s_{j,m,n} = \frac{(\max(I_{j,m,n}) - \min(I_{j,m,n})) \cdot \prod_{k \neq j} \max(I_{k,m,n})}{\left( \prod_{k} \max(I_{k,m,n}) - \prod_{k} \min(I_{k,m,n}) \right)} \tag{4}$$

where $s_{j,m,n}$ is the sensitivity of the input factor $I_j$ in the grid cell $(m,n)$, $I$ is the set of the analyzed input factors $R$, $K$, $LS$, and $C$, and $k$ is the index of the respective input factor. The resulting sensitivity measure is normalized between 0 and 1, where a sensitivity $s_{j,m,n} = 1$ means that the total range of the calculated soil loss can result from varying the input $I_j$ and 0 means that this input shows no variation between its realizations in the grid cell $(m,n)$. In each grid cell the input factors are ranked based on their sensitivities and visualized to get a spatial reference of the importance of the model inputs.

## 3.6 Soil loss assessment at administrative levels and comparison to other studies in Uganda and Kenya

We assessed the soil loss on a national level for Kenya and Uganda as well as on an administrative levels for 27 administrative units in Uganda and Kenya. An aggregation of the calculated soil losses to clearly defined spatial units allowed a comparison of the USLE model ensemble results to previous erosion studies in Kenya and Uganda that employed single USLE model setups and evaluated the soil losses for these spatial domains. On a national level we compared the USLE model ensemble results to the results presented in Fenta et al. (2020). For the comparison we employed the descriptive statistical measures that were computed spatially distributed for the study area in section 3.4. The spatially distributed soil loss quantiles were aggregated in two different ways. First, mean values for Uganda and Kenya were computed for the spatially distributed median, minimum, and maximum soil losses and compared to the mean soil losses in Fenta et al. (2020). Second, the quantile soil losses were grouped into soil loss levels based on a classification used in Fenta et al. (2020) and area proportions were calculated for each soil loss level. These area proportions were compared to the area proportions of the soil loss levels reported in Fenta et al. (2020).

For all administrative units and all USLE model setups the mean soil loss was calculated. The distribution of the mean soil loss in each administrative unit was analyzed with descriptive statistics. Employing Eq. (3) soil loss levels were determined for all grid cells in the respective administrative units and for all USLE model setups. The areas of each soil loss class calculated from all USLE model setups per administrative unit were summed up to compute the average share of a soil loss class for each administrative unit. Only administrative units located in the erosion prone regions that are indicated in Fig. 1 are analyzed in the main document and compared to the soil losses on the administrative level presented in Karamage et al. (2017). To provide a complete summary of the soil losses on the administrative level for all counties of Kenya and districts of Uganda we refer to the supplementary document section S.5 and the figures S.2 and S.3.

## 3.7 Comparison of the soil loss estimates to in field assessments

To provide a reference for the USLE ensemble simulations we used literature values of long-term mean annual soil loss from in-field assessments. García-Ruiz et al. (2015) compiled a comprehensive literature review for global soil loss rates, where three sources provided values for five sites within the study area of Kenya and Uganda. All three sources, however, applied different methods to assess the soil loss and cover a wide range of spatial domains. Sutherland and Bryan (1990) estimated the soil loss from the $0.3 \ \mathrm{km}^2$ Katiorin catchment located in the Lake Baringo drainage area in Kenya based on an in-stream discharge and suspended sediment sampling. Sutherland and Bryan (1990) estimated an average soil loss for the Katiorin catchment of 73 $\mathrm{tons} \ \mathrm{ha}^{-1} \ \mathrm{yr}^{-1}$ with a range between 16 and 96 $\mathrm{tons} \ \mathrm{ha}^{-1} \ \mathrm{yr}^{-1}$. Kithiia (1997) reported results from soil loss monitorings in tributaries of the Athi River Basin conducted by the Kenian Ministry of Water Development. From the tributary sampling sites in the Athi River Basin we selected the $41 \ \mathrm{km}^2$ Riara catchment with an average reported sediment load of 1474 $\mathrm{tons}$ $\mathrm{yr}^{-1}$ (0.36 $\mathrm{tons} \ \mathrm{ha}^{-1} \ \mathrm{yr}^{-1}$). Bamutaze (2010) preformed an erosion plot experiment in the Sinje catchment at Mt. Elgon in Uganda. Based on a two year monitoring, Bamutaze (2010) estimated a mean soil loss of 0.838 $\mathrm{tons} \ \mathrm{ha}^{-1} \ \mathrm{yr}^{-1}$ with a range between 0.185 and 1.761 $\mathrm{tons} \ \mathrm{ha}^{-1} \ \mathrm{yr}^{-1}$. De Meyer et al. (2011) assessed the soil loss from 36 farm compounds in the two

villages Iguluibi and Waibale close to the northern shore of Lake Victoria in Uganda. De Meyer et al. (2011) assessed the soil loss by reconstructing the historic surface level and calculating the lost soil volume. The estimations range between 56 and 460 $\text{tons ha}^{-1} \text{ yr}^{-1}$ in Iguluibi and 27 and 135 $\text{tons ha}^{-1} \text{ yr}^{-1}$ in Waibale.

To compare the ensemble soil loss estimations in this study with the literature values we calculated mean soil losses for grid cells that cover the original study site locations. Statistical measures were aggregated for the calculated site averages and plotted against the measured soil losses acquired from the selected studies.

## 3.8 Used software

The entire calculation of the USLE model realizations, most part of the input factor generation and the entire analysis of the simulation results was performed in the R programming environment (R Core Team, 2019). Spatial tasks and analyses were performed using the spatial R packages `raster` (Hijmans, 2019), `sf` (Pebesma, 2018), `rgdal` (Bivand et al., 2019), and `fasterize` (Ross, 2018). Data handling with SQLite data bases was managed through interfacing with the `RSQLite` (Müller et al., 2018) and `dbplyr` (Wickham and Ruiz, 2019) packages. Data analyses employed the R packages `dplyr` (Wickham et al., 2019b), `forcats` (Wickham, 2019), `lubridate` (Grolemund and Wickham, 2011), `purrr` (Henry and Wickham, 2019), `tibble` (Müller and Wickham, 2019), and `tidyr` (Wickham and Henry, 2019). Parallel computing to run some analyses was performed with the R packages `foreach` (Microsoft Corporation and Weston, 2017b), `doSNOW` (Microsoft Corporation and Weston, 2017a), and `parallel` (R Core Team, 2019). $LS$ factor realizations were generated with the LS Module in SAGA GIS (Conrad et al., 2015). Spatial maps were prepared in ArcGIS (ESRI, 2012) and in the R environment `ggplot2` (Wickham et al., 2019a) was used for all other figures.

## 4 Results

### 4.1 Analysis of the soil loss simulated with the USLE model ensemble

Overall, the calculated soil losses by our models follow the spatial pattern indicated by the potential erosion risk from topography that was presented in Fig. 1a). Both, the ensemble mean (Fig. 3a)) and the median soil loss (Fig. 3b)) show increased soil losses where moderate or high erosion risks were identified based on the slope thresholds suggested by Ebisemiju (1988). Although the soil loss levels shown in Fig. 3 differ from the soil loss levels that were used by Fenta et al. (2020), the spatial patterns of soil loss by water reported in Fenta et al. (2020) strongly agree with the patterns of the mean and median soil losses shown in Fig. 3a) and b). Mean soil losses of larger than 50 $\text{tons ha}^{-1} \text{ yr}^{-1}$ were found in the south-western corner of Uganda around Lake Bunyoni and along the Rift Valley in the North-West of Kenya. Excessive soil losses that exceed 200 $\text{tons ha}^{-1}$ $\text{yr}^{-1}$ were calculated for the steep slopes around the Ruwenzori Mountains, Mt. Elgon, and Mt. Kenya with ensemble mean soil losses of up to 1865, 1663, and 1438 $\text{tons ha}^{-1} \text{ yr}^{-1}$, respectively. Large variations in the calculated soil losses in each grid cell in combination with highly positively skewed distributions are two reasons why the calculated mean soil losses are generally larger than the median values.

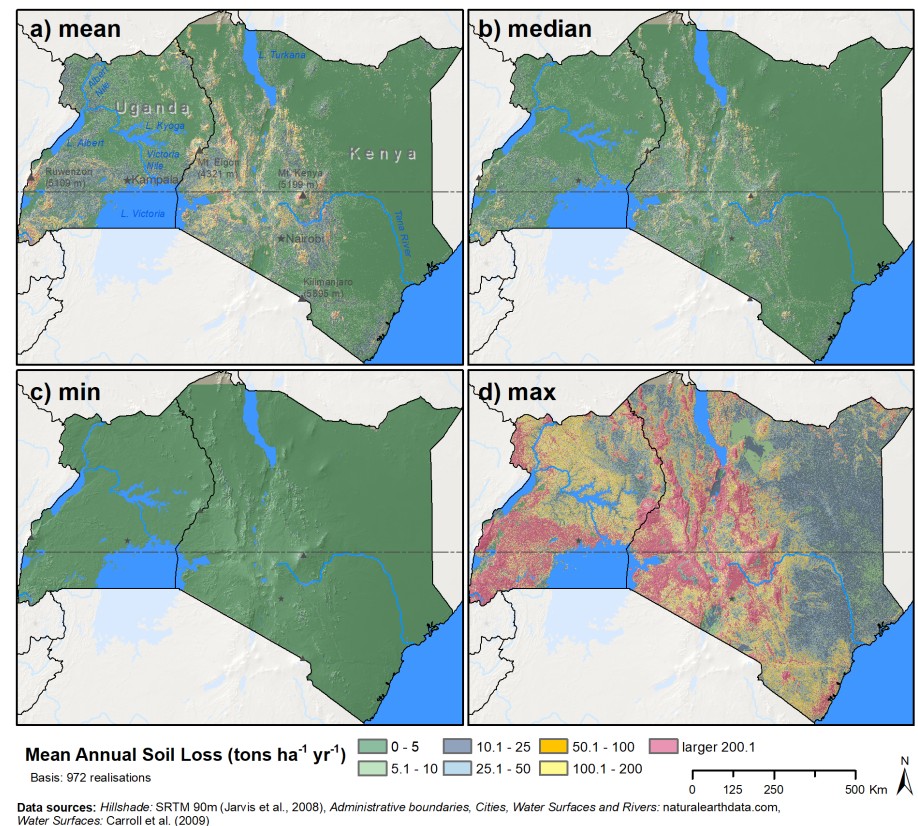

**Figure 3.** Descriptive statistics calculated for each grid cell based on the 972 USLE model realizations. Panels a) to d) show the mean, median, minimum, and maximum long-term annual soil erosion in each grid cell.

The strong discrepancy between the USLE model setups is evident from the comparison of the minimum calculated soil losses (Fig. 3c)) and the maximum soil losses (Fig. 3d)) in each grid cell. While combinations of USLE model input factors were present in the model ensemble that calculated soil losses below 10 $\mathrm{tons\ ha^{-1}\ yr^{-1}}$ for 99 % of the study region and soil losses below 100 $\mathrm{tons\ ha^{-1}\ yr^{-1}}$ for the entire study region, other input factor combinations resulted in soil losses above 200

5    $\mathrm{tons\ ha^{-1}\ yr^{-1}}$ for over 45 % of the study region and substantial soil losses of at least 50 $\mathrm{tons\ ha^{-1}\ yr^{-1}}$ for over 85 % of the study region.

Fig. 4 provides a different perspective of the same ensemble simulations. Each grid cell shows the frequency for the defined soil loss levels *slight*, *moderate*, *high*, and *severe* (panels a)-d) respectively) that were predicted by the model members of the USLE model ensemble. For large areas in the Northern Region of Uganda, the south of the lakes Kyoga and Albert in Uganda,

10   and the Northeast Province and the northern parts of the Eastern Province in Kenya over 90 % (and in many cases all) of the USLE model setups calculated slight soil losses. In the topographically heterogeneous regions of the Uganda Plateau, the South West of Uganda and the Gregory Rift in Kenya, a substantial share of up to 40 % of all model setups calculated a slight soil

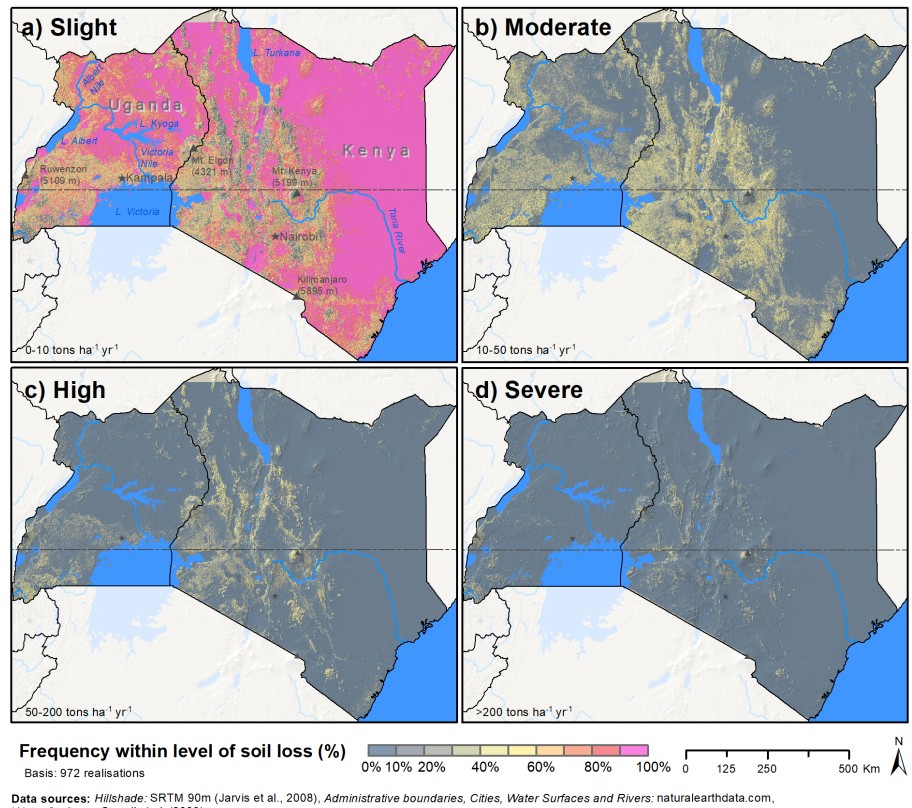

**Figure 4.** Frequency of USLE model ensemble members to predict one of the four soil loss classes *slight* ($0 - 10$ tons ha$^{-1}$ yr$^{-1}$) (a), *moderate* ($10 - 50$ tons ha$^{-1}$ yr$^{-1}$) (b), *high* ($50 - 200$ tons ha$^{-1}$ yr$^{-1}$) (c), and *severe* ($>200$ tons ha$^{-1}$ yr$^{-1}$) (d), based on the soil loss classification after FAO-PNUMA-UNESCO (1980). The pixel color illustrates the percentage of models from the model ensemble that calculated a soil loss in between the respective class boundaries.

and the majority of model setups resulted in moderate soil losses. Only along the steep mountain ridges in the Rift Valley and the mountain massifs of Mt. Kenya, Mt. Elgon, the Ruwenzori Mountains and the region around Lake Bunyoni a substantial part of USLE model setups calculated high and severe soil losses (yellow and local red regions in Fig 4 c) and d)).

    Fig. 5 combines the soil loss classification and the (un)certainties in the prediction of soil loss levels based on the USLE model ensemble into one representation. The dominant soil loss levels that a majority of model setups predicted for a grid cell are shown in green (*slight*), blue (*moderate*), orange (*high*), and purple (*severe*). The lightness of the colors indicates the percentage of models that calculated a soil loss within the respective soil loss classes. To highlight the complex patterns that result from the ensemble soil loss estimations in topographically heterogeneous regions, we show the Mt. Elgon (Fig. 5 b)), Lake Bunyoni (Fig. 5 c)), and Mt. Kenya (Fig. 5 d)) regions in detail.

    The strong agreement between the USLE model setups to calculate slight soil loss for the generally flat regions of Kenya and Uganda (shown in purple in Fig. 4 a)) is visible in dark green in Fig. 5 a). The soil loss level patterns in the erosion prone

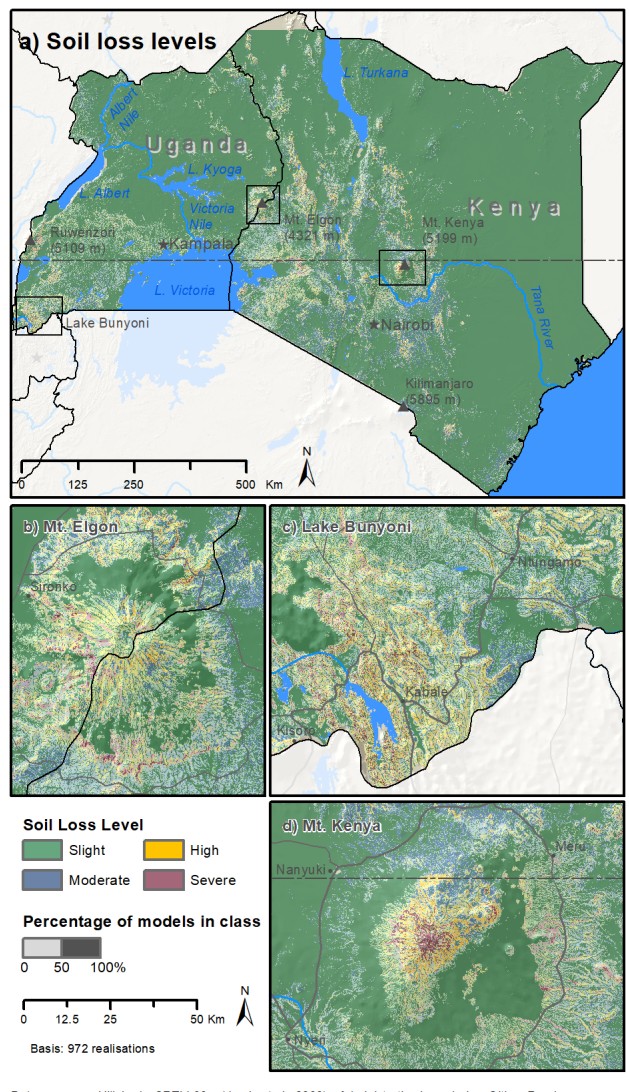

Data sources: *Hillshade:* SRTM 90m (Jarvis et al., 2008), *Administrative boundaries, Cities, Roads, Water Surfaces and Rivers:* naturalearthdata.com, *Water Surfaces:* Carroll et al. (2009)

**Figure 5.** Dominant soil loss levels. The color shows the soil loss level predicted by the majority of USLE model setups. The lightness of the color indicates the percentage of models that predicted the dominant soil loss level. Panel a) shows the study area of Kenya and Uganda. The panels b), c), and d) show erosion prone areas around Mt. Elgon, Lake Bunyoni, and Mt. Kenya, respectively.

areas of Mt. Elgon, Lake Bunyoni, and Mt. Kenya clearly follow the topographic patterns of these regions, with high and severe soil loss levels along the mountain ridges and slight to moderate soil losses in the valley bottoms. The agreement of the USLE model setups to predict the same soil loss level in such heterogeneous topographies is generally lower, showing percentages of 25 to 75 %. Only along the very steep slopes of the mountain massifs (and particularly at the top of Mt. Kenya with its steep
5    slopes and low vegetation cover) a large majority of the USLE model ensemble predicted a severe soil loss (center of Fig. 5

d)). Although the entire Mt. Elgon and the Mt. Kenya massifs show moderate to steep slopes (see. Fig. 1 b)), a large majority of the USLE model ensemble (>75 %) calculated slight soil losses for the densely forested northern part of Mt. Elgon and the forest belt around Mt. Kenya.

## 4.2 Analysis of the USLE input factors

To analyze and compare the individual realizations for the USLE inputs summary statistics were calculated for all grid cells of the study area. A detailed summary for all inputs is presented in the supplementary document section S.2. The median values of the $R$ factor realizations range between 1581 and 6851 MJ mm ha$^{-1}$ h$^{-1}$ yr$^{-1}$ where the method of Nakil (2014) shows the lowest value and the method of Roose (1975) the largest median value. All other methods show comparable median values with a range of 2243 – 3652 MJ mm ha$^{-1}$ h$^{-1}$ yr$^{-1}$. The maximum $R$ values show, however, a wide range between the

implemented methods, where $R_{Nakil}$ again shows the lowest value (6875 MJ mm ha$^{-1}$ h$^{-1}$ yr$^{-1}$) and $R_{TMPA}$ a 4.5 times larger value with 31068 MJ mm ha$^{-1}$ h$^{-1}$ yr$^{-1}$. The maximum values are however very local and the values of the third quantile of most of the $R$ values for the different methods are within a narrow range of 3606 – 5463 MJ mm ha$^{-1}$ h$^{-1}$ yr$^{-1}$. Summarized for the entire study area the implemented methods do not show any clear differences between the different types of methods that were implemented. The quantile $R$ values for $R_{GloREDa}$ (from high temporal resolution precipitation data)

for example, greatly compare to the quantiles of $R_{Fenta,MFI}$ that considers the rainfall seasonality, or the method of Moore (1979), which is based on long-term annual rainfall.

    For the $K$ factor realizations in contrast a clear difference can be observed between the implemented methods. While the $K$ factor realizations that employed the methods of Wischmeier and Smith (1987) (as implemented in Panagos et al. (2015c)) and Williams (1995) resulted in comparable values, with 0.005 – 0.038 tons h MJ$^{-1}$ mm$^{-1}$ and 0.011 – 0.039 tons h MJ$^{-1}$

mm$^{-1}$ respectively, when applied to the SoilGrids250m data set, the method of Torri results in a substantially larger range (0.00 – 0.109 tons h MJ$^{-1}$ mm$^{-1}$). Overall, all quantiles for the $K$ values that employ the method of Torri are approximately 4 times larger than the respective quantiles for the other two methods.

    Similar findings are visible for the realizations for the $LS$ factor. The median, the first and the third qunatiles for the method of Desmet and Govers (1996) resulted in substantially larger $LS$ values compared to the methods of Böhner and Selige (2006)

and Moore et al. (1991) with median values of 0.334, 0.074, and 0.013, respectively, when implemented with the SRTM v4.1 90m DEM. The methods of Böhner and Selige (2006) and Moore et al. (1991) resulted, however, in substantially larger maximum values (70.63 and 91.48) compared to the method of Desmet and Govers (1996) (19.31).

    Overall, the summary statistics for the $C$ factor values show clear differences between the methods that employed the MODIS NDVI, the ESA CCI LC, and the MODIS LC, whereas the impact of the implemented agricultural statistics, or the

temporal aggregation of the NDVI on the summary statistics of the $C$ factor is low. The median (0.214 and 0.175), the third quantile (0.402 and 0.355) and the maximum value (1) of the $C$ factor realizations that employed the NDVI are approximately twice as large as the respective quantiles for the methods that implemented the ESA CCI LC ($median = 0.080$, $q_{.75} = 0.15$ and 0.232, $maximum = 0.5$), and the MODIS LC ($median = 0.15$, $q_{.75} = 0.15$ and 0.232, $maximum = 0.5$) land cover products. The first quantiles of the $C$ factor realizations that employed the NDVI (0.059 and 0.472) show however 2 and 3

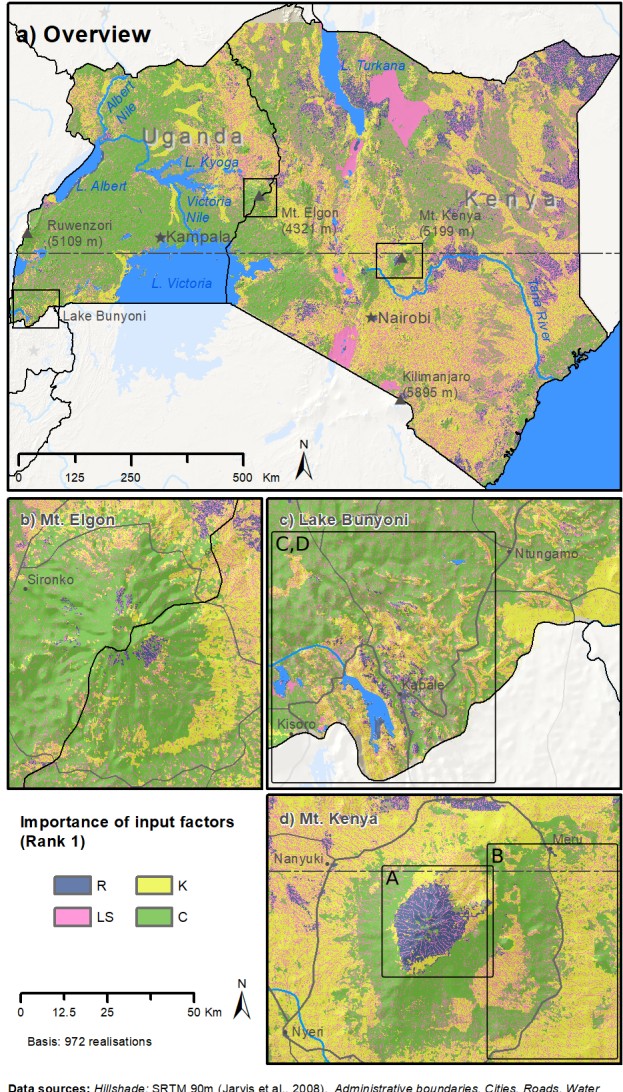

**Figure 6.** Most important USLE model input factors for the calculation of the soil loss $A$. The colours blue, yellow, pink, and green indicate whether the input factors $R$, $K$, $LS$, or $C$ caused the largest range in the calculation of $A$ in a grid cell. Panel a) shows the study area of Kenya and Uganda. The panels b), c), and d) show critical erosion hot spots around Mt. Elgon, Lake Bunyoni, and Mt. Kenya, respectively. The insets A) to D) indicate the extents for which the input factor realizations for $R$, $K$, $LS$, and $C$ were analyzed in Fig. 7.

times smaller values than the first quantiles for the realizations that implemented the ESA CCI LC (0.080) and the realizations that implemented MODIS LC (0.150), respectively.

The range of the calculated soil loss $A$ in a grid cell is the direct result of the different values stemming from the various input factor realizations. A large range in the values of an input factor in a grid cell has a greater impact on the resulting uncertainties

of the calculated soil loss compared to input factors where the different realizations show similar values. The analysis of the strongest impact of input factors on the uncertainties of $A$ revealed clear spatial patterns at different spatial scales (Fig. 6 a)). Over the whole domain, the input factors $C$, $K$, and $LS$ were identified as the most important inputs for the uncertainties in soil loss in 33.89%, 31.35%, and 28.45% of the total study area, respectively. The $R$ factor was only locally identified as the most relevant input factor in 6.31 % of the total study area. The $C$ factor and the $K$ factors show large aggregated patterns in both countries. The importance of the $LS$ factor, however, generally shows small structured, heterogeneous patterns scattered over the entire study region. Exceptions are visible in larger depressions along the Gregory Rift in zones where the slope is close to 0. Lake Magadi (100 $\text{km}^2$), an alkine lake located in an endorheic basin in the Rift Valley south of Nairobi, or a larger region in the east of Lake Turkana are the most distinct examples for large patterns of $LS$. Clusters of high importance of the $R$ factor were only identified in high altitudes with generally large precipitation sums, but also in very dry regions in the northern Kenya, where the precipitation sums are close to 0.

Fig. 6 b)-d) provides more detail of the spatial patterns of the input factors and their importance for the calculation of the soil loss in regions around Mt. Elgon, Lake Bunyoni, and Mt. Kenya (that were also analyzed in Fig 5). In contrast to Fig. 6 a), finer-scale characteristics of input factor importance become visible. The patterns around the two mountains Mt. Elgon and Mt. Kenya show similarities. Although the $R$ factor is spatially highly concentrated at the top of Mt. Kenya and only slightly visible on the east of Mt. Elgon, both regions show a high importance of the $R$ factor for the calculation of $A$ in high altitudes. High altitude areas are mostly characterised by a sparse observation network for precipitation. $R$ is highly correlated to some, in our case spatially distributed, rainfall estimates. High uncertainties in rainfall records, but also in the modelling chain to derive remotely sensed precipitation explain these patterns. Moving down from the summits, belts of a high importance of the $C$ and $K$ factor are visible. These distinct patterns result from the vertical bands of changes in vegetation in such mountainous regions and the impact of sparse and dense natural vegetation and agricultural land uses on the calculation of the $C$ factor. The Lake Bunyoni region shows more heterogeneous patterns for the most important input factors. In the north, the calculation of $A$ is affected by the $C$ factor in large regions and the $LS$ factor on very small scaled patterns. In the east and west of Lake Bunyoni, patterns for all input factors are visible that follow the terrain topography. The $LS$ and $K$ factor are the most relevant input factors for the calculation of $A$ along the ridge lines, while the $C$ factor becomes more important closer to the valley bottoms.

The importance of an input factor for the calculation of $A$ in Fig. 6 results from the differences in the estimated input factor values for the individual input factor realizations. In addition to the general analysis on the quantiles of the input factor realizations for the entire study region, we analyzed the input factor realizations of $R$, $K$, $LS$, and $K$ in the four regions A) to D) (indicated in Fig. 6) with greater detail in Fig. 7. For the analysis only grid cells in the defined extents A) to D) were selected and only (i) where the respective input factor was the most relevant one and (ii) where the calculated soil loss was classified to be high or severe.

Case A) (Fig. 7 A)) shows the differences of $R$ factor realizations at the top of Mt. Kenya. In this specific case (and other locations with high altitudes, data not shown), a difference between the rainfall erosivity products derived from temporally high resolution rainfall (GloREDa (Panagos et al., 2017) and TMPA (Vrieling et al., 2014)) and the distributions of the $R$ values

obtained from long-term annual precipitation is visible. While both, GloREDa and TMPA show low $R$ values between 1869 and 3486 MJ mm ha$^{-1}$ h$^{-1}$ yr$^{-1}$ and 3000 and 4602 MJ mm ha$^{-1}$ h$^{-1}$ yr$^{-1}$, respectively, the methods of Roose (1975), Moore (1979), Renard and Freimund (1994), Lo et al. (1985), and Fenta et al. (2017) (employing $P_{annual}$) resulted in a wide range of $R$ values between 4821 MJ mm ha$^{-1}$ h$^{-1}$ yr$^{-1}$ (minimum value using the method of Fenta et al. (2017)) and 16207

MJ mm ha$^{-1}$ h$^{-1}$ yr$^{-1}$ (maximum value using the method of Roose (1975)). Hence, a strong impact of the selected equation to calculate $R$ from long-term annual precipitation is observable. Only the methods of Nakil (2014) and the method of Fenta et al. (2017) (that employs the $MFI$) showed low $R$ values in a comparable range as GloREDa and TMPA, with ranges of 2590 – 3757 MJ mm ha$^{-1}$ h$^{-1}$ yr$^{-1}$ and 3828 – 5046 MJ mm ha$^{-1}$ h$^{-1}$ yr$^{-1}$, respectively. The method of Nakil (2014), however, resulted in very low $R$ values overall (also where GloREDa and TMPA showed significantly larger $R$ values), as

outlined in the analysis of the entire study area (see also section S.2 in the supplementary document).

Case B) (Fig. 7 B)) compares the $K$ factor realizations in the south-eastern belt around Mt. Kenya. The six realizations of $K$ show the same pattern as it is observable for the entire study area. The methods that were employed to calculate $K$ strongly affect the calculation of $K$, while the differences between the two soil products that were used are rather insignificant. In this specific case in Fig. 7 B), the method of Torri et al. (1997) resulted in by far the largest $K$ values between 0.069 tons h MJ$^{-1}$

mm$^{-1}$ and 0.088 tons h MJ$^{-1}$ mm$^{-1}$. On average these values are three times larger than the ones calculated with the method of Williams (1995) (with a range between 0.021 tons h MJ$^{-1}$ mm$^{-1}$ and 0.031 tons h MJ$^{-1}$ mm$^{-1}$) and up to 13 times larger than the values calculated with the method of Wischmeier and Smith (1987) when using the SoilGrids data set (with a range between 0.011 tons h MJ$^{-1}$ mm$^{-1}$ and 0.028 tons h MJ$^{-1}$ mm$^{-1}$).

Case C) (Fig. 7 C)) shows the differences between the the $LS$ factor realizations along the ridges of the hills around Lake

Bunyoni. Eventually, only the SRTM 90m DEM was used as input data and is shown in Fig. 7. Panel C) compares the three methods of Moore et al. (1991), Desmet and Govers (1996), and Böhner and Selige (2006). While the methods of Moore et al. (1991) and Böhner and Selige (2006) resulted in comparable values with ranges between 1.47 and 3.90 and between 1.65 and 5.03, respectively, the method of Desmet and Govers (1996) resulted in five times larger values with a range between 8.22 and 18.79. In this specific case the method of Desmet and Govers (1996) resulted in values close to the overall maximum value that was calculated for the study region (19.31). The methods of Moore et al. (1991) and Böhner and Selige (2006) resulted

in lower values although their maxima for the entire study region exceed the maximum value that results from the method of Desmet and Govers (1996) by a factor of 3 – 4.

Case D) (Fig. 7 D)) compares the implemented $C$ factor realizations for the same extent around Lake Bunyoni as for case C). In general two patterns are observable. A strong difference between the realizations that employ the NDVI as input and

the $C$ factor realization that were derived from land cover products and literature $C$ factor values is visible. Further, using the gridded crop distribution product of Monfreda et al. (2008) to derive spatially distributed mean $C$ factor values from the literature resulted in larger values compared to the implementation of agricultural census data on the administrative unit level for Kenya and Uganda. The impact of the used land cover product (ESA LC or MODIS LC) are low. Both realizations based on NDVI (NDVI, annual and NDVI, rainy season) show mean $C$ factor values of 0.04 and 0.03, respectively. The $C$ values for the

realizations that employed crop data from Monfreda et al. (2008) and agricultural census data were on average six times and

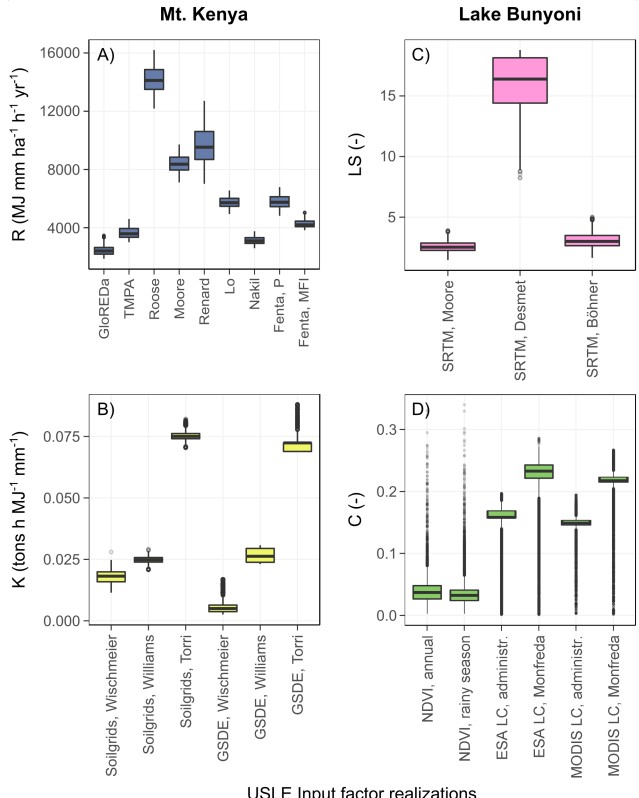

**Figure 7.** Variability between the realizations of the most important USLE model input factors. The cases A) to D) (delineated in Fig 6) exemplify the differences in the distributions of the input factor $R$, $K$, $LS$, and $C$, respectively. The cases A) to D) include the values of input factor realizations for grid cells, in which the respective input factor was the most sensitive one and the majority of models of the model ensemble predicted high to severe soil loss. Panel A) analyzes the $R$ factor realizations at the top of Mt. Kenya, panel B) shows the differences in the $K$ factor realizations in the belt around Mt. Kenya, and the panels C) and D) analyze the $LS$ and $C$ factors in the hilly topography of the Lake Bunyoni region.

4.5 times larger with mean values of 0.21 and 0.15 respectively. The results for this specific case contrast the general analysis of the $C$ factor values for the entire study region, where $C$ factor values of the realizations that implemented the NDVI are substantially larger compared to the methods that employed land cover products.

### 4.3 Soil loss assessment at administrative levels and comparison to other studies

On a national level the results reported in Fenta et al. (2020) allow a comparison to ensemble soil loss estimates of this study. Fenta et al. (2020) calculated mean soil losses of 7.3 and 6.7 $\mathrm{tons\ ha^{-1}\ yr^{-1}}$ for Uganda and Kenya, respectively. While the USLE ensemble median soil losses show comparable values of 7.7 and 7.3 $\mathrm{tons\ ha^{-1}\ yr^{-1}}$ on average for Uganda and Kenya, the minimum and maximum average soil losses for the two countries that result from the USLE model ensemble show extreme

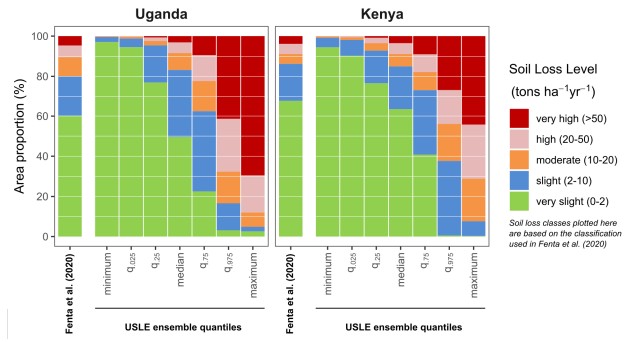

**Figure 8.** Comparison of the proportions of the areas in Kenya and Uganda that are summarized with different soil loss levels. The comparison shows the results reported in Fenta et al. (2020) to the results of the 972 USLE model realizations. The analyzed quantiles represent the soil loss quantiles in each grid cell that result from the USLE model ensemble. For the comparison the soil loss levels applied in Fenta et al. (2020) were used.

ranges (Uganda: $0.3 - 301.2$ tons ha$^{-1}$ yr$^{-1}$, Kenya: $0.5 - 207$ tons ha$^{-1}$ yr$^{-1}$). Fig. 8 compares the area proportions for Uganda and Kenya that were shown in Fenta et al. (2020) to the summarized results from the USLE model ensemble. For a comparison the ensemble soil loss quantiles in each grid cell were classified based on the soil loss levels that were used in Fenta et al. (2020) and their area proportions were summarized. Overall, the area proportions of the median soil losses agree with the findings of Fenta et al. (2020). It is, however, evident that the area proportions of the soil loss levels that were calculated for the lower and upper quantiles strongly differ from the proportions presented in Fenta et al. (2020). While the lowest two quantiles of the USLE ensemble calculated a *very slight* soil loss for over 90 % of both countries, the maximum soil losses calculated in each grid cell would result in *very high* soil loss for almost 70 % of the area in Uganda and over 40 % of the area in Kenya (compared to the 4 and 5 % shown in Fenta et al. (2020) and the 3 % shown by the ensemble median).

The selected administrative units in Uganda and Kenya are located in erosion prone areas (shown in Fig. 3 and Fig. 4). Although, averaging the soil loss for the domain of an administrative unit reduces the impact of areas with excessive soil loss, the median values of mean soil loss for the selected administrative units that result from the USLE model ensemble result in a moderate (blue) soil loss in 22 of the 27 administrative units. Four administrative units show even a high (yellow) mean soil loss, while only one administrative unit resulted in a slight (green) soil loss (Fig. 9 a)). Particularly large mean soil losses were found for the administrative units Kabale and Kisoro in the Lake Bunyoni region and the administrative units Kasese and Bududa on the slopes of the Ruwenzori Mountains and Mt. Elgon, respectively. The data points shown as coloured squares in Fig. 9 a) provide a reference to the soil loss assessment performed by Karamage et al. (2017) on district level in Uganda. As we included the realizations of the USLE input factors developed in Karamage et al. (2017) in the present assessment, the calculated soil loss from Karamage et al. (2017) is a member of the USLE model ensemble. In 9 of the 16 districts the soil losses calculated by Karamage et al. (2017) are lower than the 25 % quantile of soil losses that resulted from the USLE model

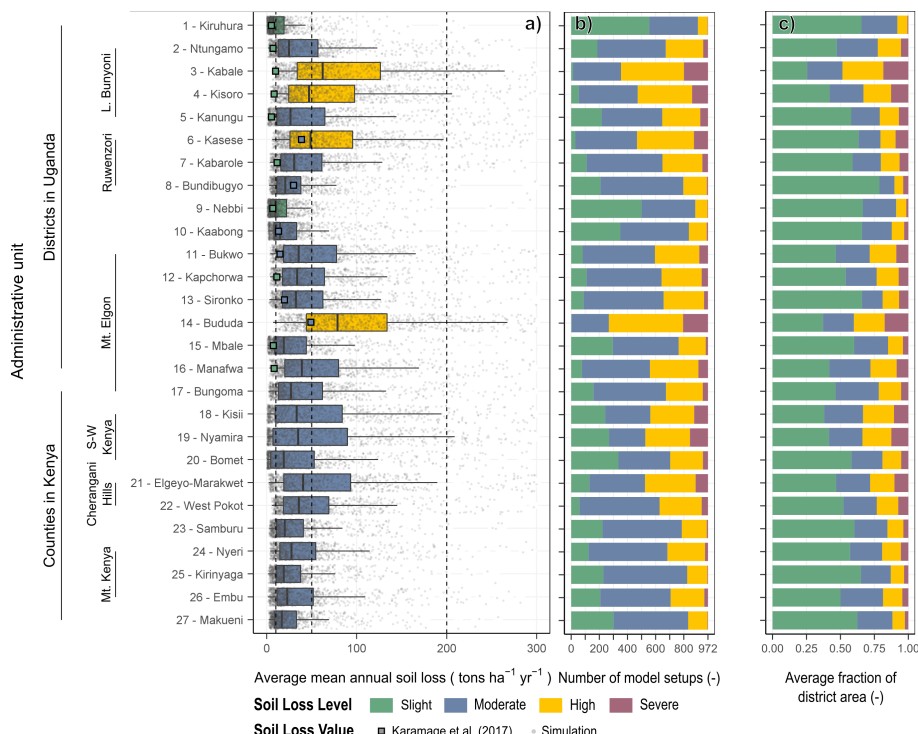

**Figure 9.** Mean soil loss in selected erosion prone administrative units of Uganda and Kenya. Panel a) shows the mean soil loss from all 972 USLE realizations in the selected administrative units with grey dots and aggregated as boxplots. The colors indicate whether the median soil loss in an administrative unit is *slight* (green), *moderate* (blue), *high* (yellow), or *severe* (purple). For comparison the results from Karamage et al. (2017) are plotted as colored squares. Panel b) shows the distributions of soil loss levels that were predicted by the USLE model realizations for the selected administrative units. Panel c) shows the average shares of soil loss classes for the domains of the selected administrative units.

ensemble. Only for a few districts, such as Kasese, Bundibugyo, Nebbi, or Kaabong the soil losses calculated by Karamage et al. (2017) and the ensemble means show comparable values.

For each administrative unit, the mean soil losses that resulted from the individual USLE model ensemble members show wide spreads (indicated by box plots and light grey dots in Fig. 9 a)). The spreads were particularly large in the administrative units with overall high soil losses. In all administrative units the mean soil loss that resulted from the individual USLE model setups are scattered over several soil loss classes (class boundaries indicated by dashed lines in Fig. 9 a)). Fig. 9 b) summarizes the numbers of model setups that predicted one of the four soil loss classes for each administrative unit. Although the median soil loss class for the majority of the administrative units is *moderate* on average 48 % (462 out of 972 models; with a range of 26.5 % to 61.2 % between the 27 administrative units) of the models from the USLE model ensemble predicted moderate soil loss, while all other model setups predicted one of the other four soil loss classes.

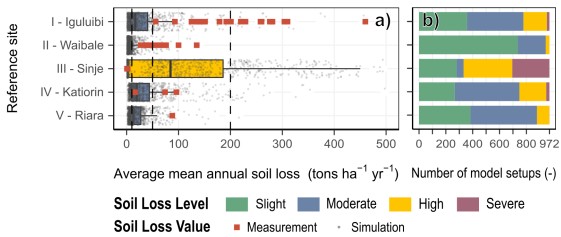

**Figure 10.** Comparison of soil loss simulations from the USLE model ensemble to in field soil loss assessments acquired from selected studies. The reference soil loss values are shown with red squares for the sites Iguluibi and Waibale (De Meyer et al., 2011), Sinje (Bamutaze, 2010), Katiorin (Sutherland and Bryan, 1990), and Riara (Kithiia, 1997) in panel a). The soil loss simulations for the reference extents from all 972 USLE model realizations are shown as grey circles. Corresponding boxplots show summary statistics for the model ensembles in panel a). Panel b) summarizes the numbers of models that predicted the soi loss levels *slight* (green), *moderate* (blue), *high* (orange), and *severe* (purple) for the reference sites.

Fig. 9 c) relates the soil loss classification in the selected administrative units to the average shares of the soil loss classes in the administrative unit areas. While on average only 20 % of the models from the USLE model ensemble predicted a slight soil loss almost 54 % of the areas of the administrative units show on average a slight soil loss. Areas with high and severe soil loss share only small areas in the administrative units with average fractions of 14.9 % and 7.1 %, respectively. Though, these areas have a strong impact on the mean soil loss in an administrative unit.

## 4.4 Comparison of the soil loss estimates to in field assessments

While the total ranges of the soil loss estimates calculated for the reference sites from the USLE model ensemble cover the reference soil losses from literature values in all five cases in Fig. 10 the interquartile ranges for the USLE model ensemble can strongly differ from the values that were estimated from in field experiments.

Cases I and II in Fig. 10 compare average soil losses for the domains of the villages Iguluibi and Waibale to soil loss assessments of small scale farm compounds. In both cases the soil losses assessed in the field exceed the interquartile ranges that result from the USLE model ensemble, with ranges of 56 to 460 $\mathrm{tons\ ha^{-1}\ yr^{-1}}$ and 6.5 to 40.4 $\mathrm{tons\ ha^{-1}\ yr^{-1}}$ in Iguluibi and 27 to 135 $\mathrm{tons\ ha^{-1}\ yr^{-1}}$ and 2.8 to 10.2 $\mathrm{tons\ ha^{-1}\ yr^{-1}}$ in Waibale.

For the Sinje test case (case III in Fig. 10) in the Manafwa district in Uganda Bamutaze (2010) resulted in very low soil losses between 0.185 and 1.761 $\mathrm{tons\ ha^{-1}\ yr^{-1}}$. Generally the districts along Mt. Elgon are known to be erosion prone. On average the USLE model ensemble predicted high soil loss for the location of the Sinje test catchment with a median soil loss 97.29 $\mathrm{tons\ ha^{-1}\ yr^{-1}}$ and an interquartile range between 3.7 and 228 $\mathrm{tons\ ha^{-1}\ yr^{-1}}$. Although the range of calculated soil losses is generally large, only 11 % of models from the USLE model ensemble predict soil losses that are in the range of the values reported by Bamutaze (2010).

The reported soil losses for the Katiorin catchment are comparable to the soil loss estimations for the catchments extent that resulted from the USLE model ensembles (case IV in Fig. 10). Sutherland and Bryan (1990) reports a range of soil loss

between 16 and 96 $\mathrm{tons\ ha^{-1}\ yr^{-1}}$ for the Katiorin catchment and 47 % of the USLE model setups predict a soil loss in the same range. Almost 44 %, however, result in soil losses lower than 16 $\mathrm{tons\ ha^{-1}\ yr^{-1}}$.

Kithiia (1997) reports a very low soil loss of 0.36 $\mathrm{tons\ ha^{-1}\ yr^{-1}}$ for the Riara Basin. All USLE model realizations predict larger soil losses for the domain of Riara, with a minimum value of 1.4 $\mathrm{tons\ ha^{-1}\ yr^{-1}}$ and an interquartile range of 6.3 to
27.4 $\mathrm{tons\ ha^{-1}\ yr^{-1}}$.

## 5   Discussion

With this study we illustrated how strongly the estimated soil loss magnitudes can vary, simply due to the choice of the methods and data that are implemented to calculate the USLE input factors. The statistical analysis of the generated USLE model ensemble (Fig. 3) showed that ranges of one or two magnitudes for the estimated soil loss were possible. These large ranges ultimately resulted from the differences in the individual realizations of the USLE input factors (some realizations were over a magnitude larger than others in Fig. 7 and the tables S.11 – S.14). These differences in the inputs propagate through the USLE equation by multiplication (Sonneveld and Nearing, 2003). The large uncertainties in the estimation of soil loss that result from such an ensemble approach, but also the effort that has to be put into such an analysis raise immanent question that will be discussed in the following: i) what are the benefits of such an ensemble soil loss assessment and what can we learn from a comparison to single model soil loss studies; ii) can we identify specific realizations of the input factors and USLE model combinations as implausible, exclude them from the model ensemble and eventually reduce the uncertainties in the ensemble model predictions; iii) what can we delineate from the importance of USLE inputs on the estimation of soil loss and how do these findings compare to other studies; iv) and are in-field data that are potentially available from monitoring studies a valid reference for the evaluation of large scale USLE soil loss assessments.

### 5.1   Ensemble soil loss modeling - How can we benefit from the collective

Although the calculated magnitudes and the ranges in soil loss that result from the model ensemble were extreme for some locations, the ensemble modelling approach can provide essential information on the overall simulation uncertainties that are simply not available from single model implementations. The analyses illustrated in Fig. 5 exemplify how we can utilize the information provided be the USLE model ensemble to qualitatively evaluate the erosion risk for a specific location. Such a visualization can greatly support decision making as it provides in addition to the soil loss level information whether the majority of the USLE model ensemble predicted that specific soil loss level, or whether the prediction is highly uncertain. In the specific example in Fig. 5 low soil loss levels were frequently classified by a large majority of the USLE ensemble, while in complex terrain and for more severe soil loss levels a stronger disagreement between the USLE ensemble members is visible. In such cases, however, the combination with summary statistics as illustrated in allow an evaluation of the erosion risk as well as the uncertainties in the prediction.

The comparison to the results presented in Fenta et al. (2020) and Karamage et al. (2017) greatly exemplifies the issues that may arise from a single USLE model soil loss assessment. While the results presented in Fenta et al. (2020) show a

good comparison to the ensemble median the results of Karamage et al. (2017) are substantially lower than the ensemble predictions. These circumstances can be explained to a large extent due to the selected methods that were implemented to calculate the USLE input factors in the two studies. Fenta et al. (2020) employed for example the method of Panagos et al. (2015c) to calculate the $C$ factor, which was found to be less sensitive to extremely low $C$ values in densely vegetated areas

compared to the method of Van der Knijff et al. (2000) (see Fig. 7 D)). The method of Fenta et al. (2017) that was used to calculated the $R$ factor in Fenta et al. (2020) resulted in an $R$ factor realization that was in the medium range in this study. As a consequence, the overall soil loss estimations also compared well to the ensemble median. Karamage et al. (2017) in contrast, employed the methods of Lo et al. (1985) to compute $R$ and the method of Van der Knijff et al. (2000) to caculate $C$. Both methods were found to be on the lower ends of the spectrum when compared to the other methods in this study (particularly for

the $C$ factor in the densely vegetated regions of Uganda). In addition, Karamage et al. (2017) implemented a global $P$ factor value that further reduced the soil loss estimates. As a consequence, the calculated soil loss estimates were low in general. While, the ensemble approach allows to compare each model combination to all other combinations and therefore provides a reference point to the implementation of a specific USLE input combination, a single model approach simply cannot provide such information.

**5.2   USLE input realizations - Ranges, plausibility, and their comparison to other studies**

The analysis and comparison of the USLE input realizations revealed several systematic patterns in their summary statistics calculated for the entire study area, but also in the four specific cases that were presented in Fig. 7. Some of the patterns in the differences between specific realizations that were observed in the specific cases agreed with the patterns for the entire study domain, while others showed contradicting results. The systematic differences in the $K$ factor realizations for instance were

found in the specific case in Fig. 7 B), while the cases A) and D) for instance showed opposite behaviors of the realizations of $R$ and $C$ for the smaller regions. Overall, the sets of realizations for each input resulted in wide ranges of values that eventually resulted in large ranges of the calculated soil loss. Thus, it is worth to put the input factor realizations into a reference to other studies. In any case, we have to keep however in mind that a comparison to other studies does not per se determine specific realizations to be more or less plausible, as other large scale soil erosion studies face the same issues in terms of a model

validation (see 5.4).

Locally the calculated $R$ factor realizations showed values of large maximum values, where the largest $R$ values were found for the realizations $R_{TMPA}$, $R_{Renard}$, and $R_{Roose}$ with maxima of 31068, 25755, and 22741 MJ mm ha$^{-1}$ h$^{-1}$ yr$^{-1}$, respectively. The third quantiles of all methods range, however, between 2046 and 9636 MJ mm ha$^{-1}$ h$^{-1}$ yr$^{-1}$. Other large scale studies in East Africa and on a global scale report also wide ranges in the $R$ values. In an assessment for East Africa

Moore (1979) calculated rainfall erosivities of up to 10900 MJ mm ha$^{-1}$ h$^{-1}$ yr$^{-1}$ for the Mt. Elgon region. Fenta et al. (2017) found high values for $R$ of $> 7000$ MJ mm ha$^{-1}$ h$^{-1}$ yr$^{-1}$ for the northwestern Ethiopian highlands, the area around Mt. Kilimanjaro, and the western region around Lake Victoria in Uganda. Fenta et al. (2017) found these results to be in line with the findings in Vrieling et al. (2010). Karamage et al. (2017) calculated a range of $1674 – 6358$ MJ mm ha$^{-1}$ h$^{-1}$ yr$^{-1}$ for Uganda. For Europe Panagos et al. (2015a) found a range for $R$ of $51.4 – 6228.7$ MJ mm ha$^{-1}$ h$^{-1}$ yr$^{-1}$. In a global

soil loss assessment Naipal et al. (2015) calculated values for $R$ that exceeded magnitudes of $1 \cdot 10^5$ MJ mm ha$^{-1}$ h$^{-1}$ yr$^{-1}$. Although Naipal et al. (2015) emphasize that such large values are unrealistic they stress that erosivities of over 20000 MJ mm ha$^{-1}$ h$^{-1}$ yr$^{-1}$ can be observed in the tropics, which is also reported in Panagos et al. (2017). The excessive $R$ values that are shown locally by a few of the implemented realizations of $R$ can be questioned. Overall however, the ranges of the individual

$R$ realizations are in line with the results reported in other studies.

In the specific case presented in Fig. 7 B) the $K$ values that were calculated with the method of Torri et al. (1997) showed maximum values of 0.088 tons h MJ$^{-1}$ mm$^{-1}$. For the entire study region values larger than 0.1 tons h MJ$^{-1}$ mm$^{-1}$ were found. Depending on the input data set (Soilgrids250m or GSDE) the methods of Wischmeier and Smith (1987) and Williams (1995) resulted in maximum values of 0.038 and 0.039 tons h MJ$^{-1}$ mm$^{-1}$, and 0.055 and 0.052 tons h MJ$^{-1}$ mm$^{-1}$,

respectively. Ranges of $K$ factor values that are shown in other studies show comparable values to the ranges that resulted from the methods of Wischmeier and Smith (1987) and Williams (1995). The implementation of the method of Torri et al. (1997) exceeds the ranges shown in other studies. Karamage et al. (2017) calculated a range for $K$ of $0.015 - 0.029$ tons h MJ$^{-1}$ mm$^{-1}$ for Uganda. A similar range is shown in Fenta et al. (2020) for East Africa with high erodibilities shown for the North-West of Lake Victoria and the Rift Valley and the area around Lake Turkana in Kenya. On a global scale, Borrelli et al.

(2017) implemented $K$ values that range from values lower than $< 0.01$ tons h MJ$^{-1}$ mm$^{-1}$ to values $> 0.04$ tons h MJ$^{-1}$ mm$^{-1}$. For Europe Panagos et al. (2014) found values for the soil erodibility of up to 0.076 tons h MJ$^{-1}$ mm$^{-1}$ for medium to fine textured soils. Naipal et al. (2015) implemented values for $K$ of 0.08 tons h MJ$^{-1}$ mm$^{-1}$ for highly erodible volcanic soils. As a consequence, the implementation of the method of Torri et al. (1997) as it was implemented in this study must be questioned.

The majority of erosion studies implemented the method of Desmet and Govers (1996) to calculate $LS$ (e.g. Fenta et al., 2020; Karamage et al., 2017; Borrelli et al., 2017; Panagos et al., 2015e; Yang et al., 2003). As a consequence, the ranges for $LS$ that were found in these studies are in line with the ranges for $LS$ that we found with the implementation of the method of Desmet and Govers (1996). Although the methods of Moore et al. (1991) and Böhner and Selige (2006) showed excessive maximum values, these were highly local. As shown in the specific case in Fig. 7 C) large variations in the calculated soil loss

were mostly found in locations where the method of Desmet and Govers (1996) resulted in large values for $LS$ while the other two methods resulted in low values.

Overall, the $C$ factor values reported in other studies are comparable to the ranges of the $C$ factor that were calculated in this study, since the majority of studies which we reviewed implemented either the MODIS NDVI in combination with the method of Van der Knijff et al. (2000) to calculate $C$ or employed the method of Panagos et al. (2015c) in their study regions. Thus

studies that implemented the NDVI (e.g. Karamage et al., 2017) resulted in ranges for $C$ factor of $0 - 1$. Karamage et al. (2017) for example found values of $C < 0.05$ for large areas in the western and central parts of Uganda, whereas only regions in the North East show values $> 0.2$. Fenta et al. (2020), who implemented the method of Panagos et al. (2015c) calculated $C$ values that range between 0.135 and 0.33 in the South West of Uganda and north of Lake Victoria, whereas the forested regions in central Uganda show values below 0.01. Both findings are reflected in 7 D), that documents the discrepancies between the two

methods of Van der Knijff et al. (2000) and Panagos et al. (2015c). While the method of Panagos et al. (2015c) accounts for the

agricultural areas in the South West of Uganda in the calculation of $C$, the method of Van der Knijff et al. (2000) only accounts for the vegetation density (by implementing the NDVI as a proxy).

## 5.3 Input factor importance - Findings and comparison to other studies

Fig. 6 illustrated the most dominant USLE input factor realizations with respect to their impact on the uncertainties of the
calculated soil loss. The dominant input factors revealed spatial patterns on different spatial scales. The patterns of the most
dominant inputs follow the patterns of the input data that were employed to calculate the input factor realizations. Thus, the
shown patterns can support in identifying the input data/method combination that introduced the largest share of uncertainties
in the calculation of soil loss locally. Larger patterns were mainly visible for the input factors $C$ and $K$, while $LS$ showed
very small scaled patterns and $R$ showed a lower relevance for the prediction uncertainties in general. While the $C$ is the most
important input factor for large regions in the densely vegetated part of Uganda and around Lake Victoria in Kenya, $K$ is most
relevant in the drier regions of Kenya. The $R$ factor was mainly relevant in higher altitudes. The $LS$ factor realizations were
most relevant in highly variable topographies and very flat areas where the factor is close to zero and numerical issues governed
the results of the sensitivity analysis.

Based on nine nation wide soil loss data sets, including soil loss estimates for Europe (Panagos et al., 2015e), and the original
USLE data set for the USA Estrada-Carmona et al. (2017) performed global sensitivity analysis to identify the dominant USLE
input factors. In 8 out of 9 country wide analyses of the USLE input importance Estrada-Carmona et al. (2017) identified the $C$
factor to be tho most relevant one for the soil loss estimation. The second most relevant input shown in Estrada-Carmona et al.
(2017) was, however, the $LS$ factor, that was identified to be relevant very locally in this study. In a study in the mountainous
Tongbai-Dabie region in China Zhang et al. (2013) also found that the $LS$ factor was the most important input factor on small
scales. Keyzer and Sonneveld (1997) performed a meta-model study and analyzed the USLE model relationship based on the
original US data set that was employed in the development of the USLE. Based on the data points that were available from the
US data set, Keyzer and Sonneveld (1997) concluded that larger uncertainties in the soil loss estimation can be expected for
high $R$ and $LS$ values, as well as for high and low values for the $K$ factor as the number of samples were low for these regions
in the USLE inputs in the original USLE data set. Falk et al. (2010) employed Bayesian melding to quantify the uncertainties
in the soil loss estimates and to identify the USLE inputs that contribute the most to the uncertainties for a catchment in
Eastern Australia. In an analysis of the spatial distribution of the input uncertainties, and the magnitudes and uncertainties in
the calculated soil losses Falk et al. (2010) found a relationship between the patterns of the $S$ factor and the patterns that were
observed in the calculated soil loss.

All studies that were reviewed here differ in their methodological approaches and also come to different conclusions with
respect to the importance of the USLE inputs. Overall, the analysis of the most important inputs can greatly support a soil loss
assessment in order to identify the dominant sources of uncertainties in the soil loss estimates. Yet, the importance of the the
individual inputs seems to be very specific for the the individual studies.

## 5.4 Model validation - Are in-field data a valid reference for USLE model evaluation

Although large scale meta-analysis studies exist, that provide soil loss data globally (García-Ruiz et al., 2015), or for specific regions in the world (e.g. for Africa (Vanmaercke et al., 2014), or for Europe (Maetens et al., 2012)), these studies often compile reported soil losses that result from a wide range of study settings. The presented comparison of the USLE ensemble soil losses to in-field erosion studies should therefore not be seen as best practice, but rather provides illustrative examples of potential issues that can arise in the comparison to in-field data.

Overall, we were not able to delineate a clear pattern from the comparison of estimated soil losses to in-field soil loss assessments within the study domain, as the selected reference studies had different specific scopes. While Sutherland and Bryan (1990), or Kithiia (1997) monitored the accumulated soil loss from river catchments, De Meyer et al. (2011) assessed the soil loss on small scales and on sites that are particularly erosion prone. While most of the selected reference studies report low to moderate soil losses for their study domains, De Meyer et al. (2011) reports high to excessive soil losses for several of the farm compounds they investigated. The methodologies that were used for the soil loss assessments strongly impacted the reported soil losses and result in wide ranges of soil loss between the selected studies.

Aforementioned limitations of the temporal and spatial representativeness of the reported soil losses from the selected reference studies are likely to be present and may have impacted the significance of the comparison to the soil loss estimates. At larger scales, processes other than the ones that are assessed by the USLE, such as deposition processes, gully erosion, or bank collapses have to be considered in the quantification of the soil loss (Govers, 2011). Boardman (2006) stresses that long-term monitoring schemes and additional assessments of rills and gullies would be required to allow a comparison to soil loss estimations. Records from erosion monitoring studies are, however, usually short (Evans, 2013; Govers, 2011). The reference studies of Sutherland and Bryan (1990) and Bamutaze (2010) for instance only covered monitoring periods of 1 and 2 years, respectively and thus are only snapshots in time that are difficult to compare with long-term assessments.

Apart from the short monitoring periods that are often available from reference studies it is likely that the (remote sensing) data that was employed to calculate the USLE input factors and to assess the soil loss do not reflect the conditions that were present during the monitoring period in a study region, simply because the monitoring period and the period for which input data are available do not overlap. Soil cover by vegetation perfectly illustrates the issue. Monitoring data can date back several decades (e.g. Sutherland and Bryan (1990) in our case). On large scales the vegetation cover is often estimated by employing remote sensing satellite data that can be more recent than monitoring data. Particularly, in East Africa deforestation affected the land cover over the past decades with reported decreases in the forest biomass of up to 26 % in Uganda (Jagger and Kittner, 2017), or forest clearances in protected forests in the Mt. Elgon region of 33% (Petursson et al., 2013). In such a case, a $C$ factor that was calculated with recent remote sensing data would fail to reflect the condition of the vegetation during the monitoring period.

Although the soil losses reported in De Meyer et al. (2011) are based on cumulative soil losses in farm compounds over periods of 15 to 20 years, the spatial domains of the farm compounds that were analyzed do not properly reflect the spatial resolution of the grid on which the soil loss assessment with the USLE was conducted. Other reference studies, such as

Sutherland and Bryan (1990) or Kithiia (1997) represent the average soil loss at the catchment scale. One could assume that the spatial scale of such studies better agrees with the spatial scale of a large scale soil loss assessment with the USLE. These reported loads are affected by processes, such as deposition, gully erosion, land sliding, or bank erosion that superimpose rill and inter-rill erosion (Govers, 2011). Boardman (2006) further highlights that the in-stream sediment delivery ratios (SDR) are

a function of time and scale. Boardman (2006) compares the differences in the SDR of the Yellow River and British rivers that differ by a factor of 28. Such large difference in the SDR does, however, not necessarily reflect the differences in soil erosion rates.

Evans (1995) and Boardman (2006) point out that soil losses derived in plot scale experiments do not reflect erosion taking place on the landscape scale. Evans (1995) found that plot scale soil losses are larger than soil losses in the landscape by a

factor of two to ten under comparable conditions. The soil losses reported in Bamutaze (2010) were however lower than the soil losses estimated by almost 90 % of all used USLE models in this study and thus show an opposite behavior.

Prasuhn et al. (2013), Warren et al. (2005), or Evans (2002), among others, demand that soil losses that were estimated by models must be supported by field based observations. Bosco et al. (2015) emphasize the limitations of in-field validation for large scale studies. Bosco et al. (2014) and Bosco et al. (2015) highlight the potential to employ high resolution satellite

imagery and Google Earth, or Google Streetview data for plausibility checks of soil loss estimates. Yet, the verification (and falsification) of the absolute magnitudes of soil loss estimates on large scales remains a challenge.

### 5.5   Further considerations and limitations

In this study we only implemented a selection of methods and primary data sources for the calculation of the USLE input factors. Hence, we have to recognize that the performed study does not provide a comprehensive picture of the uncertainties that

are introduced by different representations of the USLE input factors. Albeit, the calculated ranges in soil loss were substantial and considering additional realizations of USLE input factors can in the worst case increase the ranges of calculated soil loss. The demonstrated procedure, however, pinpoints the central weakness of the USLE. The model can identify relative risks for soil erosion, but fails to predict exact magnitudes of soil loss. Eventually every modeller must acknowledge the limitations of the USLE (some of them we addressed at great length) and not overestimate the predictive power of the model.

We are fully aware that such a comprehensive analysis is very likely out of scope for most studies that employ the USLE model, as in most applications the soil loss estimation is only a small part of the entire analysis. Further, extending such analysis to larger domains or increasing the spatial resolution can be limited by available computation and storage capacities. For instance, the entire ensemble of USLE model representations in the present study comprised $11225 \times 14778 \times 1944$ ($\sim 322 \cdot 10^9$) pixel values required 2.74 TB distributed in SQlite data bases on four separate hard drives to allow an efficient batch-wise

analysis of the model results.

We omitted the analysis of the conservation support or management practice factor $P$ in this study. For all USLE model setups the $P$ factor was globally set to a value of 1. According to literature values, the application and maintenance of support practise measures can substantially reduce the soil erosion in erosion prone landscapes. Conservation measures, such as contour farming, strip cropping, or terracing reduces the calculated soil loss by a factor of up to 2, 4, and 10, respectively, depending

on the slope on which the measure was applied (Karamage et al., 2017; Shin, 1999). Large scale estimations of $P$ and the implementation of the $P$ factor in large scale soil loss assessments are almost absent, as only very limited spatial data is available on soil conservation measures. Panagos et al. (2015d) generated a spatial estimate for $P$ for entire Europe, considering the effects of contouring, stone walls, and grass margins. Panagos et al. (2015d) thereby used comprehensive spatial statistics

on soil conservation based on 270000 data points available for Europe from the LUCAS data base (LUCAS, 2012). Such detailed data is, however, not available in all regions of the world. Thus, other large scale assessments omitted the $P$ factor and used a value of 1 globally (e.g., Borrelli et al., 2017), assigned a reduced $P$ value globally in the study domain (Karamage et al., 2017), assigned global values for $P$ to specific land uses (Yang et al., 2003), or used land cover and slope as a proxy for the $P$ factor estimation (Fenta et al., 2020). Such simplifications do not reflect the spatial distributions of soil conservation

measures that are actually applied in a (large scale) study domain, although their impact on large soil loss estimates can be substantial.

## 6    Conclusions

The USLE model, an empirical model to estimate the soil loss by water erosion is widely applied in large scale assessments and was implemented in a case study to assess the soil loss on the entire domain of Kenya and Uganda. Although the USLE has

a simple model structure and is therefore easy to implement, the generation of spatially distributed estimates of the USLE input factors for the study domain poses a major challenge. Large scale (remote sensing) data products and methods to employ them for the generation of the USLE inputs greatly support soil loss assessments on large scales. We generated sets of realizations for each USLE input factor and combined them to 972 USLE model setups to compute spatially distributed soil loss estimates for Kenya and Uganda. Based on the generated USLE model combinations we analyzed and quantified the impacts of frequently

used methods to calculate USLE inputs on the uncertainties in the soil loss estimation with the USLE model.

Overall, but particularly in erosion prone areas of the study domain, the calculated ranges of soil loss showed large values. In many cases, especially in areas with high soil losses, the calculated ranges exceeded the mean soil loss by greater than one order of magnitude. To condense the information provided by the USLE model ensemble we proposed to classify the soil loss into the soil levels *slight*, *moderate*, *high*, and *severe* employing common soil loss thresholds from literature. The classification

allowed to utilize the USLE ensemble predictions to analyze but consider the "certainty" of the prediction simultaneously. The employed approach enabled to identify zones with a high soil loss, but also areas where the agreement in the USLE model ensemble is low and thus suggest an evaluation and/or plausibility checks for the simulations.

A sensitivity analysis of the soil loss predictions was performed to identify the USLE input factors that introduce the strongest impact on the uncertainties of the soil loss estimates. The analysis identified clear patterns on the large scale for the input factors

$C$ and $K$, where the $C$ factor is more relevant for areas with denser vegetation and the $K$ factor showed a greater importance for the calculation of the soil loss in dry less densely vegetated areas. The $LS$ factor showed very scattered patterns in complex topographies and was relevant for the uncertainties of the calculated soil loss in sloped terrain.

The comparison of the USLE ensemble soil loss estimates to single USLE model implementations illustrate the advantages of an ensemble over single model studies. While the ensemble members provide a reference to other USLE input combinations, with a single model no reference is given to evaluate the calculated magnitudes in soil loss.

A validation of simulated soil loss on large scale domains, employing in-field assessments from the literature poses to be a challenge and in this study no clear conclusions can be drawn for the ensemble soil loss estimates when they were compared to soil loss observations. Thus, the comparison failed to falsify any of the generated USLE model combinations that would allow to exclude ensemble members to ultimately reduce the soil loss prediction uncertainties. Major issues for a valid comparison are the differing origins of the in-field soil loss data as well as spatial and temporal limitations of the observed data.

Although available computational and time resources will naturally limit such an analysis of soil loss predictions in most studies that employ the USLE model, the findings clearly highlight the importance to critically view and analyze single USLE model predictions, as the resulting soil loss estimates are highly sensitive to the combinations of realizations of the USLE model inputs. We further question the aptitude of soil loss assessments based on in-stream sediment yields or small scale plot experiments to be valid data for the evaluation of soil loss estimates. We should think of new approaches to validate soil loss estimates that employ large scale data that is now available. Bosco et al. (2014) outline a method to employ satellite imagery to check the plausibility of large scale soil loss assessments.

*Code and data availability.* The study was performed using openly available primary input data. For some of these data we do not have the permission for further distribution. All input data can, however, be acquired from the rights holders of these data sets. All intermediate and final data that were generated in this study and the corresponding R code to manage and process the data are available upon request to the corresponding authors.

*Author contributions.* CS and MH designed the study and acquired and processed the input data. CS performed all analyses. MH and CS prepared the figures. KS and JK contributed in the methodological framework. CS, MH, BM, JK, and KS compiled the manuscript.

*Competing interests.* The authors declare that they have no conflict of interest.

*Acknowledgements.* This study was conducted within the frame of the Appear Project 158: *Capacity building on the water-energy-food security Nexus through research and training in Kenya and Uganda (CapNex)* funded by the Austrian Partnership Programme in Higher Education and Research for Development (Appear). APPEAR is a programme of the Austrian Development Cooperation.

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
