# Peer review of "A systematic assessment of uncertainties in large scale soil loss estimation from different representations of USLE input factors - A case study for Kenya and Uganda"

_Hydrology and Earth System Sciences, 2019_

## Referee Comment (RC1) · Anonymous Referee #1 · 28 Dec 2019

An interesting article proposing interesting aspects in USLE modelling: a) uncertainties b) comparison of factors c) validation.

However, there are issues that authors should face in order to improve the quality and proposing it for publication. An important issue is that authors did not propose 'solutions'. They did the 756 USLE simulations but they should also propose which is the most representative one per factor. For example , which is the best method for the R-factor?

[Figure]

Authors propose new approaches to check the plausibility of large scale assessments based on Bosco et al., 2014. This study has neither been peer reviewed nor published. So, I would suggest using published literature studies for such statements.

In a similar study, Estrada-Carmona et al (2017) made a global sensitivity analysis of USLE input factors. Please compare the results of your study with the ones of this study.

The most recent study that I found in East Africa is the one of Fenta et al., 2019 Science of the Total Env. How your results compare with their results?

Soil losses estimates wit USLE are long-term averages. You cannot compare the long-term findings against short-term findings in plot experiments.

Authors made a classification of soil erosion rates. The tolerable soil erosion rate cannot be justified according to literature findings as the soil formation rates are low. This means that sustainable soil erosion rates are lower than 1-2 t per ha per year. In addition to this, authors present some really extreme mean annual soil loss rates > 200 -1,500 t ha-1 yr-1. This means that at least 2 cm of soil is lost every year. This maybe the case for very limited areas; otherwise we risk to lose completely our soils in 50 years. This means that some of the estimated combinations are not realistic. You should not be driven by the modelling outputs but somehow use also the common logic (you cannot lose 1m of soil in 50 years).

Title: I would replace the word 'representations' with 'applications'

P4 L15-24: This paragraph is not needed.

Fig. 3: attention in the measurement unit of soil erosion. It is better to use t ha-1 a-1. If you want to keep your proposed unit , then please put in parenthesis (ha a)

Fig 7. It should be applications and not realizations.

P23 end of the page and P24 beginning of the page: I would propose that some applications of the factors can be excluded. For example, the NDVI application is known to have very low C-factor results and it is known to have incorporated some problems. The same applies for R-factor. For example the methods of Lo and Fournier are based on rainfall amount and do not incorporate the rainfall intensity.

P19 the same as above. Are values of K-factor 0.088 acceptable? Can be compared to other findings in the literature?

P20 L1-10: The same as above. Can you prove that values of C-factor 0.03 are acceptable in agricultural areas?

5.2 section. It is not proper to have a section with question.

P25 L30-34: Is It possible to validate large scale models with Google Earth? Google Earth can potentially verify permanent erosion characteristics (e.g. gully erosion) and not rill and sheet erosion. The plausibility of large scale studies can be verified with model applications at regional or local

---

## Referee Comment (RC2) · Anonymous Referee #2 · 18 Jan 2020

General comments

The paper presents a large scale assessment of the uncertainties in USLE soil loss estimation as a consequence of different realizations and combinations of the corresponding input factors. A total of 756 USLE model setups were examined with a spatial detail of 90 meters (cell size). Moreover, the case study (Kenya and Uganda) is vast enough to include a great variability of topographical, climatic and land use conditions. For these reasons, the ranges of both input factors and soil loss are very wide, contributing to improve the scientific reliability and interest of the work. The spatial

variability of the model sensitivity to the different factors was examined and discussed. An attempt to compare/validate the simulated soil loss with field soil loss data was also made. All the sections of the paper are very clear and the scientific background is well detailed and discussed. The degree of agreement between the estimates obtained by the different input ensembles was evaluated not only on the basis of the quantitative values, but also and above all on the basis of the soil loss category (tolerable, moderate, high and severe). This is in fact the most rational approach for a model characterized by high uncertainty.

Specific comments

Lines 3-6 pag. 3. I suggest to mention other recent promising modifications of the USLE, such as those proposed and tested by Bagarello et al. (2010) and Di Stefano et al. (2019): - Bagarello, V., Ferro, V., Giordano, G. 2010. Testing alternative erosivity indices to predict event soil loss from bare plots in Southern Italy, Hydrological Processes 24(6), 789-797. - Di Stefano, C., Pampalone, V., Todisco, F., Vergni, L., Ferro, V. 2019. Testing the Universal Soil Loss Equation-MB equation in plots in Central and South Italy, Hydrological Processes 33(18), 2422-2433

Figure 1. I suggest to check the legend of the figure 1a, in which the erosion risk is represented according to a discrete classification based on only three colours (white, yellow and pink). However, from the figure, the colour grey is also widely present and gradients for both yellow and pink are evident. I think that a discrete classification/legend is not correct.

Figure 1. I understand that the purpose of Figure 1 is just to provide a rough description of the erosion-prone areas according to topography, vegetation cover and rainfall amounts. In relation to this last aspect, however, the authors could have chosen a proxy more appropriate than the annual precipitation: in fact it is well known that the distribution of rains has a determining role in soil loss. In particular, several studies in the literature have shown that in some areas, the annual soil loss is highly correlated

with the erosivity of a few erosive events. Therefore, other synthetic indices (e.g. the Modified Fournier Index (Arnoldus, 1980) could be proxy more reliable than annual precipitation in the description of the susceptibility to erosion due to rainfall characteristics).

Lines 5-13 pag. 8. As stated by the authors themselves (section 5.3), it is not possible to consider all the available methods for the calculation of USLE input factors and the authors made plausible choices in their selections. However, the authors started their analysis of the R factor by aggregating the long-term monthly amounts to the annual scale, thus losing the possibility of applying the methods that derive the R factor from both annual and monthly precipitations. The reasons for this choice should be provided.

Section 5.2. the discussion presented in this section was expected since the authors described in section 3.7 their intent to compare simulated yields with those collected from field observations. I agree that there are several limitations and difficulties, but the attempt is appreciable. I wonder if another possible reason for the lack of agreement could be represented by the differences between the land use at the time of field experiments and the average one considered in the simulations, (e.g. Sutherland and Bryan (1990) refers to experiments carried out before 1990, whilst the MODIS NDVI data are from 2000 to 2012).

Fig. 8a. In order to improve the clarity of the boxplots in figure 8a, I suggest to eliminate the dots, whose presence is not much effective since the data spread can be derived from the length of the whiskers of the boxplots. A similar consideration holds for fig. 9 and S1 and S2 in the supplement material.

Technical corrections

Pag 1 line 8: "challanges" should be "challenges" Pag.19 line 32 check the sentence Pag. 26 line 9 replace ULSE with USLE

602, 2019.

---

## Referee Comment (RC3) · Anonymous Referee #3 · 31 Mar 2020

General comments:

This paper presents an analysis of the variability in soil loss estimates with the USLE equation due to different representations of its factors, and subsequent comparison of the predictions with field data. It is certainly not the first time that the uncertainty of erosion predictions with the USLE is questioned. Yet, the fact that the USLE is very often applied using very different data and methods to determine its input factors still make the study relevant. The study uses a representative selection of frequently used methods to determine the USLE factors based on readily available land use,

climate, soil and topography data. The paper is generally well written, but could be more concise at some points and there are some issues that require better explanation or justification, as explained below.

The authors rightly argue that there is a huge range in erosion rates predicted in function of the methods used to obtain the model input factors. What is interesting however is that the ensemble prediction shows relatively good agreement regarding the predicted erosion severity class. So, although agreement with measured erosion data is poor, in line with earlier studies, you might argue that such ensemble prediction is useful for qualitative description of erosion severity. This can be helpful to prioritize policies.

However, the comparison of predicted soil loss with measured erosion and sediment yield data is most problematic. As the authors also mention at some point, the USLE does not consider sediment deposition and transport so it cannot be compared with sediment yield from gauging stations. On the other hand, the erosion rates provided by De Meyer et al. (2011) based on reconstructing the historic surface level and calculating the lost soil volume from 36 farm compounds are extremely high. I am not sure which method was applied exactly by De Meyer et al and for what time and spatial scale the assessments are made for example. In any case, such high values can occur a certain points, but are probably not realistic for larger areas. So, the question is how useful are these comparisons actually, and do we need them to assess the uncertainty in USLE predictions due to variations in its factors? Model validation is very important, but only useful if the modelled and measured data refer to the same processes and the same scales of assessment.

I find the classification of the predicted soil loss values in four classes, below and above tolerable soil loss rate of 10 t/ha/yr, problematic and it does not add much to the entire discussion of the uncertainty of model predictions. First of all, the tolerable soil loss rate depends on a spatially variable soil production rate, which is unknown for the area. Secondly, the USLE soil loss predictions are gross erosion rates and do not account

for deposition during transport over distances longer than a standardized erosion plot. This makes it highly arguable to look at the USLE predictions in relation to tolerable soil loss rates. You can classify the predictions in erosion severity classes but I recommend to delete reference to tolerable soil loss rates.

What exactly is the aim of the comparison of soil loss estimates at the administrative level? How does this contribute to the research objectives explained in the introduction? While it can be an interesting exercise, and may provide relevant information for local policy makers, it seems the whole section 4.3 does not really contribute to the main objectives of your study.

Please explain and illustrate with quantitative data why you did not include the ASTER DEM for calculation of the LS factor. Previous studies have also highlighted that at higher resolutions problems can occur with LS calculations, but since you first projected the ASTER DEM on the 90 SRTM grid could be expected to be less problematic. It would be interesting to see what is exactly the cause of this problem and compare this to other studies that assess the differences in ASTER and SRTM DEMs and their application in erosion studies.

The methods used to assess the C factor rest strongly on the approach used by Panagos et al (2015) and Borrelli et al (2017), but I find the description quite difficult to follow. It is not clear why and how exactly you overlay the already spatially distributed 'crop shares statistics' with the land cover maps? Moreover, it seems the approach puts a lot of detail in differentiating between different crops, but disregards the possible importance of intra-annual differences in C factors due to crop rotations.

There are several other papers that also discussed the impacts of USLE factors and structure on outcomes (e.g. Sonneveld and Nearing, 2003) that would be interesting to include in your discussion.

Detailed comments (indicated per Page and Line):

P2-L17-18: can you add a line how the revised version was different?

P3-L8-11: you may add here a few words on the often used Sediment Delivery Ratio in combination with gross erosion to obtain sediment yield predictions, correcting for the fact that the USLE does not predict sediment deposition.

P3-L14: remove 'the'

P3-L20-23: please check and preferably simplify this sentence.

P3-L33-35: It is indeed not simple to do this kind of comparisons and most plot data do not cover 20 years, but there are by now relatively good and large datasets of measured soil loss available, such as for example the data presented by Garcia Ruiz et al (2015) and Maetens et al. (2012) for Europe. For many other parts of the world this is still more difficult though.

P4-L10: Research objectives are now formulated as research questions; better write them as objectives. In the last objective correct 'we we'.

P4-L16-24: These lines do not seem necessary, and seem repetitive.

P5-L4: on the steepest slopes (>20) gully erosion can be expected to be an issue as well.

P7-L6: what about seasonality of rainfall?

P8-L8 and supplementary Table S1: It seems you only used relationships based on mean annual precipitation to estimate the R factor (not accounting for seasonality). It would have been interesting to include an equation based on the monthly data, for example those based on the Modified Fournier Index proposed by Renard & Freimund (1994) that you cite. The text above Table S1 states 'The first four methods' which should be the 'first five'.

Supplementary page 8 (above table S6) correct 'To compute the K factor realizations..' for 'To compute the LS factor realizations'.

P9L30: please correct sentence 'served as base layers for the join with..'

Table S7: what does the first column 'value' mean?

P12-L16: this may be interesting, but where exactly do we find the results of this? I couldn't find it in the results section.

P12-L19: it is not clear from this paragraph how the comparison of soil loss rates at the administrative level contributes to the papers objectives expressed in the introduction.

P14-L29: With 'the dominant soil loss levels that a majority of model setups predicted' you refer to the soil loss level for which most agreement was between the model set-ups? What if there was no majority for any of the soil loss levels? Unfortunately, in the figure 5, the lightness of the colours that should indicate the percentage of models that calculated a soil loss within the respective soil loss classes, cannot be distinguished.

Figure 7: in the heading it states that the values refer to those pixels for which 'high to severe soil loss was predicted to be likely'. How is 'to be likely' defined here? Or does this refer to high or severe soil loss as predicted per model implementation?

P20-L20: why do you highlight and compare with the data from Karamage et a (2017)? Did you introduce this in methods? I don't see the added value, especially considering that the data are already covered within your model implementations, so it seems there is nothing new.

P23-L18: But you did not really perform a plausibility check of the individual USLE model realisations, so the argument does not make too much sense.

P24-L8: the comparison with one particular study does not contribute anything to this interpretation; the wide variety between your results indicates that you cannot take conclusions based on only 1 model implementation and that an ensemble approach makes more sense.

P24-L15-18: This sentence misses a conclusive statement. Indeed, the tolerable soil

loss is controversial and does not seem to add much to your assessment.

P24-L26-27: please correct this sentence.

P24-L28-20. The detail in the patterns is not a property of the factor, or how important the factor is, but it just reflects the level of spatial variability that is present in the input data used. This does not mean anything for the relevance of one factor as compared to another or a scale influence. The interesting part of your result is the overall impact of each factor on total ensemble variability.

P25-L11: at larger spatial scales you will need to include not only different sources of sediment (rill, gullies, mass movements), but also deposition during transport, as explained in detail by numerous previous studies.

P25-L20: If the data are in stream sediment loads they are certainly do not 'better meet the spatial scale of USLE'.

P25-L26-28: Various studies have dealt in detail with the role of spatial scale in erosion assessments, and how plot scale data compare to sediment yield (e.g. de Vente et al, 2007). Further, the difference between the plot data and USLE model predictions do not have anything to do with comparing plot data with landscape scale sediment yield. Plot data and the USLE assessments in theory both consider the same erosion and deposition processes at the same scale.

P25-L31-33: I think quantitative validation via google earth will be difficult and you do not really explain how this could be done.

P26-L9: ULSE = USLE

P26-L12-14: computer capacity for these kind of calculations should nowadays for most studies not be a problem anymore.

P26-L14: Ideally yes, like in any model you need to validate the predictions. But, how do you determine the plausibility if you don't have field data to compare with?

Based on your assessment and comparison with field data would you say that the USLE assessments are plausible? You need data that can be compared with the USLE predictions, so representative for the same scale. I think the main interest is in the fact that the ensemble prediction shows relatively good agreement in the severity class of erosion, but quantitative validations are problematic.

P27-L4: please rephrase and simplify the sentence.

P27-L11: increased soil loss = high soil loss

P27-L26-28: Most important here is to make sure that the data are comparable, so representing the same erosion and sediment transport or deposition processes. In theory, USLE predictions should compare with plot data.

P27-L28: this recommendation is very vague. What kind of new approaches? How would google maps provide quantitative estimates that can be compared with model predictions?

---

## Author Comment (AC1) · 28 Apr 2020

**Reply to the reviewer comments RC1: 'Hess – USLE model Uganda' by Anonymous**

**Referee #1**

We would like to thank the Anonymous Referee #1 for the very constructive review and the valuable comments to improve the quality of the manuscript. We appreciate the positive feedback on the manuscript. In the following, we addressed each comment individually. The reviewer comments are printed in *serif, italic font*. Our replies to the individual comments are written below each comment in black non serif font. The literature that was cited in the reply is added at the end of the document.

*An interesting article proposing interesting aspects in USLE modelling: a) uncertainties b) comparison of factors c) validation. However, there are issues that authors should face in order to improve the quality and proposing it for publication.*

*An important issue is that authors did not propose 'solutions'. They did the 756 USLE simulations but they should also propose which is the most representative one per factor. For example , which is the best method for the R-factor?*

All realizations that were developed for each USLE input are based on methodologies that were proposed and implemented in peer reviewed studies. In the summary of methods for the calculation of (R)USLE inputs that was compiled by Benavidez et al. (2018) most of the methods that were implemented in the present manuscript were listed as 'valid' methods to compute the respective USLE input. The basic rule for the input generation in this manuscript, was that if a method was implemented in a peer reviewed study in Eastern Africa (or regions with comparable climatic/topographic/vegetation conditions)  before, it is considered as a plausible method for the generation of that input. From this perspective , all input realizations must be treated as equally adequate representations of that input.

The aim of this study was to assess the uncertainties which are inherent in the calculation of the long-term mean annual soil loss simply due to the choice of the methods for the calculation of the USLE inputs. It has never been the intention of this study to identify a 'best' realization for a USLE input. Any attempt to identify plausible or implausible USLE model combinations would have failed, as no measured (or other) data was available that could support a decision to verify or falsify a model combination. This was addressed in detail in the Sections 4.4 and 5.2 that highlighted the limitations of a comparison to the limited observation data that is available for the investigated study region. We suggest to revise these sections and try to strengthen the stated arguments to clarify the present limitations of model falsification.

*Authors propose new approaches to check the plausibility of large scale assessments based on Bosco et al., 2014. This study has neither been peer reviewed nor published. So, I would suggest using published literature studies for such statements.*

Bosco et al. (2015) briefly outline the methodology that is explained in much greater detail in Bosco et al. (2014). While Bosco et al. (2015) is a peer reviewed article, we agree that Bosco et al. (2014) is published as a pre-print version that was not peer-reviewed. In the present manuscript we always refer to both articles when we mention the applicability of

remote sensing data for a plausibility check of the USLE simulations. Although Bosco et al. (2014) is not peer reviewed it provides valuable information that can be accessed via a DOI and therefore the same document is available to the reader of this study to which we refered to when compiling this work.

*In a similar study, Estrada-Carmona et al (2017) made a global sensitivity analysis of USLE input factors. Please compare the results of your study with the ones of this study.*

Thank you for drawing our attention to this study. Although the approach presented in Estrada-Carmona et al (2017) differs from the approach that was presented in the present manuscript we should and will refer to this study.

*The most recent study that I found in East Africa is the one of Fenta et al., 2019 Science of the Total Env. How your results compare with their results?*

The study of Fenta et al. (2020) was not published by the time this manuscript was compiled. The workflow that is presented in Fenta et al. (2020) was adopted from Borrelli et al. (2017), Panagos et al. (2015), and Panagos et al. (2014) for the computation of the USLE K and C factors. To calculate the LS factor the method of Desmet and Govers (1996) was implemented. These three realizations for the inputs K, LS, and C are also members of the input sets in this study and should therefore result in comparable ranges. Interesting aspects in Fenta et al. (2020) are the computation of R and more importantly the consideration of soil protection measures represented by the P factor. Thus, a comparison of the calculations presented in Fenta et al. (2020) with the calculated ranges of soil loss in this study is valuable. Thus, we suggest to refer to Fenta et al. (2020) in the revised version of the manuscript.

*Soil losses estimates with USLE are long-term averages. You cannot compare the long-term findings against short-term findings in plot experiments.*

We agree that a comparison of long-term soil loss to short term measurements of e.g. sediment yields is improper. In the manuscript we specifically indicate the limitations of the comparability of USLE simulations to an instream monitoring of sediment yield. Yet, it is a common practice to employ short term records of observed soil loss (or sediment yield) to validate the results of a USLE model. See e.g. Fenta et al. (2020) where a comparison of USLE results to 'measured' short term sediment yields was performed.

Thus, we see a relevance to critically address the issue of a USLE model comparison to observation data. Eventually, the analysis of the USLE model ensemble and its comparison to the soil loss data collected by García-Ruiz et al. (2015) clearly supports a critical view on such model validation and is therefore in our opinion a relevant contribution to the soil erosion literature.

*Authors made a classification of soil erosion rates. The tolerable soil erosion rate cannot be justified according to literature findings as the soil formation rates are low. This means that sustainable soil erosion rates are lower than 1-2 t per ha per year. In addition to this, authors present some really extreme mean annual soil loss rates >200 -1,500 t ha-1 yr-1. This means that at least 2 cm of soil is lost every year. This may be the case for very limited areas; otherwise we risk to lose completely our soils in50 years. This means that some of the estimated combinations are not realistic. You should not be driven by the modelling outputs but somehow use also the common logic(you cannot lose 1m of soil in 50 years).*

We have refrained from making judgments about tolerable soil erosion rates, but we do point out that such values exist in the literature. The concept of this manuscript was to employ the information that we acquired from the peer reviewed literature to represent the current status of knowledge on the topic of "soil erosion by water" and the assessment of water erosion with USLE type models. This idea also is applied to the selection of soil loss classes that employ what is common practice in the literature. At no point in the manuscript do we imply that a soil loss of 10 t ha$^{-1}$ yr$^{-1}$ is indeed the value where the soil loss is compensated by the soil formation. We stated that suggested literature values range from 5 to 12 t ha$^{-1}$ yr$^{-1}$ and that several studies in Eastern Africa used 10 t ha$^{-1}$ yr$^{-1}$ as a threshold value. Due to the absence of more reliable values, we based the classification on literature values that included threshold values of 10 t ha$^{-1}$ yr$^{-1}$. Yet, we agree that threshold values of 1 or 2  t ha$^{-1}$ yr$^{-1}$ would be valid assumptions as well. Fenta et al. (2020), for example, classify soil loss by water as 'very slight' soil loss when the soil loss is in a range between 0-2 t ha$^{-1}$ yr$^{-1}$ and as 'slight' when the soil loss is in a range between 2-10 t ha$^{-1}$ yr$^{-1}$. Yet, Fenta at al. (2020) do not provide any reference on which their classification is based. In conclusion the classification of soil loss always involves a highly subjective view on the calculated soil losses. A classification should primarily reduce information and support the reader in the interpretation of data and we are aware that poorly chosen class names can mislead the reader. Nevertheless, we decided to use a classification that was implemented in the literature before and should be considered to be as valid as any other classification.

We agree with the reviewer that calculated soil losses of over 200 t ha$^{-1}$ yr$^{-1}$are extreme. Nevertheless, for the informative value of this manuscript it is relevant to keep these model combinations as potential USLE realizations. Two arguments for the value of these model members are as follows:

Indeed such high values were calculated only locally and only by a few model combinations. This is exactly what this manuscript tries to address. Models can be wrong. Thus, if other representations of the USLE estimate soil losses that are substantially lower then the modeler has a chance to evaluate such large soil loss estimates based on other estimates. Yet, with a single model approach (that is common practice in the literature) it is simply infeasible to evaluate large calculated soil losses if no reference by observation data or a model ensemble is given. Thus, if a model setup calculated excessive soil losses locally, what does this mean for the evaluation of the remaining areas in a study region?

USLE type models do not account for soil deposition and therefore do not reflect the sediment balance. Thus, the soil that is strongly eroded locally is also deposited and not completely lost but displaced.

*Title: I would replace the word 'representations' with 'applications'*

We think that 'applications' is not the appropriate term for the entities that we analyzed in this work. 'Applications' could also imply an application of USLE inputs in for instance different independent studies, or locations and would thus be misleading. In this manuscript we represented the USLE input factors R, K, LS, and C for the same study by employing different methods to compute them and would therefore prefer to use the term 'representation' or 'realization' when we refer to one member in the set of realizations for a USLE input.

*P4 L15-24: This paragraph is not needed.*

We agree that this paragraph does not provide any new information, but outlines the structure of the manuscript. We think that this is a subjective question of style and preference and believe that it helps the reader to get an overview of the content of the paper at hand.

*Fig. 3: attention in the measurement unit of soil erosion. It is better to use t ha-1 a-1. If you want to keep your proposed unit , then please put in parenthesis (ha a)*

Thank you for pointing this out! We will revise the units in all figures to be consistent with the units in the text and tables.

*Fig 7. It should be applications and not realizations.*

See the response above. The same argument applies as above for the title of the manuscript.

*P23 end of the page and P24 beginning of the page: I would propose that some applications of the factors can be excluded. For example, the NDVI application is known to have very low C-factor results and it is known to have incorporated some problems. The same applies for R-factor. For example the methods of Lo and Fournier are based on rainfall amount and do not incorporate the rainfall intensity.*

As responded previously, the goal of this work was to provide a comprehensive assessment of frequently implemented methods to calculate the USLE input factors and to evaluate the uncertainties in the soil loss estimates that arise from the input uncertainties due to the impact of different methods, published in peer-reviewed journals, to calculate the model inputs. Thus, it was not our intention to judge the implemented methods, but to consider them as potential methods if they have been implemented in similar study settings. This also applies to the methods used to calculate the R factor and the C factor.

Both methods addressed by the Anonymous referee #1 (the implementation of the NDVI and the method of Lo et al. (1985)) were recently implemented in a large scale soil loss assessment by Karamage et al. (2017). In general, the implementation of methods that use long-term precipitation instead of rainfall intensity is common due to the absence of rainfall intensity records. Thus, in terms of a comprehensive uncertainty assessment we must consider these types of methods for the calculation of C and R as well.

Concerning the limitations of any implementation of the NDVI to calculate C, we were not able to find information in the literature that NDVI is known for a calculation of low C-factor values. Yet, the analysis that we illustrated in Fig. 7 of the manuscript would support that statement for the selected region in Uganda. Also, any well documented issues with the application of the NDVI to calculate the C-factor is not known to us (or is at least not reported in the literature at hand). However, we specifically address in the manuscript that the method of Van der Knijff et al. (2000) was never validated against ground truth data.

Concerning the C-factor value ranges calculated with the method of Van der Knijff (2000) we want to take the analysis in Fig.7 of the manuscript as an example. Although the mean values and the quantile values for the C-factor with the NDVI are significantly lower to the other methods, the absolute ranges of C-factor values when employing the NDVI are in a plausible range. For Europe Panagos et al. (2015) calculated C-factor values as low as 0.00116 for forests and C-factor values of up to 0.2651 for sparse vegetation. The ranges presented in this manuscript are in line with the range presented in Panagos et al. (2015).

We see however that a comparison to other large scale studies would provide valuable information and will consider that in the revised version of the manuscript.

*P19 the same as above. Are values of K-factor 0.088 acceptable? Can be compared to other findings in the literature?*

Panagos et al. (2014) for instance calculated a range for the K-factor of 0.004 – 0.076 t h $MJ^{-1}$ $mm^{-1}$. Naipal et al. (2015) implemented the method of Torri et al. (1997) that resulted in such large K-factor values in this manuscript. Naipal et al. (2015) also applied a K-factor value of 0.08 to vulcanic soils as these are particularly erodible. Although single grid cells in the analysis in this manuscript exceed the mentioned literature values, the calculated values are not completely out of range.
Yet, we fully agree that it is relevant to set the calculated values into a reference with other literature. We suggest to add a comparison to the above mentioned litarture values and additional literature in the revised version of the manuscript.

*P20 L1-10: The same as above. Can you prove that values of C-factor 0.03 are acceptable in agricultural areas?*

Thank you for raising this point. The same arguments as in our replies to the previous two points apply here.

*5.2 section. It is not proper to have a section with question.*

We cannot verify this point, as the manuscript guidelines do not disallow questions as section headers., in essence, freedom of the authors. See Blöschl and Montanari (2010), Savenije (2009), or Schürz et al. (2019) as examples with questions as section headers.

*P25 L30-34: Is It possible to validate large scale models with Google Earth? GoogleEarth can potentially verify permanent erosion characteristics (e.g. gully erosion) and not rill and sheet erosion. The plausibility of large scale studies can be verified with model applications at regional or local*

We cannot confirm the feasibility of the implementation of satellite imagery for the validation of soil loss assessments. It was not our intention to illustrate such an approach as a verified method for the validation of large scale soil loss assessments. From a technical perspective it should however be possible to employ multi-angle high resolution imagery for different time steps to apply any stereographic analysis. Furthermore, intense soil erosion might be visible in satellite imagery and can be related to an erosion class, which would allow a use as 'soft' information for model validation. Regardless of the exact approach to employ satellite imagery, the conclusion in the presented manuscript was that our traditional approaches of model validation failed in this large scale study (and would likely fail in others) and new concepts for model validation would be valuable to be implemented in large scale assessments. The validation of a large scale model application with a model application at a smaller scale does not sound plausible unless the small scale model is validated, as any soil erosion model involves uncertainties.

References

Benavidez, R., Jackson, B., Maxwell, D., & Norton, K. (2018). A review of the (Revised) Universal Soil Loss Equation ((R) USLE): with a view to increasing its global applicability and improving soil loss estimates. *Hydrology and Earth System Sciences*, *22*(11), 6059-6086.

Blöschl, G., & Montanari, A. (2010). Climate change impacts—throwing the dice?. *Hydrological Processes: An International Journal*, *24*(3), 374-381.

Borrelli, P., Robinson, D. A., Fleischer, L. R., Lugato, E., Ballabio, C., Alewell, C., ... & Bagarello, V. (2017). An assessment of the global impact of 21st century land use change on soil erosion. *Nature communications*, *8*(1), 1-13.

Bosco, C., de Rigo, D., Dewitte, O., Poesen, J., & Panagos, P. (2015). Modelling soil erosion at European scale: towards harmonization and reproducibility. *Natural Hazards and Earth System Sciences*, *15*(2), 225-245.

Bosco, C., de Rigo, D., Dewitte, O. (2014). Visual Validation of the e-RUSLE Model Applied at the Pan-European Scale. *Scientific Topics Focus 1*, MRI-11a13. Notes Transdiscipl. Model. Env., Maieutike Research Initiative. https://doi.org/10.6084/m9.figshare.844627

Desmet, P. J. J., & Govers, G. (1996). A GIS procedure for automatically calculating the USLE LS factor on topographically complex landscape units. *Journal of soil and water conservation*, *51*(5), 427-433.

Estrada-Carmona, N., Harper, E. B., DeClerck, F., & Fremier, A. K. (2017). Quantifying model uncertainty to improve watershed-level ecosystem service quantification: a global sensitivity analysis of the RUSLE. *International Journal of Biodiversity Science, Ecosystem Services & Management*, *13*(1), 40-50.

Fenta, A. A., Tsunekawa, A., Haregeweyn, N., Poesen, J., Tsubo, M., Borrelli, P., ... & Kawai, T. (2020). Land susceptibility to water and wind erosion risks in the East Africa region. *Science of The Total Environment*, *703*, 135016.

García-Ruiz, J. M., Beguería, S., Nadal-Romero, E., González-Hidalgo, J. C., Lana-Renault, N., & Sanjuán, Y. (2015). A meta-analysis of soil erosion rates across the world. *Geomorphology*, *239*, 160-173.

Karamage, F., Zhang, C., Liu, T., Maganda, A., & Isabwe, A. (2017). Soil erosion risk assessment in Uganda. *Forests*, *8*(2), 52.

Lo, A., El-Swaify, S. A., Dangler, E. W., & Shinshiro, L. (1985). Effectiveness of EI30 as an erosivity index in Hawaii, in: Soil Erosion and Conservation, edited by El-Swaify, S. A., Moldenhauer, W. C., and Lo, A., pp. 384–392, Soil Conservation Society of America, Ankeny, IA, USA.

Naipal, V., Reick, C. H., Pongratz, J., & Van Oost, K. (2015). Improving the global applicability of the RUSLE model-adjustment of the topographical and rainfall erosivity factors. *Geoscientific Model Development*, *8*, 2893-2913.

Panagos, P., Meusburger, K., Ballabio, C., Borrelli, P., & Alewell, C. (2014). Soil erodibility in Europe: A high-resolution dataset based on LUCAS. *Science of the total environment*, *479*, 189-200.

Panagos, P., Borrelli, P., Meusburger, K., Alewell, C., Lugato, E., & Montanarella, L. (2015). Estimating the soil erosion cover-management factor at the European scale. *Land use policy*, *48*, 38-50.

Savenije, H. H. G. (2009). HESS Opinions. *The art of hydrology"." Hydrology and Earth System Sciences*, *13*(2), 157.

Schürz, C., Hollosi, B., Matulla, C., Pressl, A., Ertl, T., Schulz, K., & Mehdi, B. (2019). A comprehensive sensitivity and uncertainty analysis for discharge and nitrate-nitrogen loads involving multiple discrete model inputs under future changing conditions. *Hydrology & Earth System Sciences*, *23*(3).

Torri, D., Poesen, J., & Borselli, L. (1997). Predictability and uncertainty of the soil erodibility factor using a global dataset. *Catena*, *31*(1-2), 1-22.

Van der Knijff, J., Jones, R., & Montanarella, L (2000). Soil Erosion Risk Assessment in Europe, EUR 19044 EN., Tech. rep., European Soil Bureau, European Comission.

---

## Author Response (AR1)

**Response to reviewer comments on hess-2019-602**

June 20, 2020

Dear Dr. Nunzio Romano,

herewith we submit the revised version of the manuscript hess-2019-602 "A systematic assessment of uncertainties in large scale soil loss estimation from different representations of USLE input factors – A case study for Kenya and Uganda". We would like to thank you for handling the manuscript and would also like to thank the three anonymous reviewers for their constructive reviews. We have considered and responded to all comments in the revised version of the manuscript to the best of our knowledge. Below we added all reviewer comments, our initial suggestions to revise the manuscript, a detailed documentation of the decision and changes we made for the manuscript revision, as well as a document indicating the differences between the revised and the original version of the manuscript.

We hope the revision of the manuscript satisfactorily replies to all of the comments made by the reviewers accordingly. If there are any further questions or any further issues from our side to handle, please contact me and we will try to clarify them as soon as possible.

Sincerely,
Christoph Schürz

**Reply to the reviewer comments RC1: 'Hess – USLE model Uganda' by Anonymous**

**Referee #1**

We would like to thank the Anonymous Referee #1 for the very constructive review and the valuable comments to improve the quality of the manuscript. We appreciate the positive feedback on the manuscript. In the following, we addressed each comment individually. The reviewer comments are printed in *serif, italic font*. Our replies to the individual comments are written below each comment in black non serif font. The actual changes in the revised version of the manuscript are outlined in blue non serif font. The literature that was cited in the reply is added at the end of the document.

*An interesting article proposing interesting aspects in USLE modelling: a) uncertainties b) comparison of factors c) validation. However, there are issues that authors should face in order to improve the quality and proposing it for publication.*

*An important issue is that authors did not propose 'solutions'. They did the 756 USLE simulations but they should also propose which is the most representative one per factor. For example , which is the best method for the R-factor?*

All realizations that were developed for each USLE input are based on methodologies that were proposed and implemented in peer reviewed studies. In the summary of methods for the calculation of (R)USLE inputs that was compiled by Benavidez et al. (2018) most of the methods that were implemented in the present manuscript were listed as 'valid' methods to compute the respective USLE input. The basic rule for the input generation in this manuscript, was that if a method was implemented in a peer reviewed study in Eastern Africa (or regions with comparable climatic/topographic/vegetation conditions) before, it is considered as a plausible method for the generation of that input. From this perspective , all input realizations must be treated as equally adequate representations of that input.

The aim of this study was to assess the uncertainties which are inherent in the calculation of the long-term mean annual soil loss simply due to the choice of the methods for the calculation of the USLE inputs. It has never been the intention of this study to identify a 'best' realization for a USLE input. Any attempt to identify plausible or implausible USLE model combinations would have failed, as no measured (or other) data was available that could support a decision to verify or falsify a model combination. This was addressed in detail in the Sections 4.4 and 5.2 that highlighted the limitations of a comparison to the limited observation data that is available for the investigated study region. We suggest to revise these sections and try to strengthen the stated arguments to clarify the present limitations of model falsification.

In the revised version of the manuscript we added the section 5.2 "USLE input realizations - Ranges, plausibility, and their comparison to other studies" in the discussion. The section puts the input realizations into a reference with other large scale studies and contains some thoughts on the plausibility of some of the calculated input factors.

*Authors propose new approaches to check the plausibility of large scale assessments based on Bosco et al., 2014. This study has neither been peer reviewed nor published. So, I would suggest using published literature studies for such statements.*

Bosco et al. (2015) briefly outline the methodology that is explained in much greater detail in Bosco et al. (2014). While Bosco et al. (2015) is a peer reviewed article, we agree that Bosco et al. (2014) is published as a pre-print version that was not peer-reviewed. In the present manuscript we always refer to both articles when we mention the applicability of remote sensing data for a plausibility check of the USLE simulations. Although Bosco et al. (2014) is not peer reviewed it provides valuable information that can be accessed via a DOI and therefore the same document is available to the reader of this study to which we referred to when compiling this work.

Based on our argument we kept the reference Bosco et al. (2014) in the revised version of the manuscript. Yet, we specified sections where we refer to the methods proposed by Bosco et al. (2014) (e.g. P27 L26-28 of the previous version of the manuscript).

*In a similar study, Estrada-Carmona et al (2017) made a global sensitivity analysis of USLE input factors. Please compare the results of your study with the ones of this study.*

Thank you for drawing our attention to this study. Although the approach presented in Estrada-Carmona et al (2017) differs from the approach that was presented in the present manuscript we should and will refer to this study.

In the revised version of the manuscript we added a brief literature review on the implementation of uncertainty and sensitivity analysis to evaluate the soil loss estimates and the impact of the USLE inputs on the soil loss estimates on P4 L2ff

Further, results of the reviewed studies that were summarized in the introduction were included in the discussion section 5.3.

*The most recent study that I found in East Africa is the one of Fenta et al., 2019 Science of the Total Env. How your results compare with their results?*

The study of Fenta et al. (2020) was not published by the time this manuscript was compiled. The workflow that is presented in Fenta et al. (2020) was adopted from Borrelli et al. (2017), Panagos et al. (2015), and Panagos et al. (2014) for the computation of the USLE K and C factors. To calculate the LS factor the method of Desmet and Govers (1996) was implemented. These three realizations for the inputs K, LS, and C are also members of the input sets in this study and should therefore result in comparable ranges. Interesting aspects in Fenta et al. (2020) are the computation of R and more importantly the consideration of soil protection measures represented by the P factor. Thus, a comparison of the calculations presented in Fenta et al. (2020) with the calculated ranges of soil loss in this study is valuable. We suggest to refer to Fenta et al. (2020) in the revised version of the manuscript.

We set the USLE ensemble soil loss estimates into reference with the findings reported in Fenta et al. (2020). Fenta et al. (2020) provided country wide mean soil losses, these were compared to the ensemble median, minimum and maximum soil loss estimates calculated with the USLE model ensemble and averaged for both countries. The comparison was added in section 4.1. Fenta et al. (2020) also report on area proportions for soil loss levels in the countries of East Africa. We compared their findings with the results of the USLE model ensemble and added the following figure to the revised version of the manuscript:

[Figure]

**Figure 4.** *Comparison of the proportions of the areas in Kenya and Uganda that are summarized with different soil loss levels. The comparison shows the results reported in Fenta et al. (2020) to the results of the 972 USLE model realizations. The analyzed quantiles represent the soil loss quantiles in each grid cell that result from the USLE model ensemble. For the comparison the soil loss levels applied in Fenta et al.(2020) were used.*

*Soil losses estimates with USLE are long-term averages. You cannot compare the long-term findings against short-term findings in plot experiments.*

We agree that a comparison of long-term soil loss to short term measurements of e.g. sediment yields is improper. In the manuscript we specifically indicate the limitations of the comparability of USLE simulations to an instream monitoring of sediment yield. Yet, it is a common practice to employ short term records of observed soil loss (or sediment yield) to validate the results of a USLE model. See e.g. Fenta et al. (2020) where a comparison of USLE results to 'measured' short term sediment yields was performed.

Thus, we see a relevance to critically address the issue of a USLE model comparison to observation data. Eventually, the analysis of the USLE model ensemble and its comparison to the soil loss data collected by García-Ruiz et al. (2015) clearly supports a critical view on such model validation and is therefore in our opinion a relevant contribution to the soil erosion literature.

Based on our arguments stated above we think it is relevant to keep the sections on model validation. Thus, the sections 3.7, 4.4, and 5.2 are still present in the revised version of the manuscript.

*Authors made a classification of soil erosion rates. The tolerable soil erosion rate cannot be justified according to literature findings as the soil formation rates are low. This means that sustainable soil erosion rates are lower than 1-2 t per ha per year. In addition to this, authors present some really extreme mean annual soil loss rates >200 -1,500 t ha-1 yr-1. This means that at least 2 cm of soil is lost every year. This may be the case for very limited areas; otherwise we risk to lose completely our soils in50 years. This means that some of the estimated combinations are not realistic. You should not be driven by the modelling outputs but somehow use also the common logic(you cannot lose 1m of soil in 50 years).*

We have refrained from making judgments about tolerable soil erosion rates, but we do point out that such values exist in the literature. The concept of this manuscript was to employ the information that we acquired from the peer reviewed literature to represent the current status of knowledge on the topic of "soil erosion by water" and the assessment of water erosion with USLE type models. This idea also is applied to the selection of soil loss classes that employ what is common practice in the literature. At no point in the manuscript do we imply that a soil loss of 10 t ha$^{-1}$ yr$^{-1}$ is indeed the value where the soil loss is compensated by the soil formation. We stated that suggested literature values range from 5 to 12 t ha$^{-1}$ yr$^{-1}$ and that several studies in Eastern Africa used 10 t ha$^{-1}$ yr$^{-1}$ as a threshold value. Due to the absence of more reliable values, we based the classification on literature values that included threshold values of 10 t ha$^{-1}$ yr$^{-1}$. Yet, we agree that threshold values of 1 or 2 t ha$^{-1}$ yr$^{-1}$ would be valid assumptions as well. Fenta et al. (2020), for example, classify soil loss by water as 'very slight' soil loss when the soil loss is in a range between 0-2 t ha$^{-1}$ yr$^{-1}$ and as 'slight' when the soil loss is in a range between 2-10 t ha$^{-1}$ yr$^{-1}$. Yet, Fenta at al. (2020) do not provide any reference on which their classification is based. In conclusion the classification of soil loss always involves a highly subjective view on the calculated soil losses. A classification should primarily reduce information and support the reader in the interpretation of data and we are aware that poorly chosen class names can mislead the reader. Nevertheless, we decided to use a classification that was implemented in the literature before and should be considered to be as valid as any other classification.

We agree with the reviewer that calculated soil losses of over 200 t ha$^{-1}$ yr$^{-1}$are extreme. Nevertheless, for the informative value of this manuscript it is relevant to keep these model combinations as potential USLE realizations. Two arguments for the value of these model members are as follows:

Indeed such high values were calculated only locally and only by a few model combinations. This is exactly what this manuscript tries to address. Models can be wrong. Thus, if other representations of the USLE estimate soil losses that are substantially lower then the modeler has a chance to evaluate such large soil loss estimates based on other estimates. Yet, with a single model approach (that is common practice in the literature) it is simply infeasible to evaluate large calculated soil losses if no reference by observation data or a model ensemble is given. Thus, if a model setup calculated excessive soil losses locally, what does this mean for the evaluation of the remaining areas in a study region?

USLE type models do not account for soil deposition and therefore do not reflect the sediment balance. Thus, the soil that is strongly eroded locally is also deposited and not completely lost but displaced.

We think that we substantially commented on the issue raised by the anonymous referee #1 above. Thus, we prefer to keep the class boundaries that we used in the classification in the previous version of the manuscript. We changed however the term 'tolerable' to 'slight' and added a sentence to explain why we avoid using the term 'tolerable'.

To allow a better comparison of the model ensemble simulations (and particularly the average and extreme ranges of the estimated soil losses) we referred to the findings of Fenta et al. (2020) (see comment above)

*Title: I would replace the word 'representations' with 'applications'*

We think that 'applications' is not the appropriate term for the entities that we analyzed in this work. 'Applications' could also imply an application of USLE inputs in for instance different independent studies, or locations and would thus be misleading. In this manuscript we represented the USLE input factors R, K, LS, and C for the same study by employing different methods to compute them and would therefore prefer to use the term 'representation' or 'realization' when we refer to one member in the set of realizations for a USLE input.

We kept the terms 'representation' and 'realization' to define a layer that was calculated for one of the USLE inputs employing a specific method from the literature in the revised version of the manuscript.

*P4 L15-24: This paragraph is not needed.*

We agree that this paragraph does not provide any new information, but outlines the structure of the manuscript. We think that this is a subjective question of style and preference and believe that it helps the reader to get an overview of the content of the paper at hand.

Based on our arguments we preferred to keep the paragraph in the revised version of the manuscript.

*Fig. 3: attention in the measurement unit of soil erosion. It is better to use t ha-1 a-1. If you want to keep your proposed unit , then please put in parenthesis (ha a)*

Thank you for pointing this out! We will revise the units in all figures to be consistent with the units in the text and tables.

All figures in the manuscript were revised. The units in all figures and maps are now written in round brackets. The unit style now agrees with the style that was used in the text of the manuscript.

*Fig 7. It should be applications and not realizations.*

See the response above. The same argument applies as above for the title of the manuscript.

See the comment above.

*P23 end of the page and P24 beginning of the page: I would propose that some applications of the factors can be excluded. For example, the NDVI application is known to have very low C-factor results and it is known to have incorporated some problems. The same applies for R-factor. For example the methods of Lo and Fournier are based on rainfall amount and do not incorporate the rainfall intensity.*

As responded previously, the goal of this work was to provide a comprehensive assessment of frequently implemented methods to calculate the USLE input factors and to evaluate the uncertainties in the soil loss estimates that arise from the input uncertainties due to the impact of different methods, published in peer-reviewed journals, to calculate the model inputs. Thus, it was not our intention to judge the implemented methods, but to consider them as potential methods if they have been implemented in similar study settings. This also applies to the methods used to calculate the R factor and the C factor.

Both methods addressed by the Anonymous referee #1 (the implementation of the NDVI and the method of Lo et al. (1985)) were recently implemented in a large scale soil loss assessment by Karamage et al. (2017). In general, the implementation of methods that use long-term precipitation instead of rainfall intensity is common due to the absence of rainfall intensity records. Thus, in terms of a comprehensive uncertainty assessment we must consider these types of methods for the calculation of C and R as well.

Concerning the limitations of any implementation of the NDVI to calculate C, we were not able to find information in the literature that NDVI is known for a calculation of low C-factor values. Yet, the analysis that we illustrated in Fig. 7 of the manuscript would support that statement for the selected region in Uganda. Also, any well documented issues with the application of the NDVI to calculate the C-factor is not known to us (or is at least not reported in the literature at hand). However, we specifically address in the manuscript that the method of Van der Knijff et al. (2000) was never validated against ground truth data.

Concerning the C-factor value ranges calculated with the method of Van der Knijff (2000) we want to take the analysis in Fig.7 of the manuscript as an example. Although the mean values and the quantile values for the C-factor with the NDVI are significantly lower to the other methods, the absolute ranges of C-factor values when employing the NDVI are in a plausible range. For Europe Panagos et al. (2015) calculated C-factor values as low as 0.00116 for forests and C-factor values of up to 0.2651 for sparse vegetation. The ranges presented in this manuscript are in line with the range presented in Panagos et al. (2015).

We see however that a comparison to other large scale studies would provide valuable information and will consider that in the revised version of the manuscript.

In the revised version of the manuscript we added a section in the discussion (section 5.2) to compare the values of the calculated USLE input factor realizations to the ranges in other large scale studies.

*P19 the same as above. Are values of K-factor 0.088 acceptable? Can be compared to other findings in the literature?*

Panagos et al. (2014) for instance calculated a range for the K-factor of $0.004 – 0.076$ t h $MJ^{-1}$ $mm^{-1}$. Naipal et al. (2015) implemented the method of Torri et al. (1997) that resulted in such large K-factor values in this manuscript. Naipal et al. (2015) also applied a K-factor value of 0.08 to volcanic soils as these are particularly erodible. Although single grid cells in the analysis in this manuscript exceed the mentioned literature values, the calculated values are not completely out of range.
Yet, we fully agree that it is relevant to set the calculated values into a reference with other literature. We suggest to add a comparison to the above mentioned literature values and additional literature in the revised version of the manuscript.

See the comment above

*P20 L1-10: The same as above. Can you prove that values of C-factor 0.03 are acceptable in agricultural areas?*

Thank you for raising this point. The same arguments as in our replies to the previous two points apply here.

See the comment above.

*5.2 section. It is not proper to have a section with question.*

We cannot verify this point, as the manuscript guidelines do not disallow questions as section headers. This should be, in essence, the freedom of the authors. See Blöschl and Montanari (2010), Savenije (2009), or Schürz et al. (2019) as examples with questions as section headers.

The discussion section was completely restructured. Thus the section headers also changed.

*P25 L30-34: Is It possible to validate large scale models with Google Earth? GoogleEarth can potentially verify permanent erosion characteristics (e.g. gully erosion) and not rill and sheet erosion. The plausibility of large scale studies can be verified with model applications at regional or local*

We cannot confirm the feasibility of the implementation of satellite imagery for the validation of soil loss assessments. It was not our intention to illustrate such an approach as a verified method for the validation of large scale soil loss assessments. From a technical perspective it should however be possible to employ multi-angle high resolution imagery for different time steps to apply any stereographic analysis. Furthermore, intense soil erosion might be visible in satellite imagery and can be related to an erosion class, which would allow a use as 'soft' information for model validation. Regardless of the exact approach to employ satellite imagery, the conclusion in the presented manuscript was that our traditional approaches of model validation failed in this large scale study (and would likely fail in others)

and new concepts for model validation would be valuable to be implemented in large scale assessments. The validation of a large scale model application with a model application at a smaller scale does not sound plausible unless the small scale model is validated, as any soil erosion model involves uncertainties.

**Reply to the reviewer comments RC2: 'Referee comments and suggestions' by Anonymous Referee #2**

General comments

*The paper presents a large scale assessment of the uncertainties in USLE soil loss estimation as a consequence of different realizations and combinations of the corresponding input factors. A total of 756 USLE model setups were examined with a spatial detail of 90 meters (cell size). Moreover, the case study (Kenya and Uganda) is vast enough to include a great variability of topographical, climatic and land use conditions. For these reasons, the ranges of both input factors and soil loss are very wide, contributing to improve the scientific reliability and interest of the work. The spatial variability of the model sensitivity to the different factors was examined and discussed. An attempt to compare/validate the simulated soil loss with field soil loss data was also made. All the sections of the paper are very clear and the scientific background is well detailed and discussed. The degree of agreement between the estimates obtained by the different input ensembles was evaluated not only on the basis of the quantitative values, but also and above all on the basis of the soil loss category (tolerable, moderate, high and severe). This is in fact the most rational approach for a model characterized by high uncertainty.*

We would like to thank the Anonymous Referee #2 for their positive and supportive feedback on this manuscript, the very constructive review and the valuable comments to improve the quality of the manuscript. In the following, we addressed all the comments made by the Anonymous Referee #2. The reviewer comments are printed in *serif, italic font*. Our replies to the individual comments are written below each comment in black non serif font. The actual changes in the revised version of the manuscript are outlined in blue non serif font. The literature that was cited in the reply is added at the end of the document.

Specific comments

*Lines 3-6 pag. 3. I suggest to mention other recent promising modifications of the USLE, such as those proposed and tested by Bagarello et al. (2010) and Di Stefano et al. (2019): - Bagarello, V., Ferro, V., Giordano, G. 2010. Testing alternative erosivity indices to predict event soil loss from bare plots in Southern Italy, Hydrological Processes 24(6), 789-797. - Di Stefano, C., Pampalone, V., Todisco, F., Vergni, L., Ferro, V. 2019. Testing the Universal Soil Loss Equation-MB equation in plots in Central and South Italy, Hydrological Processes 33(18), 2422-2433*

Both suggested references describe the modification and improvement of the event based USLE-M model (Kinnell, 2010). In the manuscript the topic of event based soil loss assessment with USLE type models was not addressed, but we focused on long-term annual soil loss assessment to limit the breadth of the manuscript content. To write an additional paragraph and acknowledge all (or many) USLE derivatives would be out of the scope of this study.

Based on our arguments, we preferred to keep a focus on the "traditional" form of the (R)USLE and analyze it with respect to uncertainties that result from different realizations of the USLE inputs. Thus, we refrained to additionally discuss further USLE derivatives. Yet, as the focus should be uncertainty and sensitivity analysis, we added a section on uncertainty and sensitivity analysis with USLE type models in the revised version of the manuscript on P4 L2ff and in the discussion section 5.3 to provide a better foundation for our work.

*Figure 1. I suggest to check the legend of the figure 1a, in which the erosion risk is represented according to a discrete classification based on only three colours (white, yellow and pink). However, from the figure, the colour grey is also widely present and gradients for both yellow and pink are evident. I think that a discrete classification/legend is not correct.*

We understand that the maps together with the provided legends can be confusing to the reader. Therefore, we suggest to revise the legends and the figure caption of Fig.1 the following:
- The legend symbol for 'Very gentle inclinations…' will be changed to add a slash in the box, thus indicating that no color was applied for that class.
- Will will add the information that the hillshade is plotted in grey as a background layer.

The discrete erosion risk classes were taken from Ebisemiju (1988) and provide a reasonable first differentiation of erosion risk in the study area. We tested a continuous color ramp to visualize the soil risk while compiling the manuscript. The information was however not presented well in such a visualization. Therefore, we would prefer to remain with only three soil risk classes.

Fig. 1 was strongly updated in the revised version of the manuscript. We tried to improve Fig. 1 as follows:

- The legend symbol for 'Very gentle inclinations…' was changed and a slash was added in the box to indicate that no color was applied for that class.
- The information that the grey background shows the hillshade was added in the figure caption.
- The unit style was changed to match the units in the text.
- A panel showing the Seasonality Index (SI; Walsh and Lawler, 1981) was added to provide information on the spatial distribution of the rainfall seasonality (see also comment below)

The updated figure is illustrated below:

[Figure]

**a) Erosion Risk from Topography**

⬜ Very gentle inclinations (< 3°) - first domain of sheet erosion

🟨 Moderate to steep slopes (3° to 20°) - domains of active gully erosion & growth

🟪 Very steep slopes (> 20°) - prone to mass movement, severe rain splash and sheet erosion

**c) Mean Annual Rainfall (1970-2000) (mm)**

| | |
|---|---|
| 🟫 below 250 | 🟩 1 000.1 - 1 250 |
| 🟧 250.1 - 500 | 🟩 1 250.1 - 1 500 |
| 🟨 500.1 - 750 | 🟦 1 500.1 - 2 000 |
| 🟨 750.1 - 1 000 | 🟦 over 2 000.1 |

**b) Mean Annual NDVI (2001-2018) (-)**

0 - 0.2   0.2 - 0.3  0.3 - 0.4  0.4 - 0.5  0.5 - 0.6  0.6 - 0.7  0.7 - 0.9

**d) Seasonality Index (SI, 1970-2000) (-)**

🟦 very equable (SI: ≤ 0.19) (not present)

🟦 equable but with a definite wetter season (SI: 0.20 - 0.39)

🟩 rather seasonal with a short drier season (SI: 0.40 - 0.59)

🟩 seasonal (SI: 0.60 - 0.79)

🟨 markedly seasonal with a long drier season (SI: 0.80 - 0.99)

🟫 most rain in 3 months or less (SI: 1.00 - 1.19)

🟥 extreme seasonality (SI: ≥ 1.20) (not present)

**Data sources:** *DEM and hillshade:* SRTM 90m (Jarvis et al., 2008), *NDVI:* MODIS MOD13Q1 (Didan, 2015), *Precipitation:* WordClim Vers. 2 (Fick and Hijmans, 2017), *Administrative boundaries, Cities, Water Surfaces and Rivers:* naturalearthdata.com, *Water Surfaces:* Carroll et al. (2009)

0    250    500    1 000 Km

N

*Figure 1. I understand that the purpose of Figure 1 is just to provide a rough description of the erosion-prone areas according to topography, vegetation cover and rainfall amounts. In relation to this last aspect, however, the authors could have chosen a proxy more appropriate than the annual precipitation: in fact it is well known that the distribution of rains has a determining role in soil loss. In particular, several studies in the literature have shown that in some areas, the annual soil loss is highly correlated with the erosivity of a few erosive events. Therefore, other synthetic indices (e.g. the Modified Fournier Index (Arnoldus, 1980) could be proxy more reliable than annual precipitation in the description of the susceptibility to erosion due to rainfall characteristics).*

We think that the Modified Fournier Index (MFI, Arnoldus, 1980) can be an interesting index to characterize the erosion risk in the study region. Therefore, we suggest to calculate the MFI for the study region and analyze the spatial pattern. If the shown patterns strongly differ from the patterns of the shown long-term annual precipitation we suggest to add a panel to additionally show the MFI in Fig.1.

Please see the comment and the Figure above to also see the changes that were made based on this comment.

*Lines 5-13 pag. 8. As stated by the authors themselves (section 5.3), it is not possible to consider all the available methods for the calculation of USLE input factors and the authors made plausible choices in their selections. However, the authors started their analysis of the R factor by aggregating the long-term monthly amounts to the annual scale, thus losing the possibility of applying the methods that derive the R factor from both annual and monthly precipitations. The reasons for this choice should be provided.*

We agree that the aggregation of long-monthly precipitation to long-term annual precipitation reduces the information that is provided by the data. The simple reason why primarily long-term annual precipitation was implemented to calculate the rainfall erosivity factor, was that the literature on large scale soil loss assessments as well implemented primarily long-term annual precipitation products to calculate the rainfall erosivity.
We agree with this comment that this decision in the analysis must be discussed. We suggest to perform a comparison of the application of the MFI (Arnoldus, 1980) with the methods that were implemented in the manuscript. We suggest, however, to add any analysis in the supplementary materials and add a section on monthly rainfall erosivity in the discussion.

To account for the seasonality of the rainfall we additionally added one realization to the set of R factor realizations that employs the MFI to calculate R factor values. The set of realizations for the rainfall erosivity R now includes two additional realizations that were presented in Fenta et al. (2017), where one method employs long-term mean annual precipitation to calculate R and the second method uses the MFI to account for the seasonality of the rainfall. Fenta et al. (2017) applied both methods in a large scale study in Eastern Africa. As a result 972 realizations of the USLE model are analyzed in the revised version of the manuscript, compared to the 756 realizations that were presented in the previous version of the manuscript.

*Section 5.2. the discussion presented in this section was expected since the authors described in section 3.7 their intent to compare simulated yields with those collected from field observations. I agree that there are several limitations and difficulties, but the attempt is appreciable. I wonder if another possible reason for the lack of agreement could be represented by the differences between the land use at the time of field experiments and the average one considered in the simulations, (e.g. Sutherland and Bryan (1990) refers to experiments carried out before 1990, whilst the MODIS NDVI data are from 2000 to 2012).*

We think that the Anonymous Referee #2 raises a very relevant point here. In the manuscript we addressed a few selected, but certainly dominant limitations for the comparison of in-field data with the calculated soil losses. Yet, other possible sources that can potentially limit a comparison were not mentioned (such as the addressed impact of land use change, particularly as deforestation in the previous century is a frequently mentioned issue for soil loss in Eastern Africa). We suggest to add other potential limitations for the comparability of in-field data to the calculated soil losses in the discussion section 5.2.

We added the following paragraph in the discussion in section 5.2:

Apart from the short monitoring periods that are often available from reference studies it is likely that the (remote sensing) data that was employed to calculate the USLE input factors and to assess the soil loss do not reflect the conditions that were present during the monitoring period in a study region, simply because the monitoring period and the period for which input data are available do not overlap. Soil cover by vegetation perfectly illustrates the issue. Monitoring data can date back several decades (e.g. Sutherland and Bryan (1990) in our case). On large scales the vegetation cover is often estimated by employing remote sensing satellite data that can be more recent than monitoring data. Particularly, in East Africa deforestation affected the land cover over the past decades with reported decreases in the forest biomass of up to 26 % in Uganda (Jagger and Kittner, 2017), or forest clearances in protected forests in the Mt. Elgon region of 33% (Petursson et al., 2013). In such a case, a C factor that was calculated with recent remote sensing data would fail to reflect the condition of the vegetation during the monitoring period.

*Fig. 8a. In order to improve the clarity of the boxplots in figure 8a, I suggest to eliminate the dots, whose presence is not much effective since the data spread can be derived from the length of the whiskers of the boxplots. A similar consideration holds for fig. 9 and S1 and S2 in the supplement material.*

From our experience the statistical summary measures illustrated by boxplots strongly reduce the information provided by data and can be misleading with small sample sizes and strongly non-normal distributions. We therefore prefer to also show the data that results in the illustrated boxes.

We preferred to keep all data points in the plots of the figures Fig. 8, Fig. 9, and the figures in the supplementary document.

*Pag 1 line 8: "challanges" should be "challenges" Pag.19 line 32 check the sentence Pag. 26 line 9 replace ULSE with USLE*

Thank you. We will revise the misspellings accordingly.

The typos were changed accordingly in the revised version of the manuscript

**Reply to the reviewer comments RC3:   'referee comments', by Anonymous Referee #3**

General comments

*This paper presents an analysis of the variability in soil loss estimates with the USLE equation due to different representations of its factors, and subsequent comparison of the predictions with field data. It is certainly not the first time that the uncertainty of erosion predictions with the USLE is questioned. Yet, the fact that the USLE is very often applied using very different data and methods to determine its input factors still make the study relevant.  The study uses a representative selection of frequently used methods to determine the USLE factors based on readily available land use climate, soil and topography data.  The paper is generally well written, but could be more concise at some points and there are some issues that require better explanation or justification, as explained below.*

We would like to thank the Anonymous Referee #3 for the highly detailed review of our study and the constructive comments to improve the quality of the manuscript. We appreciate the positive feedback on the manuscript. In the following, we addressed each comment individually. The reviewer comments are printed in *serif, italic font*. Our replies to the comments are written in black, non serif font. The actual changes in the revised version of the manuscript are outlined in blue non serif font. The cited literature is added at the end of the document.

*The authors rightly argue that there is a huge range in erosion rates predicted in function of the methods used to obtain the model input factors. What is interesting however is that the ensemble prediction shows relatively good agreement regarding the predicted erosion severity class. So, although agreement with measured erosion data is poor, in line with earlier studies, you might argue that such ensemble prediction is useful for qualitative description of erosion severity.  This can be helpful to prioritize policies.*

We appreciate the positive comment. Yet, we think that this perspective on the agreement of the ensemble with respect on soil loss levels is a little too optimistic. Fig. 4 (and particularly panel a)) shows a strong agreement of the model ensemble for soil losses below a threshold of 10 t ha$^{-1}$ yr$^{-1}$ that was defined as "tolerable" soil loss in this study. Thus, the model ensemble is able to identify regions with no or a low erosion risk. In potentially erosion prone areas, however, a large spread is visible in the ensemble prediction of the soil loss classes. This is particularly visible for large parts of the Rift valley and the South-West of Uganda where large ranges from "tolerable" to "high" are visible.

Fig. 5 was intended to support the reader to assess the confidence of the model ensemble to predict a soil loss level. From that perspective we think, however, that the picture that is conveyed by Fig. 5 can be too optimistic and gives the impression of a strong agreement for the areas that were analyzed in detail (see panes c) to d)). Thus, we suggest to revise Fig. 5 to improve the readability of the probability of models in a class in the figure.

Despite the described limitations we think that the argument that the Anonymous referee #3 stresses is highly valuable. Section 5.1 briefly discusses the relevance of the model ensemble to provide information on the confidence to predict the severity of the soil erosion

risk. We suggest, however, to strengthen this argument and add a short paragraph on the potential of a model ensemble to support policy making in Section 5.1.

The suggested section was implemented as section 5. in the revised version of the manuscript.

*However, the comparison of predicted soil loss with measured erosion and sediment yield data is most problematic. As the authors also mention at some point, the USLE does not consider sediment deposition and transport so it cannot be compared with sediment yield from gauging stations. On the other hand, the erosion rates provided by De Meyer et al. (2011) based on reconstructing the historic surface level and calculating the lost soil volume from 36 farm compounds are extremely high. I am not sure which method was applied exactly by De Meyer et al and for what time and spatial scale the assessments are made for example. In any case, such high values can occur at certain points, but are probably not realistic for larger areas. So, the question is how useful are these comparisons actually, and do we need them to assess the uncertainty in USLE predictions due to variations in its factors? Model validation is very important, but only useful if the modelled and measured data refer to the same processes and the same scales of assessment.*

We fully agree that a comparison of soil loss calculations with USLE type models to in-field data is problematic. We think, however, that it is relevant to illustrate potential issues that arise from a comparison of soil loss calculations to in-field data that may stem from different types of erosion monitorings, simply because it is frequently performed in erosion studies and we think that such common practice must be critically evaluated. As this point was addressed by other referees in a similar way we think that it is relevant to better specify the intention of the model validation as it was performed in this study. We therefore suggest to revise section 3.7 of the manuscript accordingly.

In the revised version of the manuscript we want to emphasize that a comparison must be done with great care due to the differences in the approaches that were employed the soil loss in the field and the processes that are actually considered in the soil loss estimation. Thus, we revised the discussion section on the comparison to in field monitorings (now section 5.4)

*I find the classification of the predicted soil loss values in four classes, below and above tolerable soil loss rate of 10 t/ha/yr, problematic and it does not add much to the entire discussion of the uncertainty of model predictions. First of all, the tolerable soil loss rate depends on a spatially variable soil production rate, which is unknown for the area. Secondly, the USLE soil loss predictions are gross erosion rates and do not account for deposition during transport over distances longer than a standardized erosion plot. This makes it highly arguable to look at the USLE predictions in relation to tolerable soil loss rates. You can classify the predictions in erosion severity classes but I recommend to delete reference to tolerable soil loss rates.*

Thank you for this advice. We agree that the selected terminology is highly critical, particularly due to the arguments that were stated by the Anonymous referee #3. The term "tolerable" seems to be too specific and implies that process relationships are known that are indeed unknown. We therefore suggest to add a paragraph in section 3.4 to indicate that the actual soil formation rate is unknown, but that we use this threshold as it is frequently used in other literature. We additionally suggest that we change the term "tolerable" to e.g. "slight" in

the revised version of the manuscript and indicate that we refrained from using the term "tolerable" due to the above stated issues.

The following paragraph was added in section 3.4:

In this study low soil losses were classified by employing the same threshold. Yet, no information on soil formation was included and thus the term *tolerable* is misleading. Consequently a soil loss between 0 and 10 tons ha$^{-1}$yr$^{-1}$ is defined as slight soil loss, as suggested by Fenta et al. (2020).

Throughout the revised version of the manuscript the term *slight* is now used instead of *tolerable*.

*What exactly is the aim of the comparison of soil loss estimates at the administrative level? How does this contribute to the research objectives explained in the introduction? While it can be an interesting exercise, and may provide relevant information for local policy makers, it seems the whole section 4.3 does not really contribute to the main objectives of your study.*

We agree that the analysis of the mean soil loss on an administrative level is not specifically outlined in the introduction. Yet, often erosion studies perform the analysis of the soil loss (or any related measures) for defined administrative units (see e.g. Gourevitch et al. (2016), or Karamage et al. (2017) for districts in Uganda, or Panagos et al. (2015b) for provinces at the NUTS3 level in Europe). In theory one could assume that local disagreements of the model combinations will be reduced by a spatial averaging and as a consequence the input uncertainties are less critical for a soil loss assessment on the administrative level. Yet, the uncertainty analysis on the administrative level shows that large uncertainties are present in the aggregated soil loss estimates. Thus, any decision making based on a single USLE model estimate is highly questionable. The comparison with the results of Karamage et al. (2017) for Ugandan districts clearly illustrates this issue. Therefore, we conclude that the results presented on the administrative level provide essential insights in the evaluation of uncertainties in soil loss estimation. We agree, however, that our intention with the soil loss assessment on the administrative level was not specified well enough. Therefore, we suggest to add a brief section to outline the intention sketched above.

In the revised version of the manuscript we added the comparison to previous studies in the region that employ single USLE models to the research questions.

The focus of the analysis on the administrative level was shifted from the simple analysis of aggregated soil losses towards the analysis of the ensemble soil losses aggregated for different administrative domains and their comparison to studies that calculated the soil loss for the same administrative units. In the revised version of the manuscript an analysis on a national level compares the calculated soil losses to the ones reported in Fenta at al. (2020) additionally to the comparison on the administrative level and the comparison to the results of Karamage et al. (2017) that was already shown in the previous version of the manuscript.

Therefore, the methodology in section 3.6 and the results section 4.3 was revised. A more detailed discussion of the comparison to Fenta et al. (2020) and Karamage et al. (2017) was added in the discussion.

*Please explain and illustrate with quantitative data why you did not include the ASTER DEM for calculation of the LS factor. Previous studies have also highlighted that at higher resolutions problems can occur with LS calculations, but since you first projected the ASTER DEM on the 90 SRTM grid could be expected to be less problematic. It would be interesting to see what is exactly the cause of this problem and compare this to other studies that assess the differences in ASTER and SRTM DEMs and their application in erosion studies.*

We agree that the found issues with ASTER DEM to calculate the LS factor could be elaborated with more detail. Due to the length of the present manuscript we, however, tried to keep this section concise. We suggest to add a section that illustrates the artefacts in the calculated LS factor realizations using ASTER DEM in the supplementary materials.

We added the Figure and the text below in the supplementary document:

[Figure]

**Slope length and steepness factor LS after Desmet and Govers (1996) using different DEM inputs (-)**

0 - 1    1.1 - 5    5.1 - 10    10.1 - 25    > 25.1

**Data sources:** *DEM and hillshade:* SRTM 90m (Jarvis et al., 2008), ASTER GDEM V2 (NASA/METI/AIST/Japan Spacesystems, and U.S./Japan ASTER Science Team, 2009), *Administrative boundaries, Cities, Roads, Water Surfaces and Rivers:* naturalearthdata.com, *Water Surfaces:* Carroll et al. (2009)

Figure S.1: LS factor realizations that result from SRTM 90m (Jarvis et al., 2008) a) and ASTER DEM (NASA/METI/AIST/Japan Spacesystems, and U.S./Japan ASTER Science Team, 2009) b) when employing the method of Desmet and Govers (1996) for the LS calculation. The map detail indicated in panel a) is shown for SRTM 90m and ASTER DEM in the panels c) and d), respectively. ASTER based calculations show random and systematic noise in the detail in panel d).

As outlined in the main document of this work, the realizations for the LS factor that used ASTER DEM (NASA/METI/AIST/Japan Spacesystems, and U.S./Japan ASTER Science Team, 2009) for their computation were excluded from the analyses as issues were encountered in the simulation results that stem from noise and artefacts in the ASTER DEM inout data. As a result, only three instead of six realizations for the LS factor were used. Fig. S1 shows the LS factor realizations that result from SRTM 90m (Jarvis et al., 2008) and ASTER DEM when employing the method of Desmet and Govers (1996) for the LS calculation. The panels a) and b) show the results for the entire study area. The panels c) and d) show the detail located in the North-East of Kenya. While the LS calculations using the SRTM show low values close to 0 for large regions where the terrain is overall flat, the LS realization that implemented the ASTER DEM shows (i) random, non-systematic noise (the yellow scatters in flat terrain with no noteworthy heterogeneity in landscape) and (ii) systematic errors (strips), which originate from satellite malfunction or issues in data processing. (i) and (ii) can be clearly seen in the detail showing the dry, north-eastern part of Kenya. Based on the identified noise patterns (among other patterns e.g. in northern Uganda, that are not shown in detail here) we decided to exclude the LS realizations that are based on the ASTER DEM from the set of LS realizations that were used in the analyses of the manuscript. Although not illustrated here, other methods to calculate the LS factor showed comparable results and supported the assumption that the visible errors result from the input DEM rather than from the computation method itself.

*The methods used to assess the C factor rest strongly on the approach used by Panagos et al (2015) and Borrelli et al (2017), but I find the description quite difficult to follow. It is not clear why and how exactly you overlay the already spatially distributed 'crop shares statistics' with the land cover maps? Moreover, it seems the approach puts a lot of detail in differentiating between different crops, but disregards the possible importance of intra-annual differences in C factors due to crop rotations.*

The method to calculate the C factor that was proposed by Panagos et al (2015a) was adopted in several preceding studies (see e.g. Fenta et al. (2020), Batista et al. (2017), Borrelli et al. (2017), Lugato et al. (2016)). The main concept in this manuscript was to employ frequently used methods to compute the USLE input factors. Therefore, it was also relevant to consider the method of Panagos et al. (2015a) as a member in the set of C factor realizations and to follow the C factor calculation as it is described in Panagos et al. (2015).

Further, the options for the computation of the C factor are often limited by a lack of available crop data. (Nationwide) Agricultural statistics are usually not available for every year. As a consequence, data to consider inter-annual variations in crop statistics are usually not available. A typical assumption that is drawn in the calculation of the C factor based on crop statistics is that the available statistical data is a good average value for the entire analyzed time period.

Statistical agricultural data is not spatially distributed but provided aggregated on an administrative level (see e.g. the National Agricultural census data that was implemented in this manuscript). Also the agricultural data that is available from Monfreda et al. is not fully spatially distributed but aggregated with a spatial resolution of 5 minutes.

*There are several other papers that also discussed the impacts of USLE factors and structure on outcomes (e.g. Sonneveld and Nearing, 2003) that would be interesting to include in your discussion.*

We were not able to identify studies in our literature review that employed similar analyses as the one presented in this manuscript. Thank you for pointing out the study of Sonneveld and Nearing (2003) which we will mention in this manuscript. We suggest to include further relevant literature in the discussion.

In the revised version of the manuscript we added a brief literature review on the implementation of uncertainty and sensitivity analysis to evaluate the soil loss estimates and the impact of the USLE inputs on the soil loss estimates on P4 L2ff

Further, results of the reviewed studies that were summarized in the introduction were included in the discussion section 5.3.

Detailed comments (indicated per Page and Line):

*P2-L17-18: can you add a line how the revised version was different?*

We will add the following to the sentence on P2 L16-17:

Further data were collected over the following decades and the methods to calculate the USLE input factors were substantially revised. This resulted in an update of the iso-erodent maps, the consideration of seasonality and rock fragments in the K factor, or a consideration of additional sub factors for the computation of the C factor.

We added the suggested sentence in the revised version of the manuscript on P2 L17.

*P3-L8-11: you may add here a few words on the often used Sediment Delivery Ratio in combination with gross erosion to obtain sediment yield predictions, correcting for the fact that the USLE does not predict sediment deposition.*

We will add a short section that acknowledges approaches that also account for deposition processes and employ the Sediment Delivery Ratio (e.g. Rajbanshi et al. (2020), De Rosa, et al. (2016), or Sharp et al. (2015)).

On P3 L11 we added a sentence to refer to methods that employ e.g. the SDR to compute the sediment delivery from soil loss estimates.

*P3-L14: remove 'the'*

Will be removed accordingly

'the' was removed.

*P3-L20-23: please check and preferably simplify this sentence.*

We will revise the sentence on P3 L20-23 as follows:

The implemented remote sensing data products describe (or are a proxy for) features in the landscape (e.g. a DEM represent the topography and the NDVI is often employed to describe vegetation density). In large scale assessments methods are implemented that employ these large scale data products to infer spatially distributed estimates for the USLE inputs.

The sentence on P3 L20-23 was replaced by the suggested sentence in the revised version of the manuscript.

*P3-L33-35: It is indeed not simple to do this kind of comparisons and most plot data do not cover 20 years, but there are by now relatively good and large datasets of measured soil loss available, such as for example the data presented by Garcia Ruiz et al (2015)and Maetens et al. (2012) for Europe. For many other parts of the world this is stillmore difficult though.*

We agree. Observation data that was collected by García-Ruiz et al. (2015) in their comprehensive meta-analysis was implemented in the present manuscript to compare the soil loss estimates with. Nevertheless, this is one argument that we wanted to stress with this study, that although such data exists a comparison is not always feasible.

We suggest to additionally mention these data sets on P4 L1 in the following:

Large scale meta-analysis studies of soil erosion plot data and sediment yield records exist, such as García-Ruiz et al. (2015) globally, Vanmaercke et al. (2014) for Africa, or Meatens et al. (2012) for Europe.

We added the suggested sentence on P4 L1 in the revised version of the manuscript.

*P4-L10: Research objectives are now formulated as research questions; better write them as objectives. In the last objective correct 'we we'.*

We think that this is a question of style and preference and would prefer to keep the research questions. We will remove the second 'we'.

We kept the research objectives formulated as questions. The second 'we' was removed.

*P4-L16-24: These lines do not seem necessary, and seem repetitive.*

We agree that this paragraph does not provide any new information, but outlines the structure of the manuscript. We think that this is a subjective question of style and preference and believe that it helps the reader to get an overview of the content of the paper at hand.

We preferred to keep this section in the revised version of the manuscript.

*P5-L4: on the steepest slopes (>20) gully erosion can be expected to be an issue as well.*

We cannot identify where this statement applies in the text.

We realized, that anonymous referee #3 was referring to the figure legend. Yet, we would prefer to keep this classification, as it shows the classification that was done by Ebisemiju (1988)

*P7-L6: what about seasonality of rainfall?*

As discussed in another reply to the comments by the Anonymous referee #2 we argued that the measures such as the Modified Fournier Index (MFI, Arnoldus, 1980) can provide valuable information to characterize the seasonality of the rainfall erosivity. Therefore, we suggest to calculate the MFI for the study region and analyze the spatial pattern. If the shown patterns strongly differ from the patterns of the shown long-term annual precipitation we suggest to add a panel to additionally show the MFI in Fig.1.

The rainfall seasonality is now considered in the revised version of the manuscript as follows:

In Fig. 1 an additional plot panel was included showing the Seasonality Index (SI; Walsh and Lawler, 1981), as a measure for the seasonality of the rainfall.

The set of realizations for the rainfall erosivity R now includes two additional realizations that were presented in Fenta et al. (2017) where one method employs long-term mean annual precipitation to calculate R and the second method uses the MFI to account for the seasonality of the rainfall. Fenta et al. (2017) applied both methods in a large scale study in Eastern Africa. As a result 972 realizations of the USLE model are analyzed in the revised version of the manuscript, compared to the 756 realizations that were presented in the previous version of the manuscript.

*P8-L8 and supplementary Table S1: It seems you only used relationships based on mean annual precipitation to estimate the R factor (not accounting for seasonality). It would have been interesting to include an equation based on the monthly data, for example those based on the Modified Fournier Index proposed by Renard & Freimund(1994) that you cite. The text above Table S1 states 'The ïˇn ¸Arst four methods' which should be the 'first five'.*

The simple reason why primarily long-term annual precipitation was implemented to calculate the rainfall erosivity factor, was that the literature on large scale soil loss assessments as well implemented primarily long-term annual precipitation products to calculate the rainfall erosivity. We agree that a comparison of long-term annual precipitation to the MFI can provide valuable insight. We suggest to apply the MFI (Arnoldus, 1980) for the study region and compare the results with long-term annual precipitation to put it into reference. We suggest to add any analysis in the supplementary materials.

We will change 'first four' to 'first five' in the text above Table S1.

Please see the comment above how rainfall seasonality is accounted for in the revised version of the manuscript.

Due to the additions in the R factor input set, the section in the supplementary document was substantially revised.

*Supplementary page 8 (above table S6) correct 'To compute the K factor realizations..'for 'To compute the LS factor realizations'.*

Thank you. This will be changed accordingly.

This was changed accordingly.

*P9L30: please correct sentence 'served as base layers for the join with..'*

P9 L29 - L31 will be modified as follows:

Two land cover products, the MODIS Collection 5 LC with a spatial resolution of 250m (Channan et al., 2014; Friedl et al., 2010) and the ESA CCI LC Map v2.0.7 with a spatial resolution of 300m (ESA, 2017) served as base land cover layers. The agricultural, forest, and naturally vegetated land cover in these maps were superimposed with C factor literature values. The C factor values for agricultural land uses were calculated based on agricultural statistics.

The sentence on P9 L29ff was replaced with the suggested sentence in the revised version of the manuscript.

*Table S7: what does the first column 'value' mean?*

The value represents the crop group ID which is the same as in the following table. Therefore, the column name "value" will be changed to "Group ID" in the tables S.7 and S.8

The column names in the tables S.7 and S.8 were changed accordingly.

*P12-L16: this may be interesting, but where exactly do we find the results of this? I couldn't find it in the results section.*

The calculated values for the sensitivity index for all four USLE inputs was used to rank the inputs. Therefore, the calculated sensitivities values are not shown. Due to the length of the manuscript we decided to only keep Fig. 6 that shows the most dominant input in each grid cell. We suggest to add a figure that shows the sensitivity indices for all four USLE inputs in the supplementary materials.

The following figure was added in the supplementary document:

[Figure]

Figure S.4: Results for the sensitivity index calculated for all four USLE input factors.

*P12-L19: it is not clear from this paragraph how the comparison of soil loss rates at the administrative level contributes to the papers objectives expressed in the introduction.*

Please see our reply to the comment on the analysis of soil loss on the administrative level above.

*P14-L29: With 'the dominant soil loss levels that a majority of model setups predicted' you refer to the soil loss level for which most agreement was between the model setups? What if there was no majority for any of the soil loss levels? Unfortunately, in the figure 5, the lightness of the colours that should indicate the percentage of models that calculated a soil loss within the respective soil loss classes, cannot be distinguished.*

Numerically there should be a dominant input when we apply Eq. 4 to calculate the sensitivities. The only exception is when several inputs are 0 and therefore both inputs have a sensitivity of 1. This should however not be the case as no input is exactly 0 in any grid cell. Nevertheless, we agree that Fig. 6 does not show if two inputs are almost equally

relevant. Therefore, we suggest to add a figure that shows the individual sensitivity indices in the supplementary document as suggested above.

Further, we try to improve the readability of Fig. 5 to better distinguish between the percentages of agreement.

We tried to improve the readability of Fig. 5 by reducing to only two classes (0-50% and 50-100% of the models that agree on a soil loss level) that indicate the model ensemble agreement.

*Figure 7: in the heading it states that the values refer to those pixels for which 'high to severe soil loss was predicted to be likely'. How is 'to be likely' defined here? Or does this refer to high or severe soil loss as predicted per model implementation?*

This sentence in the figure caption will be changed to:
The cases A) to D) include the values of input factor realizations for grid cells, in which the respective input factor was the most sensitive one and the majority of models of the model ensemble predicted high to severe soil loss.

The sentence in the figure caption of Fig. 7 was replaced accordingly.

*P20-L20: why do you highlight and compare with the data from Karamage et a (2017)? Did you introduce this in methods? I don't see the added value, especially considering that the data are already covered within your model implementations, so it seems there is nothing new.*

Karamage et al. (2017) performed a soil loss assessment for the districts of Uganda. Therefore, a comparison of the model ensemble that we have calculated to their results can be easily performed. The comparison to the results of Karamage et al. (2017) is relevant, because it illustrates the central issue with soil erosion studies that we want to address in this manuscript. Karamage et al. (2017) implemented a single USLE model setup. The comparison with the model ensemble highlights that the USLE input combination that was implemented in Karamage et al. (2017) results in soil loss estimates that are in some cases even lower than the interquartile range that is provided by the USLE model ensemble. When single USLE model setups are implemented in erosion studies this information is simply not available.

*P23-L18: But you did not really perform a plausibility check of the individual USLE model realisations, so the argument does not make too much sense.*

We admit that the individual generated input realizations were not analyzed for plausibility. All of the implemented methods were however implemented in previous peer reviewed studies that had similar study settings. Therefore, we considered all or the implemented methods as potential methods to compute the USLE input factors.
We agree that this statement implies that we thoroughly checked all input realizations for plausibility. Thus, we suggest to remove this statement.

This paragraph was completely revised and this statement is removed.

*P24-L8: the comparison with one particular study does not contribute anything to this interpretation; the wide variety between your results indicates that you cannot take conclusions based on only 1 model implementation and that an ensemble approach makes more sense.*

We provide the comparison to the study of Karamage et al. (2017) as an example here that describes, however, a general problem. In principle P24 L8-L10 supports the statement outlined by the Anonymous referee #3.

*P24-L15-18: This sentence misses a conclusive statement. Indeed, the tolerable soil loss is controversial and does not seem to add much to your assessment.*

We will add the following statement:

The terms that represent certain ranges of soil loss, such as "tolerable", or "moderate" must therefore be interpreted carefully.

The proposed statement was added accordingly.

*P24-L26-27: please correct this sentence.*

The sentence will be revised accordingly:

Fig. 6 illustrated the most dominant USLE input factor realizations with respect to their impact on the uncertainties of the calculated soil loss. The dominant input factors revealed spatial patterns on different spatial scales.

The sentence on P20 L26-27 was replaced by the proposed sentence accordingly.

*P24-L28-20. The detail in the patterns is not a property of the factor, or how important the factor is, but it just reflects the level of spatial variability that is present in the input data used. This does not mean anything for the relevance of one factor as compared to another or a scale influence. The interesting part of your result is the overall impact of each factor on total ensemble variability.*

We do not fully agree with this statement. Yes, the patterns which are shown in Fig. 6 are not a property of the input factors. The patterns are however a result of the variability of the input factor realizations in each grid cell. In each grid cell the analysis that is illustrated in Fig.6 assesses the uncertainty range in the calculated soil loss that is caused by each set of input realizations for the four analyzed USLE inputs and shows the input with the largest impact. If neighboring pixels form a spatial pattern then we can assume that the employed methods to compute the respective input strongly disagree. This insight can help to understand what the dominant source for the simulation uncertainties are locally. We fully agree with the statement that the shown patterns follow the patterns that are given in the used input data that are combined with the methods to calculate the USLE input factors. But that is exactly what we want to analyze with this figure.

We will revise the sentence to specify the statement accordingly:

The patterns of the most dominant inputs follow the patterns of the input data that were employed to calculate the input factor realizations. Thus, the shown patterns can support in identifying the input data/method combination that introduced the largest share of uncertainties in the calculation of soil loss locally.

The sentence on P24 L28ff was replaced by the suggested sentence accordingly.

*P25-L11:  at larger spatial scales you will need to include not only different sources of sediment (rill, gullies, mass movements), but also deposition during transport, as explained in detail by numerous previous studies.*

We will add the following statement:

At larger scales, processes other than the ones that are assessed by the USLE, such as deposition processes, gully erosion, or bank collapses have to be considered in the quantification of the soil loss (Govers, 2011).

The proposed sentence was included on P25 L11 in the revised version of the manuscript.

*P25-L20: If the data are in stream sediment loads they are certainly do not 'better meet the spatial scale of USLE'.*

We agree. The statement will be revised accordingly:

Other reference studies, such as Sutherland and Bryan (1990) or Kithiia (1997) represent the average soil loss at the catchment scale. One could assume that the spatial scale of such studies better agrees with the spatial scale of a large scale soil loss assessment with the USLE.

We revised the sentence on P25 L19 accordingly.

*P25-L26-28: Various studies have dealt in detail with the role of spatial scale in erosion assessments, and how plot scale data compare to sediment yield (e.g. de Vente et al,2007).  Further, the difference between the plot data and USLE model predictions do not have anything to do with comparing plot data with landscape scale sediment yield. Plot data and the USLE assessments in theory both consider the same erosion and deposition processes at the same scale.*

Thank you for addressing this article. We suggest to include the findings of this work (and others e.g. Sidle et al. (2017)) in the discussion on P25 as it contributes to provide a more differentiated view on the comparison of in-field data to our model ensemble calculations.

*P25-L31-33: I think quantitative validation via google earth will be difficult and you do not really explain how this could be done.*

It was not our intention to provide new approaches for a USLE model evaluation in this manuscript. The key message was that model evaluation as it is frequently done is strongly limited due to the arguments that we have stated in the discussion. Yet, we wanted to indicate that we have to think of other potential options to evaluate soil loss estimates and simply provided the method described in Bosco et al. (2014) as an example.

*P26-L9: ULSE = USLE*

This will be changed accordingly.

The typo was changed accordingly.

*P26-L12-14: computer capacity for these kind of calculations should nowadays for most studies not be a problem anymore.*

Our analyses were indeed limited by the computational resources that were available at our Institute, at least when time resources are taken into account as well. Adding additional input factor realizations would not have been feasible with our available resources, as the required storage capacities would easily have exceeded 50+ TB.

Based on our reply we kept the statement on P26 L12-14 in the previous version of the manuscript also in the revised version of the manuscript, as particularly RAM storage can be a limiting factor in such a study design when employed in large scale applications.

*P26-L14: Ideally yes, like in any model you need to validate the predictions. But, how do you determine the plausibility if you don't have field data to compare with? Based on your assessment and comparison with field data would you say that the USLE assessments are plausible? You need data that can be compared with the USLE predictions, so representative for the same scale. I think the main interest is in the fact that the ensemble prediction shows relatively good agreement in the severity class of erosion, but quantitative validations are problematic.*

We agree that the statement on P26 L14 misses a clear argument on how to check the plausibility. We think that it overall does not contribute much to the paragraph. Thus, we suggest to delete this sentence.

The sentence: "Nevertheless, checking the plausibility of estimated soil loss must be the minimum requirement for every study employing the USLE (see suggestion above and Bosco et al., 2015, 2014)." was deleted accordingly.

*P27-L4: please rephrase and simplify the sentence.*

Will be rephrased as follows:

We generated sets of realizations for each USLE input factor and combined them to 756 USLE model setups to compute spatially distributed soil loss estimates for Kenya and Uganda. Based on the generated USLE model combinations we analyzed and quantified the impacts of frequently used methods to calculate USLE inputs on the uncertainties in the soil loss estimation with the USLE model.

We replaced the sentence on P27 L2-5 in the previous version of the manuscript with the paragraph suggested above.

*P27-L11: increased soil loss = high soil loss*

This will be changed accordingly.

We rephrased accordingly.

*P27-L26-28: Most important here is to make sure that the data are comparable, so representing the same erosion and sediment transport or deposition processes. In theory, USLE predictions should compare with plot data.*

We agree that in theory this statement is true. Yet, it assumes that the employed USLE model was properly parameterized and that the used in-field data stems from a long-term plot experiment. Both assumptions might not hold in the presented study setting.

Based on our analyses we still cannot confirm the applicability of plot scale data to be employed as a reference in large scale studies. We replaced the sentence on P27 L26-28 in the previous version of the manuscript with the paragraph suggested below.

*P27-L28: this recommendation is very vague. What kind of new approaches? How would google maps provide quantitative estimates that can be compared with model predictions?*

We agree that the wording is too vague. We suggest to rephrase to:

We further question the aptitude of soil loss assessments based on in-stream sediment yields or small scale plot experiments to be valid data for the evaluation of soil loss estimates. We should think of new approaches to validate soil loss estimates that employ large scale data that is now available. Bosco et al. (2014) outline a method to employ satellite imagery to check the plausibility of large scale soil loss assessments.

We replaced the sentence on P27 L26-28 in the previous version of the manuscript with the paragraph suggested above.

[revised manuscript text omitted]